# A Random Matrix Theory Perspective on the Consistency of Diffusion Models

**Binxu Wang** [1]  **Jacob A. Zavatone-Veth** [2 3]  **Cengiz Pehlevan** [1 3 4]

## Abstract

Diffusion models trained on different, non-overlapping subsets of a dataset often produce strikingly similar outputs when given the same noise seed. We trace this consistency to a simple linear effect: the shared Gaussian statistics across splits already predict much of the generated images. To formalize this, we develop a random matrix theory (RMT) framework that quantifies how finite datasets shape the expectation and variance of the learned denoiser and sampling map in the linear setting. For expectations, sampling variability acts as a renormalization of the noise level through a self-consistent relation $\sigma^2 \mapsto \kappa(\sigma^2)$, explaining why limited data overshrink low-variance directions and pull samples toward the dataset mean. For fluctuations, our variance formulas reveal three key factors behind cross-split disagreement: *anisotropy* across eigenmodes, *inhomogeneity* across inputs, and overall scaling with dataset size. Extending deterministic-equivalence tools to fractional matrix powers further allows us to analyze entire sampling trajectories. The theory sharply predicts the behavior of linear diffusion models, and we validate its predictions on UNet and DiT architectures in their non-memorization regime, identifying where and how samples deviates across training data split. This provides a principled baseline for reproducibility in diffusion training, linking spectral properties of data to the stability of generative outputs.

---

[1]Kempner Institute, Harvard University, Boston, MA, USA [2]Society of Fellows, Harvard University, Cambridge, MA, USA [3]Center for Brain Science, Harvard University, Cambridge, MA, USA [4]John A. Paulson School of Engineering and Applied Sciences (SEAS), Harvard University, Cambridge, MA, USA. Correspondence to: Binxu Wang <binxu_wang@hms.harvard.edu>, Jacob Zavatone-Veth <jzavatoneveth@fas.harvard.edu>, Cengiz Pehlevan <cpehlevan@seas.harvard.edu>.

*Proceedings of the $43^{rd}$ International Conference on Machine Learning*, Seoul, South Korea. PMLR 306, 2026. Copyright 2026 by the author(s).

## 1. Introduction

Diffusion models and their relatives such as flow matching have become the dominant generative modeling paradigm across diverse domains, including images, video, and proteins. By learning a time-dependent vector field, these models transform Gaussian noise into structured samples through an ordinary differential equation (ODE) or its stochastic variants (Song et al., 2021; Albergo et al., 2023).

A distinctive feature of diffusion models is their striking *consistency across training runs* (Figure 1). When trained on the same distribution, even with disjoint datasets, different architectures, or repeated initializations, diffusion models often map the same noise seed to highly similar outputs under the deterministic probability flow (Kadkhodaie et al., 2024; Zhang et al., 2024). This phenomenon contrasts with other generative modeling frameworks including GANs and VAEs, where the isotropic Gaussian latent space admits arbitrary rotations, leading to run-to-run variability in the mapping from latent codes to data (Martinez & Pearson, 2022).

**Why does consistency matter?**  Consistency across non-overlapping data splits suggests that diffusion models recover aspects of the underlying *data manifold* that are insensitive to the specific training set. This raises fundamental questions about how such models generalize beyond their training samples, to what extent they memorize idiosyncratic data, and whether their outputs reflect universal statistical regularities of the distribution. These issues connect to emerging theoretical and empirical debates on generalization, memorization, and creativity in diffusion models (Kamb & Ganguli, 2024; Niedoba et al., 2024; Kadkhodaie et al., 2024; Chen, 2025; Vastola, 2025; Bonnaire et al., 2025); see also further discussion in App. A.

**Our approach.**  We analyze this phenomenon through the lens of random matrix theory (RMT), beginning with the observation that the consistency effect can already be predicted by a linear Gaussian model (Fig. 1). Building on the linear denoiser framework, we develop a precise RMT analysis of how finite-sample variability in the empirical covariance affects both the expectation and fluctuation of denoisers and sampling maps (Fig. 2**A**). We then validate these theoretical predictions against deep diffusion models

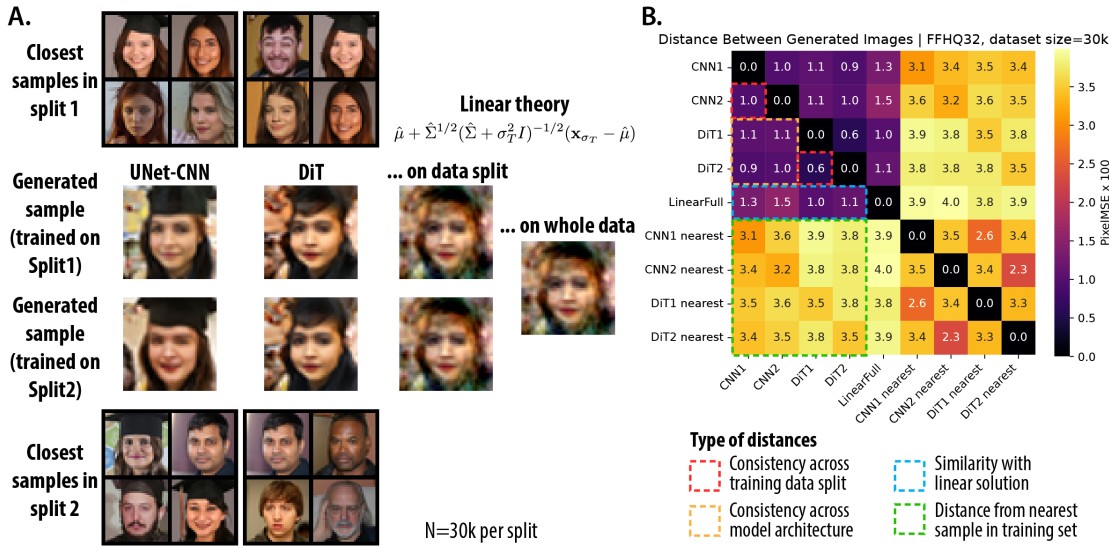

*Figure 1.* **Motivating observation and the linear theory**. **A.** Diffusion models trained on non-overlapping data splits generate similar images from the same initial noise, even with different neural network architectures, consistent with results in (Kadkhodaie et al., 2024; Zhang et al., 2024). Notably, generated samples from both splits are visually similar to the prediction from the Gaussian linear theory (Wang & Vastola, 2024b). **B.** Quantification of **A** by paired image distances (MSE) averaging from 512 initial noises. The low-MSE block structure of the four DNNs and linear solution emphasize that this consistency effect is related to the linear structure. CNN1 denotes the CNN trained on split1, similar for CNN2, DiT1, DiT2; CNN1 nearest denotes the set of closest training set sample for the 512 generated image. We hide results for linear predictor of two splits since their samples are nearly identical with the linear predictor for the full dataset. Similar analysis for FFHQ64 is showed in Fig. 6.

(CNNs and DiTs), showing that the same RMT principles still govern their inhomogeneity of consistency across data splits. Our **main contributions** are as follows:

- **Linear origin of consistency:** We show that shared Gaussian statistics i.e. linear denoiser already predict cross-split agreement.

- **Finite-sample RMT:** We prove that randomness enters through a *renormalized* noise scale $\sigma^2 \mapsto \kappa(\sigma^2)$, explaining overshrinkage of low-variance modes.

- **Variance law:** We derive a factorized form for fluctuation—explaining structures in cross-split deviation: anisotropy across eigenmodes, inhomogeneity across inputs, and global scaling with training set size.

- **Fractional-power DE:** We extend deterministic equivalence (DE) to fractional matrix powers, enabling analysis of full sampling trajectories.

- **Deep-net validation:** We qualitatively confirm overshrinkage, anisotropy, and inhomogeneity phenomenon in UNet and DiT models beyond the linear regime.

## 2. Notation and Set up

**Score-based Diffusion Models.** Let $p_0(\mathbf{x})$ be the target data distribution. For each noise scale $\sigma > 0$, define the noised distribution as $p(\mathbf{x}; \sigma) = \left(p_0 * \mathcal{N}(0, \sigma^2 \mathbf{I})\right)(\mathbf{x}) = \int p_0(\mathbf{y}) \mathcal{N}(\mathbf{x} \mid \mathbf{y}, \sigma^2 \mathbf{I}) \, d\mathbf{y}$. The corresponding *score function* is $\nabla_{\mathbf{x}} \log p(\mathbf{x}; \sigma)$, i.e. the gradient of the log–density. In

the EDM formulation (Karras et al., 2022), the probability flow ODE (PF-ODE) reads,

$$\frac{d\mathbf{x}}{d\sigma} = -\sigma \, \nabla_{\mathbf{x}} \log p(\mathbf{x}; \sigma). \qquad \text{(PF)}$$

This ODE transports samples from $p(\,\cdot\,; \sigma_2)$ to $p(\,\cdot\,; \sigma_1)$ when integrating $\sigma$ from $\sigma_2$ to $\sigma_1$. In particular, by starting from Gaussian noise $\mathcal{N}(0, \sigma_T^2 I)$ and integrating the PF-ODE from a sufficiently large $\sigma_T$ down to $\sigma = 0$, one recovers clean samples from $p_0$. We adopt the EDM parametrization for its notational simplicity; other common diffusion formalisms are equivalent up to simple rescalings of time and space (Karras et al., 2022).

To estimate the score function of $p_0(\mathbf{x})$, we minimize the denoising score matching (DSM) objective (Vincent, 2011). Parametrizing the denoiser $\mathbf{D}_\theta$ with a function approximator, then at noise level $\sigma$ the DSM objective reads

$$\mathcal{L}_\sigma = \mathbb{E}_{\mathbf{x}_0 \sim p_0, \, \mathbf{z} \sim \mathcal{N}(0, \mathbf{I})} \left\| \mathbf{D}_\theta(\mathbf{x}_0 + \sigma\mathbf{z}; \sigma) - \mathbf{x}_0 \right\|_2^2. \quad \text{(DSM)}$$

Using Tweedie's formula (Robbins, 1992), the score function can be obtained via the optimized 'denoiser' $\mathbf{s}_\theta(\mathbf{x}, \sigma) = \left(\mathbf{D}_\theta(\mathbf{x}, \sigma) - \mathbf{x}\right)/\sigma^2$. In practice (Karras et al., 2022), diffusion models balance these scale-specific objectives with a weighting function $w(\sigma)$, yielding the overall training loss $\mathcal{L} = \int_\sigma d\sigma \, w(\sigma) \mathcal{L}_\sigma$.

**Data distribution.** Consider a ground truth data distribution $p_0(\mathbf{x})$, $\mathbf{x} \in \mathbb{R}^d$, with population mean $\boldsymbol{\mu}$ and covariance $\boldsymbol{\Sigma}$. From this underlying distribution, we construct

an empirical distribution $\{\mathbf{x}_i\}$ with $n$ samples, stacked as $X \in \mathbb{R}^{n \times d}$, then we denote the empirical mean $\hat{\boldsymbol{\mu}}$ and covariance $\hat{\boldsymbol{\Sigma}}$.

Here we are interested in the effect of the number of samples $n$, and different realizations of $X$ on the expectation (mean) and fluctuation (variance) of learned diffusion model. More specifically, we will study how randomness in the empirical covariance $\hat{\boldsymbol{\Sigma}}$ drives variability in the denoiser, relative to the population covariance $\boldsymbol{\Sigma}$.

**Linear Denoiser.** A tractable setting for analytical study is the linear denoiser

$$\mathbf{D}(\mathbf{x}; \sigma) = \mathbf{W}_\sigma \mathbf{x} + \mathbf{b}_\sigma, \qquad (1)$$

which is an affine function of the noised state, independent across noise scales. As in linear regression, the training data enters the learned denoiser only through their first two moments (Hastie et al., 2022). More explicitly, minimizing DSM $\mathcal{L}_\sigma$ for the empirical dataset $p_0 = \{\mathbf{x}_i\}$[1] yields the optimal empirical linear denoiser, depending on $\hat{\boldsymbol{\mu}}, \hat{\boldsymbol{\Sigma}}$.

$$\mathbf{D}^*_{\hat{\boldsymbol{\Sigma}}}(\mathbf{x}; \sigma) = \hat{\boldsymbol{\mu}} + (\hat{\boldsymbol{\Sigma}} + \sigma^2 \mathbf{I})^{-1} \hat{\boldsymbol{\Sigma}} (\mathbf{x} - \hat{\boldsymbol{\mu}}). \qquad (2)$$

For simplicity, we will later set $\hat{\boldsymbol{\mu}} = \boldsymbol{\mu}$ to isolate the effect of the empirical covariance $\hat{\boldsymbol{\Sigma}}$.

**Sampling trajectory and sampling map.** Given an initial noise pattern $\mathbf{x}_{\sigma_T} \sim \mathcal{N}(0, \sigma_T^2 I)$, the PF -ODE evolves it to a final sample $\mathbf{x}_0$. We refer to this mapping from $\mathbf{x}_{\sigma_T}$ to $\mathbf{x}_0$ as the *sampling map*; the phenomenon of consistency is precisely about the stability of this mapping across different realizations of training data. When the denoiser is linear and optimal at each noise scale, the PF-ODE can be solved in closed-form by projecting onto the eigenbasis of the data, yielding the analytic sampling trajectory (Wang & Vastola, 2024b; Pierret & Galerne, 2024).

$$\begin{aligned} &\mathbf{x}_{\hat{\boldsymbol{\Sigma}}}(\mathbf{x}_{\sigma_T}, \sigma) \qquad (3) \\ &= \hat{\boldsymbol{\mu}} + (\hat{\boldsymbol{\Sigma}} + \sigma^2 I)^{1/2} (\hat{\boldsymbol{\Sigma}} + \sigma_T^2 I)^{-1/2} (\mathbf{x}_{\sigma_T} - \hat{\boldsymbol{\mu}}) \end{aligned}$$

Taking $\sigma \to 0$ recovers the Wiener filter with Gaussian prior (Wiener, 1964), which has been shown to be a strong predictor of the sampling map of the learned diffusion networks (Wang & Vastola, 2024b; Li et al., 2024b; Lukoianov et al., 2025). In the linear case, the mapping remains affine in the initial state, with the matrix $\hat{\boldsymbol{\Sigma}}^{1/2}(\hat{\boldsymbol{\Sigma}} + \sigma_T^2 I)^{-1/2}$ emerging as the central object of analysis.

## 3. Motivating Empirical Observation

We begin with a simple experiment illustrating the consistency phenomenon. We train UNet-CNN (Song & Ermon,

2019) and DiT (Peebles & Xie, 2023) diffusion models under the EDM framework (Karras et al., 2022), each on two non-overlapping splits of FFHQ32 (30k images each; details in App. D.3). When sampling from the same noise seed with a deterministic solver, the outputs are visually similar across both splits and architectures (Fig. 1A). Quantification via pixel MSE confirms this effect: generated images are more similar across splits than to their nearest neighbors in the training set (Fig. 1B), ruling out memorization (Kadkhodaie et al., 2024; Zhang et al., 2024).

Strikingly, the linear Gaussian predictor (Wiener filter) (Wang & Vastola, 2024b) already accounts for much of this behavior. Using the empirical mean and covariance $(\hat{\boldsymbol{\mu}}, \hat{\boldsymbol{\Sigma}})$ of each split in Eq. 3, the linear predictor yields nearly identical outputs across splits, also sharing visual similarities with CNN and DiT results (Fig. 1A,B). This suggests that consistency arises because different data splits share nearly identical Gaussian statistics, the only feature the linear diffusion can absorb (Wang & Pehlevan, 2025). Pointwise, samples closer to the Gaussian solution are also more consistent across splits (Pearson $r = 0.244, p = 5 \times 10^{-15}$), suggesting convergence toward the Gaussian predictor underlies consistency. More visual examples and quantitative comparisons for other datasets (CIFAR10, CIFAR100, FFHQ at 32 and 64 pixels, LSUN church and bedroom dataset at 32 and 64 pixels) are provided in Appendix B.1.

In summary, (i) diffusion models trained on independent splits converge to nearly identical sampling maps, (ii) this property holds across architectures, and (iii) a simple Gaussian predictor already captures much of the effect. While linear diffusion is more consistent than deep networks—which can exploit higher-order statistics—it provides a necessary baseline: *if Gaussian statistics differ, deep models may not yield consistent samples.* To test this directly, we perform a counterfactual experiment in which the training data are intentionally partitioned to induce mismatches in mean and/or variance by stratifying samples along a chosen principal component. Diffusion models trained on these statistically misaligned splits produce markedly less consistent generations (Fig. 13, 14). These observations motivate our random matrix theory analysis of finite-sample effects.

## 4. Theory of Diffusion Consistency Across Independent Data

The goal of the theoretical analysis is to calculate the expectation and covariance of various quantities in diffusion model under independent instantiation of dataset (Fig. 2**A**).

---

[1] With $n$ samples, we average over infinite noise draws, so each sample is reused infinitely.

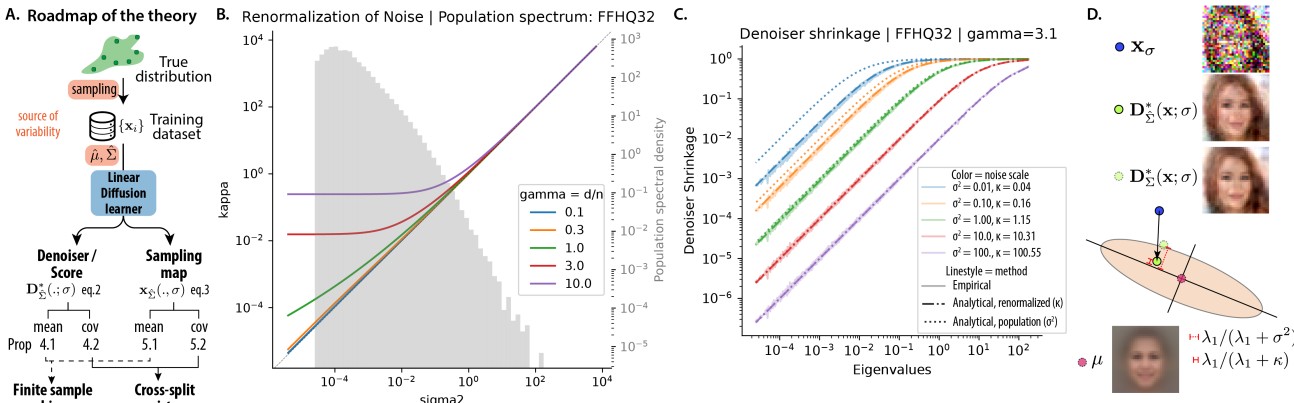

*Figure 2.* **Renormalization of noise and its effect on expectation of linear denoiser**. **A.** Roadmap of our theory. **B.** The relationship between the renormalized and raw noise variance $\kappa(\sigma^2)$ as a function of $\gamma = d/n$, using the empirical spectrum of FFHQ32 as the limiting spectrum (plot underneath). See D.1 for numerical methods. **C.** Shrinkage factor of linear denoiser along population eigenvectors at different noise scales. Empirical shows $\mathbf{v}^\top \hat{\mathbf{\Sigma}}(\hat{\mathbf{\Sigma}} + \sigma^2 I)^{-1}\mathbf{v}$, when $\mathbf{v} = \mathbf{u}_k$ population PCs, at dataset size $n = 1000, \gamma \approx 3.1$. **D.** Schematics showing the overshrinking effect at lower eigenspaces, using linear denoiser outcome of faces as example.

## 4.1. Self consistency equation and renormalized noise scale

**Deterministic equivalence of sample covariance.** Our central technical tool is deterministic equivalence (Potters & Bouchaud, 2020; Bun et al., 2016), which allows random matrices to be replaced by deterministic surrogates—an approximation that becomes exact in the large-dimensional limit. In particular, we rely on the deterministic equivalence relation for the empirical covariance matrix $\hat{\mathbf{\Sigma}}$ (Atanasov et al., 2026b; Bach, 2024),

$$\hat{\mathbf{\Sigma}}(\hat{\mathbf{\Sigma}} + \lambda I)^{-1} \asymp \mathbf{\Sigma}(\mathbf{\Sigma} + \kappa(\lambda)I)^{-1} \qquad \text{(DE)}$$

where $\kappa$ is the unique positive solution to the self-consistent equation (Silverstein, 1995; Marchenko & Pastur, 1967).

$$
\begin{aligned}
\kappa(\lambda) - \lambda &= \gamma\kappa(\lambda)\int_0^\infty \frac{s\,d\boldsymbol{\mu}(s)}{\kappa(\lambda) + s} \\
&= \gamma\kappa(\lambda)\operatorname{tr}[\mathbf{\Sigma}(\mathbf{\Sigma} + \kappa(\lambda)I)^{-1}],
\end{aligned} \qquad (4)
$$

where $\gamma = d/n$ is the aspect ratio, and $\boldsymbol{\mu}$ is the (limiting) spectral measure of $\mathbf{\Sigma}$.[2] Note we use $\operatorname{tr}$ to denote the *normalized trace*, such that $\operatorname{tr}[I] = 1$, and $\operatorname{Tr}$ the unnormalized one. More elaborate two-point deterministic equivalences (Bach, 2024; Atanasov et al., 2025; 2026a) are required to derive the variance results in the paper, which can be found in Appendix C.1.

**Property of renormalized noise $\kappa(\sigma^2)$.** As Eq. DE suggests, with trace-like measurement, the stochastic effects of sample covariance $\hat{\mathbf{\Sigma}}$ can be absorbed into the scalar $\kappa(\lambda)$

[2]We write $A_n \asymp B_n$ for *deterministic equivalence*: for any sequence of deterministic matrices $C_n$ with uniformly bounded spectral norm, $\operatorname{tr}[C_n(A_n - B_n)] \to 0$ as $d, n \to \infty, d/n \to \gamma$. Equivalences of scalar trace expressions are denoted similarly with $\asymp$.

leaving the population covariance $\mathbf{\Sigma}$ otherwise unchanged, similar to the renormalization of self-energy in field theory (Zee, 2010; Atanasov et al., 2026b; Hastie et al., 2022; Bach, 2024). In our context, $\lambda$ usually corresponds to noise variance $\sigma^2$, so we could understand $\kappa$ as the renormalized noise variance. To build intuition, we numerically evaluate this nonlinear mapping using the spectrum of natural images (FFHQ) (Fig. 2B, Method in D.1). The renormalization effect $\kappa(\sigma^2)$ is most pronounced at low noise scales, and when the sample number is much fewer than the data dimension ($\gamma = d/n \gg 1$).

**Notation** We define the degrees-of-freedom functions

$$
\begin{aligned}
\mathrm{df}_1(\lambda) &:= \operatorname{Tr}[\mathbf{\Sigma}(\mathbf{\Sigma} + \lambda I)^{-1}], \\
\mathrm{df}_2(\lambda) &:= \operatorname{Tr}[\mathbf{\Sigma}^2(\mathbf{\Sigma} + \lambda I)^{-2}], \\
\mathrm{df}_2(\lambda, \lambda') &:= \operatorname{Tr}[\mathbf{\Sigma}^2(\mathbf{\Sigma} + \lambda I)^{-1}(\mathbf{\Sigma} + \lambda' I)^{-1}].
\end{aligned} \qquad (5)
$$

We have $0 \le \mathrm{df}_2(\lambda) \le \mathrm{df}_1(\lambda) \le \operatorname{rank}(\mathbf{\Sigma}) \le d$, and $\mathrm{df}_2(\lambda, \lambda') \le \sqrt{\mathrm{df}_2(\lambda)\,\mathrm{df}_2(\lambda')}$. Moreover, on the solution of the Silverstein equation for $\lambda > 0$, we have the strict inequality $\mathrm{df}_1(\kappa) < n$.

## 4.2. Expectation: Finite Data Renormalize Noise Scales

Next we apply these tools to compute the expectation and fluctuation of the denoiser under dataset realizations. The form of Eq. 2 naturally suggests the deterministic equivalence in Eq. DE, leading to the following result.

**Proposition 4.1** (Deterministic equivalent of the denoiser expectation)**.** *Assuming $\hat{\boldsymbol{\mu}} = \boldsymbol{\mu}$, and given a fixed probe vector $\mathbf{v} \in \mathbb{R}^d$, then the optimal empirical linear denoiser has the following deterministic equivalent. (Proof in App. C.2).*

$$\mathbb{E}_{\hat{\mathbf{\Sigma}}}\left[\mathbf{v}^\top \mathbf{D}_{\hat{\mathbf{\Sigma}}}^*(\mathbf{x}; \sigma)\right] \asymp \mathbf{v}^\top \mathbf{D}_{\mathbf{\Sigma}}^*(\mathbf{x}; \sqrt{\kappa(\sigma^2)}) \qquad (6)$$

$$= \mathbf{v}^\top\left[\boldsymbol{\mu} + \mathbf{\Sigma}(\mathbf{\Sigma} + \kappa(\sigma^2)I)^{-1}(\mathbf{x} - \boldsymbol{\mu})\right].$$

**Interpretation** In expectation, finite data act by renormalizing the noise scale, $\sigma^2 \to \kappa(\sigma^2)$, in the population denoiser. This is equivalent to adding an adaptive Ridge penalty to the DSM objective. Compared to the population solution $\mathbf{D}_{\boldsymbol{\Sigma}}^*$, the finite-sample denoiser shrinks low-variance directions more aggressively, treating them as noise and pulling outputs toward the dataset mean (Fig. 2**D**). Numerically, deviations are indeed most pronounced in the lower spectrum and at lower noise levels, where the renormalization effect is the strongest (Fig. 2**C**). Since smaller noise scale is associated with generation of high frequency details in image, this result suggests these detail eigenmodes take more samples to be learn correctly, which we will confirm in next section.

### 4.3. Fluctuation: Anisotropic and Inhomogeneity of Denoiser Consistency

Next, we tackle the fluctuation due to dataset realizations, which addresses the consistency of diffusion models trained on independent data splits. We prove the following equivalence using two-point and one-point deterministic equivalence identities (Eq. 22,20, (Bach, 2024)).

**Proposition 4.2** (Deterministic equivalent of the denoiser variance). *Assuming $\hat{\boldsymbol{\mu}} = \boldsymbol{\mu}$, across dataset realizations of size $n$, the covariance of the optimal empirical linear denoiser is denoted as $\mathcal{S}_D$; at point $\mathbf{x}$, along direction $\mathbf{v}$, the variance is given by $\mathbf{v}^\top \mathcal{S}_D(\mathbf{x})\mathbf{v}$, which admits the following deterministic equivalence.*

$$\mathbf{v}^\top \mathcal{S}_D(\mathbf{x})\mathbf{v} := \mathrm{Var}_{\hat{\boldsymbol{\Sigma}}}[\mathbf{v}^\top \mathbf{D}_{\hat{\boldsymbol{\Sigma}}}^*(\mathbf{x};\sigma)] \quad (7)$$

$$\asymp \frac{\kappa(\sigma^2)^2}{n - \mathrm{df}_2\big(\kappa(\sigma^2)\big)} \underbrace{\diamond(\mathbf{v}, \kappa(\sigma^2), \boldsymbol{\Sigma})}_{anisotropy} \underbrace{\diamond(\mathbf{x} - \boldsymbol{\mu}, \kappa(\sigma^2), \boldsymbol{\Sigma})}_{inhomogeneity}$$

*where $\diamond(\mathbf{u}, \kappa, \boldsymbol{\Sigma}) := \mathbf{u}^\top (\boldsymbol{\Sigma} + \kappa I)^{-2} \boldsymbol{\Sigma} \mathbf{u}$. Proof in App. C.3.*

**Interpretation.** The variance of denoiser across dataset realizations factorizes into three interpretable components: a dependence on probe direction (*anisotropy*), a dependence on noised sample location (*inhomogeneity*), and an overall scale with $n$ and $\sigma$ (*global scaling*). Note, given the relation between score and denoiser, the covariance of score is $\sigma^{-4}\mathcal{S}_D(\mathbf{x})$, *i.e.* all results translate by scaling.

**Anisotropy in probe direction.** The anisotropy of consistency is governed by $\diamond(\mathbf{v}, \kappa, \boldsymbol{\Sigma})$. When the probe $\mathbf{v}$ aligns with a principal component (PC) $\mathbf{u}_k$ of $\boldsymbol{\Sigma}$ with eigenvalue $\lambda_k$, this reduces to $\chi(\lambda_k, \kappa) := \lambda_k/(\lambda_k + \kappa)^2$. The function $\chi(\lambda, \kappa)$ is bell-shaped in $\lambda$, uniquely maximized at $\lambda = \kappa$ with value $1/(4\kappa)$. Thus, for each noise scale, the directions of greatest uncertainty are precisely those whose variances match the renormalized noise $\kappa(\sigma^2)$ (Fig. 3**B**).

This effect is evident visually. For linear denoisers trained on non-overlapping splits of human face dataset (FFHQ), their differences follow the spectral structure of natural images (Ruderman, 1994): at high noise the deviations appear as low-frequency facial envelopes, while at low noise they shift to high-frequency specular patterns (Fig. 3**B**). Quantitatively, the MSE between two denoisers along each PC matches the variance prediction of Eq. 7, with the expected factor of two from independent sampling (Lemma C.3).

**Inhomogeneity in input location.** The inhomogeneity of denoiser variance across input space is governed by $\diamond(\mathbf{x} - \boldsymbol{\mu}, \kappa, \boldsymbol{\Sigma})$. While structurally similar to the anisotropy factor, here $\mathbf{x} - \boldsymbol{\mu}$ is drawn from the noised data distribution rather than a unit probe. Approximating $\mathbf{x} - \boldsymbol{\mu}$ as lying on the ellipsoidal shell of $\mathcal{N}(0, \boldsymbol{\Sigma} + \sigma^2 I)$, its displacement along eigenvector $\mathbf{u}_k$ has typical radius $\sqrt{\sigma^2 + \lambda_k}$. Substituting gives $\diamond(\sqrt{\sigma^2 + \lambda_k}\,\mathbf{u}_k, \kappa, \boldsymbol{\Sigma}) = (\sigma^2 + \lambda_k)\,\chi(\lambda_k, \kappa)$. Unlike the pure anisotropy factor, this expression grows monotonically with $\lambda_k$. Thus, denoiser variability is amplified for inputs displaced along high-variance modes, yielding larger uncertainty for such locations (Fig. 3**C**), which agree quantitatively with numerical results. Based on this factor, denoiser consistency can be predicted for noisy images point by point (e.g. Pearson $r = 0.94$ across noised images, at $\sigma^2 = 1$, $n = 1000$, Fig. 15).

**Global scaling with sample size.** Finally, marginalizing over all directions and noised samples yields a closed-form expression for the overall denoiser variance (Eq. 25, Fig. 3**D**). At large $n$ limit, denoiser variance scale inversely with sample number $n^{-1}$, reminiscent of classic statistical laws; while at smaller $n$, the renormalization effects modify the scaling.

**Summary.** In sum, the variance structure reveals three key effects. *Anisotropy*: uncertainty is maximized along eigenmodes whose variance $\lambda_k$ is comparable to the renormalized noise $\kappa(\sigma^2)$. *Inhomogeneity*: noised points displaced along high-variance directions experience larger uncertainty. *Scaling*: the overall variance shrinks with dataset size $n$, recovering the population model in the large-sample limit. Together, these predictions yield a detailed spatial and spectral map of where denoisers trained on different data splits are most likely to disagree.

## 5. Consistency of Diffusion Samples for Linear Denoisers

Beyond the consistency of single-step denoiser output or score, we are interested in the final diffusion sample from the same initial noise seed $\mathbf{x}_{\sigma_T}$. For linear denoisers, sampling map from initial noise to generated sample is captured by Wiener filter (Eq. 3, $\sigma = 0$). However, unlike one-

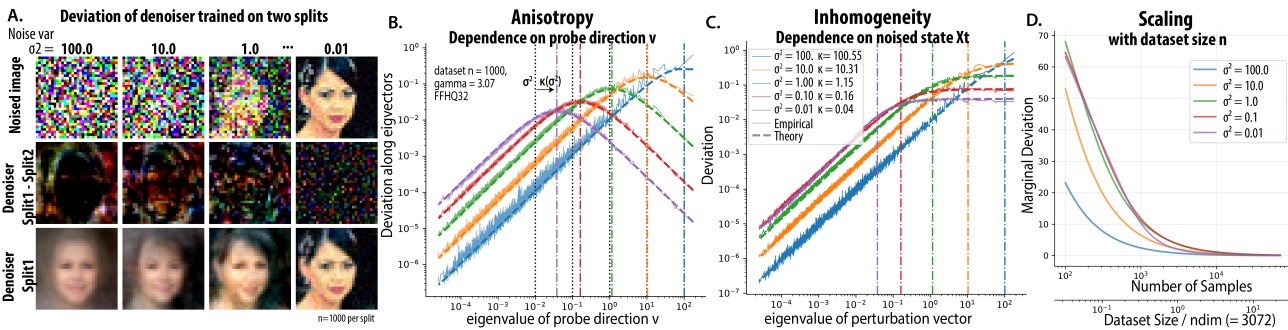

*Figure 3.* **Structure of denoiser deviation across dataset splits. A.** Visual examples of linear denoisers trained on two disjoint splits of FFHQ32 as noise variance $\sigma^2$ decreases, $n = 1000$. **Top**, $\mathbf{x}_t$ noised sample; **Bottom**, output of linear denoiser (trained on split 1) $\mathbf{D}_{\hat{\mathbf{\Sigma}}_1}(\mathbf{x}_t, \sigma)$; **Middle**, deviation between two denoisers (normalized) $\mathbf{D}_{\hat{\mathbf{\Sigma}}_1}(\mathbf{x}_t, \sigma) - \mathbf{D}_{\hat{\mathbf{\Sigma}}_2}(\mathbf{x}_t, \sigma)$. At high noise, denoisers diverge on global, low-frequency content; at low noise, they deviate at specular details. **B.** *Anisotropy:* variance depends on probe direction $\mathbf{v}$; deviation is maximized when the eigenvalue $\lambda_k$ of $\mathbf{v}$ matches the renormalized noise $\kappa(\sigma^2)$, in agreement with theory. **C.** *Inhomogeneity:* variance depends on probe location $\mathbf{x}_t$; samples displaced along high-variance eigenmodes induce larger deviations. **D.** *Global scaling:* marginal deviation decays with dataset size $n$, vanishing in the infinite-sample limit.

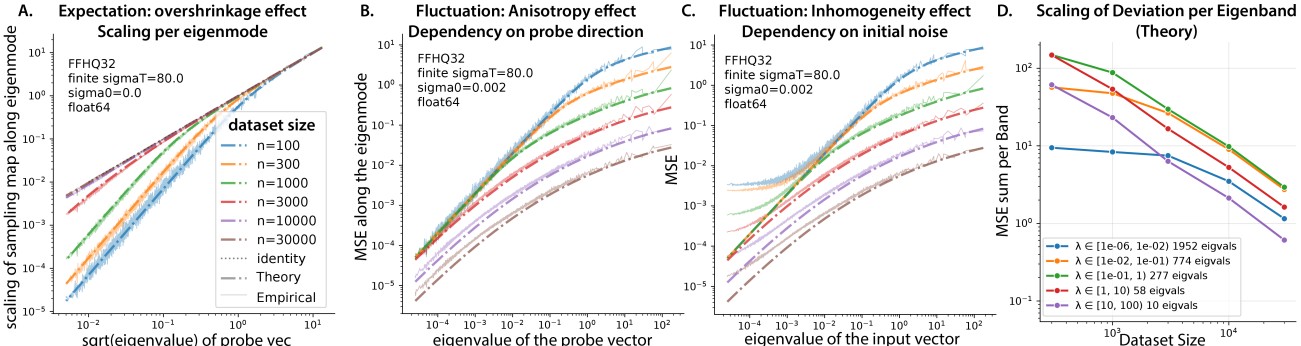

*Figure 4.* **Finite sample effect on diffusion sampling map. A. Overshrinkage of expectation**. Expected scaling along eigenmode of the empirical sampling map $\mathbf{u}_k^\top \hat{\mathbf{\Sigma}}^{1/2} \mathbf{u}_k$ compared to the ideal $\sqrt{\lambda_k}$, showing overshrinking along lower eigenmodes. **B. Anisotropy of consistency**. Cross-split MSE depends on probe direction $\mathbf{v}$, with larger deviation on top eigenspaces. **C. Inhomogeneity of consistency.** Cross-split MSE depends on input location $\bar{\mathbf{x}}$; samples displaced along high-variance modes exhibit larger disagreement. Colors denote dataset size, shared across **A,B,C**. **D. Scaling of consistency by eigenband.** Decomposition of MSE across eigenbands shows that lower-variance modes require substantially more samples before cross-split MSE decays. See also Fig. 16.

step denoiser, this mapping involves fractional power of covariances $\mathbf{\Sigma}^{1/2}(\mathbf{\Sigma} + \sigma^2 I)^{-1/2}$, for which the deterministic equivalence is not readily available. Here, we leveraged the integral representation of fractional power (Balakrishnan (1960)'s formula) and deterministic equivalence, and arrived at a few novel equivalence of these matrices (Prop. C.6, C.8, Proof in App. C.4). Using these developments, we can calculate the expectation and fluctuation of sampling map.

### 5.1. Expectation of diffusion sample: over-shrinkage to the mean

We note that when the initial noise scale $\sigma_T$ is large, the sampling map admits the approximation

$$\mathbf{x}_{\hat{\mathbf{\Sigma}}}(\mathbf{x}_{\sigma_T}, 0) = \boldsymbol{\mu} + \hat{\mathbf{\Sigma}}^{1/2}(\hat{\mathbf{\Sigma}} + \sigma_T^2 I)^{-1/2}(\mathbf{x}_{\sigma_T} - \boldsymbol{\mu})$$

$$\approx \boldsymbol{\mu} + \hat{\mathbf{\Sigma}}^{1/2}\bar{\mathbf{x}}, \qquad (8)$$

where we denote the shift and normalized noise $\bar{\mathbf{x}} := \frac{\mathbf{x}_{\sigma_T} - \boldsymbol{\mu}}{\sigma_T}$. At the $\sigma_T \to \infty$ limit, this approximation becomes exact, and $\bar{\mathbf{x}} \sim \mathcal{N}(0, I)$. For clarity, we present results under this infinite-$\sigma_T$ approximation; the expressions accounting for finite $\sigma_T$ effects are provided in App. C.6.

**Proposition 5.1** (Deterministic equivalence for expectation of diffusion sampling map)**.** *The sample generated from initial state $\mathbf{x}_{\sigma_T}$ has the following deterministic equivalence. Proof in App. C.5.*

$$\mathbb{E}_{\hat{\mathbf{\Sigma}}}[\mathbf{x}_{\hat{\mathbf{\Sigma}}}(\mathbf{x}_{\sigma_T}, 0)] \approx \boldsymbol{\mu} + \mathbb{E}_{\hat{\mathbf{\Sigma}}}[\hat{\mathbf{\Sigma}}^{1/2}]\frac{\mathbf{x}_{\sigma_T} - \boldsymbol{\mu}}{\sigma_T} \qquad (9)$$

$$\asymp \boldsymbol{\mu} + \frac{2}{\pi}\int_0^\infty \mathbf{\Sigma}\left(\mathbf{\Sigma} + \kappa(u^2)I\right)^{-1}\bar{\mathbf{x}}\, du.$$

**Interpretation** This expression mirrors the deterministic equivalence of denoisers (Eq. 6), but with an integration over effective noise scales. Comparing to the population sampling map, where $\kappa(u^2)$ reduces to $u^2$, the finite data

case integrates over a stronger shrink factor $\mathbf{\Sigma}(\mathbf{\Sigma} + \kappa I)^{-1}$ (since $\kappa(u^2) > u^2$), especially on the lower eigenmodes. This effect is confirmed with numerics of empirical covariance (Fig. 4**A**). This leads to a systematic overshrinkage toward the dataset mean along these modes, reducing the generated variance along lower-variance directions [3].

### 5.2. Variance of diffusion sample: Anisotropy and inhomogeneity

**Proposition 5.2** (Deterministic equivalence for variance of diffusion sampling map). *Due to dataset realization, the variance of generated sample starting from initial state $\mathbf{x}_{\sigma_T}$, along vector $\mathbf{v}$ admits the following deterministic equivalence,*

$$\mathrm{Var}_{\hat{\mathbf{\Sigma}}}[\mathbf{v}^\top \mathbf{x}_{\hat{\mathbf{\Sigma}}}(\mathbf{x}_{\sigma_T}, 0)] = \mathrm{Var}_{\hat{\mathbf{\Sigma}}}[\mathbf{v}^\top \hat{\mathbf{\Sigma}}^{1/2} \bar{\mathbf{x}}] \quad (10)$$

$$\asymp \frac{4}{\pi^2} \int_0^\infty \int_0^\infty \frac{\kappa \, \kappa'}{n - \mathrm{df}_2(\kappa, \kappa')} \underbrace{\bigcirc(\mathbf{v}; \kappa, \kappa', \mathbf{\Sigma})}_{anisotropy} \underbrace{\bigcirc(\bar{\mathbf{x}}; \kappa, \kappa', \mathbf{\Sigma})}_{inhomogeneity} \, du \, dv$$

*where $\bigcirc(\mathbf{a}; \kappa, \kappa', \mathbf{\Sigma}) := \mathbf{a}^\top \mathbf{\Sigma} \, (\mathbf{\Sigma} + \kappa I)^{-1} (\mathbf{\Sigma} + \kappa' I)^{-1} \, \mathbf{a}$, and $\kappa := \kappa(u^2), \kappa' := \kappa(v^2)$ are variables to be integrated over. Proof in App. C.7.*

**Interpretation** The variance of sampling map Eq. 10 simplifies to a double integral of the denoiser-variance (Eq. 7). The integrand factorizes into a direction-dependent term (*anisotropy*), a initial noise-dependent term (*inhomogeneity*), and a scaling term. Note the anisotropy and inhomogeneity factors rely on the same $\bigcirc(.; \kappa, \kappa', \mathbf{\Sigma})$ function, showing that dependency on $\mathbf{v}$ and $\bar{\mathbf{x}}$ has the same spectral structure.

We resort to numerical simulation to provide more intuition. We note that integrals in Eqs. 9,10 are nontrivial to evaluate; we describe our numerical scheme in App. D.1. Using this procedure, the theoretical predictions align closely with direct computations of linear diffusion (Fig. 4). **Inhomogeneity.** Spatially, when initial noise $\bar{\mathbf{x}}$ deviates more along the top eigenspace of $\mathbf{\Sigma}$, there will be larger uncertainty (Fig. 4**C**), this enables us to predict the sample difference point by point. **Anisotropy.** Directionally, the dependency on $\mathbf{v}$ has the same structure, in absolute term, the deviation is larger at higher eigenspace (Fig. 4**B**). Note that when scaling up the dataset size, the variance in the top eigenspace decay immediately from small sample size; while the deviation in lower eigenspace will stay put and start decaying only later at larger dataset size (Fig. 4**D**). This shows that the fine detail of the samples needs a larger dataset size to be consistency across training.

---

[3]Though the sample covariance $\hat{\mathbf{\Sigma}}$ is an unbiased estimator of the population covariance $\mathbf{\Sigma}$, taking the square root introduces finite sample bias: $\mathbf{\Sigma} = \mathbb{E}[\hat{\mathbf{\Sigma}}] = \mathbb{E}[\hat{\mathbf{\Sigma}}^{1/2}\hat{\mathbf{\Sigma}}^{1/2}] \neq (\mathbb{E}[\hat{\mathbf{\Sigma}}^{1/2}])^2$.

## 6. Validating Predictions on Deep Networks

Finally, given that linear diffusion behavior is well captured by our random matrix theory (RMT), we test the applicability of its prediction to practical deep diffusion networks.

**Setup.** We trained UNet- and DiT-based denoisers under the EDM framework on FFHQ64, FFHQ32, AFHQ32 (Choi et al., 2020), LSUN church and bedroom at 32 and 64 pixels (Yu et al., 2015), CIFAR10, and CIFAR100 (UNet on all; DiT on a subset). For each dataset we trained on two non-overlapping splits at sizes $n \in \{300, 1000, 3000, 10^4, 3 \cdot 10^4\}$ (10 runs total per architecture). Sampling was performed with the same random seed using the Heun solver (Karras et al., 2022). We train for 50,000 steps with Adam optimizer, further details are provided in App. D.3.

**Expectation: from memorization to renormalization.** We observe a clear two-phase behavior as dataset size increases. *Memorization phase* ($n \leq 1000$): models largely reproduce training samples (Fig. 5**A,B**), and samples are much closer to the nearest neighbor in their training split than the control split, consistent with prior observations. This regime is outside the scope of linear theory, since linear score models cannot memorize individual points (Wang & Pehlevan, 2025). *Renormalization phase* ($n \geq 3000$): the samples have comparable distance to the neighbor in the training split and control split, showing generalization. Further, as $n$ grows, samples increasingly resemble and approach the linear predictors (Fig. 23) (Li et al., 2024b). In this regime, the overshrinkage predicted by Prop. 5.1 becomes visible: generated face samples resemble the average face (Langlois et al., 1994), with smoother textures and background (Fig. 5A, $n = 3000$). Quantitatively, we observe reduced variance along low- and mid-spectrum eigenmodes of the generated samples (Fig. 5 **D**). This bias decreases as dataset size increases, and vanishes when learned and population spectra coincide at $n \sim 30000$. The same transition occurs across architectures, though the dataset size at which it occurs depends on model capacity and image resolution (Fig. 17,18,19).

**Fluctuations: inhomogeneity of consistency.** Within the renormalization phase, RMT further predicts which initial noise and direction exhibit the largest discrepancies across data splits, due to their alignment with data covariance (Prop. 5.2). Spectrally, measuring the cross-split deviation along population eigenbases, we can see the characteristic anisotropy profile. Further the decrease of MSE majorly occurs in top eigenspace, while the middle or lower eigenspace remains unchanged or becomes less consistent when sample size increases (Fig. 5 **E**). This is consistent with the prediction of the theory that lower eigenmodes needs more training samples to be consistent (Fig. 4**B**). Spatially, the inhomo-

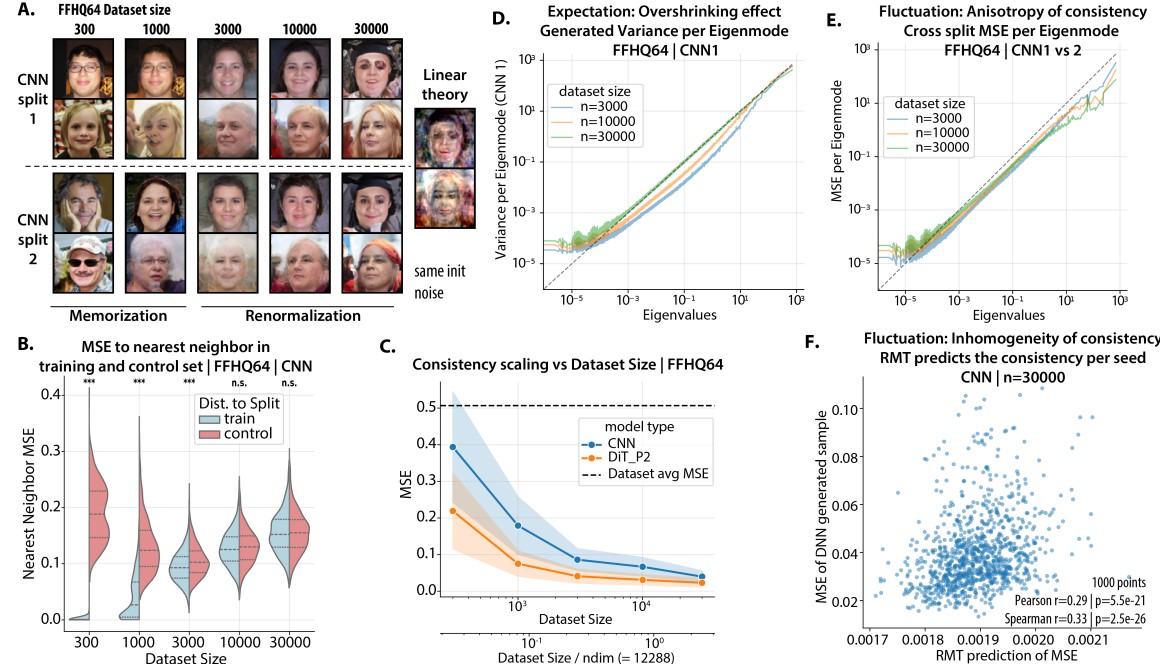

*Figure 5.* **DNN validation of theory. A.** Samples generated by UNet (same two seeds) across training set sizes and splits (FFHQ64); similarity increases with $n$, and increasingly matches the population linear predictor (right). **B.** Nearest-neighbor MSE in training vs. control sets reveals memorization at small $n$, $n > 3000$ shows no statistical difference between the splits. **C.** Overall consistency improve as a function of dataset size, with DiT more consistent than UNet at each $n$ (cross split MSE, mean±std). **D.** Variance of generated samples per eigenmode highlight insufficient variance (*overshrinkage*) in mid-to-low eigenmodes with limited dataset size. **E.** Cross-split MSE per eigenmode shows *anisotropy* of consistency (Fig. 4**B**). Further, per dataset size, deviation in top eigenmodes decrease the most. **F.** In the renormalization regime ($n = 30$k), RMT predictions of seed-wise consistency correlate with empirical deviations.

geneity effect is borne out: RMT predictions correlate with observed cross-split deviations for each initial noise point by point; e.g., UNets trained on FFHQ64 with $n = 30000$ achieve a Spearman correlation of $0.33$ ($p = 2.5 \times 10^{-26}$) over 1000 seeds (Fig. 5**F**). Remarkably, the prediction requires only the population covariance and dataset size, with no knowledge of split identities or network architecture. The absolute deviation magnitudes, however, are much larger in deep networks than predicted by linear theory, reflecting nonlinear source of variability. As controls, correlations collapse in the memorization regime (Fig. 33,34) and disappear when mismatched noise seeds are used.

**Summary.** Across architectures and datasets, the predictions of our linear RMT framework extend to deep diffusion models: limited data induce overshrinkage toward the mean, and the variance structure across splits exhibits inhomogeneity and anisotropy predicted by theory.

## 7. Discussion

Our analysis shows that much of the consistency in diffusion models across training data is already captured by Gaussian statistics, which inspires an in-depth analysis of consistency in linear diffusion. Random matrix theory sharpens this picture by showing that finite data act through a renormalized noise scale $\sigma^2 \mapsto \kappa(\sigma^2)$, and that fluctuations across splits factor into anisotropy over eigenmodes, inhomogeneity across inputs, and a global scaling with dataset size. These results extend deterministic-equivalence tools to fractional matrix powers, allowing closed-form predictions for both denoisers and sampling trajectories, and align qualitatively with deep networks in terms of where deviation accentuates, even if nonlinear effects amplify the magnitudes of deviation.

**Why are diffusion models special?** Previous work suggests that consistency and reproducibility are distinctive features of diffusion and flow-based models with deterministic samplers (Zhang et al., 2023). Why should this be the case? In diffusion models, the function approximator learns a score vector field $\nabla \log p(\mathbf{x}; \sigma)$, which is uniquely determined by the data distribution.[4] The lowest-order approximation to this score is a linear vector field determined by the Gaussian statistics of the data, namely its mean and covariance. Our random matrix analysis shows that these Gaussian statistics, and observables derived from them (e.g. denoiser and sampling map), are highly stable across in-

---

[4]For finite data, the underlying population distribution is unknown, so there remains ambiguity in how the model generalizes beyond the empirical distribution; nevertheless, the score of the empirical distribution itself is uniquely defined.

dependent data splits. Moreover, at high noise levels, the linear Gaussian score approximation is often accurate; since the high-variance, low-frequency aspects of samples are determined primarily at these high-noise scales, even non-linear score networks can inherit substantial stability from this shared Gaussian structure. This provides a mechanism by which independently trained diffusion models can share a common mapping from initial noise to generated samples.

This mechanism is less directly applicable to other generative modeling paradigms. In autoregressive models, randomness typically enters through sequential sampling from conditional token distributions at nonzero temperature, so repeated generation depends on a chain of stochastic discrete choices rather than a deterministic flow from a fixed initial noise. In VAEs and GANs, samples are generated by a learned map from a usually lower-dimensional latent space, $x = f(z)$ with $z \sim \mathcal{N}(0, I)$. This latent space is not identifiable: composing $f$ with an orthogonal transformation of $z$ preserves the latent and generated distributions, but changes which latent vector corresponds to which generated sample. Thus, two independently trained models may learn rotated latent coordinate systems, so the same latent vector need not produce similar outputs. One could in principle align the latent spaces of VAEs or GANs by fixing this orthogonal freedom. For example, latent vectors can be expressed in a meaningful local basis, such as the right singular vector basis of the generator Jacobian $\partial f / \partial z$. Previous work showed that, in this basis, axes of different GANs trained on human face datasets can have relatively consistent semantics, such as face orientation or background changes (Wang & Ponce, 2021, Fig. 10). A full analysis of consistency in VAEs or GANs is left for future investigation.

**Limitations** At the same time, our framework has limitations. Linear surrogates underestimate variability in expressive models and do not capture architecture-specific inductive biases. Extending the theory to random-feature models would better explain the transition from memorization to renormalization (Bonnaire et al., 2025; George et al., 2025), and help quantify how model capacity shifts the required dataset size for consistency. Another promising direction is to study the *anisotropy of the initial noise space* and its alignment with the data manifold. The seemingly unstructured noise space is already anchored by the data covariance before generation. Manipulating noise components along dataset principal-component subspaces leads to interpretable changes in the sampled outputs, such as suppressing background structure or reducing specific facial variations (Fig. 36, 37). Our analysis shows that different directions in the noise space contribute unequally to cross-split deviations (Prop. 5.2); consequently, the alignment between the initial noise and the covariance eigenframe predicts the degree of generation consistency across dataset splits. Such

anisotropic structures might explain why certain "magic" random seeds may consistently yield better generations (Xu et al., 2025). This echoes anisotropic effects observed in GANs' latent space, where noise vectors aligned strongly with top eigenspaces of Jacobian can lead to degraded generations (Wang & Ponce, 2021). Such connections suggest that spectral geometry of the input space deserves closer attention as a unifying factor across generative models.

## Impact Statement

This paper presents work whose goal is to advance the field of Machine Learning. There are many potential societal consequences of our work, none which we feel must be specifically highlighted here.

### Acknowledgments

B.W. was supported by the Kempner Fellowship from the Kempner Institute at Harvard. J.Z.-V. was supported by a Junior Fellowship from the Harvard Society of Fellows. C.P. is supported by an NSF CAREER Award (IIS-2239780), DARPA grants DIAL-FP-038 and AIQ-HR00112520041, the Simons Collaboration on the Physics of Learning and Neural Computation, and the William F. Milton Fund from Harvard University. We thank David Alvarez-Melis for his insightful suggestions on the counterfactual moment manipulation experiments. This work has been made possible in part by a gift from the Chan Zuckerberg Initiative Foundation to establish the Kempner Institute for the Study of Natural and Artificial Intelligence, and by the generous computing resources provided by the Kempner Institute and Harvard Research Computing.

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

# Contents

# A. Extended Related Work

**Consistency and Reproducibility in Diffusion**    As a motivating observation, (Kadkhodaie et al., 2024) found that diffusion models trained on non-overlapping splits of training data could produce visually highly similar images. The seminal paper studying this effect is (Zhang et al., 2024); there, the authors found that models trained on the same dataset have a consistent mapping from noise to sample across architectures (transformer vs. UNet), objectives, training runs, samplers, and noising kernels, provided that an ODE deterministic sampler is used. In their Appendix B, they also discuss the lack of reproducibility in VAEs and GANs. The consistency studied in our paper is more closely related to reproducibility in the generalization regime.

**Hidden Linear Score Structure in Diffusion Models**    Recent work has shown that, for much of diffusion time (*i.e.*, across a broad range of signal-to-noise ratios), the learned neural score is closely approximated by the linear score of a Gaussian fit to the data, which is usually the best linear approximation (Wang & Vastola, 2023; Li et al., 2024b). Crucially, this Gaussian linear score admits a closed-form solution to the probability-flow ODE, which can be exploited to accelerate sampling and improve its quality (Wang & Vastola, 2024a). Moreover, this same linear structure has been linked to the generalization–memorization transition in diffusion models (Li et al., 2024b). In sum, across many noise levels, the Gaussian linear approximation captures many salient aspects of the learned score. Here, we leverage it to explain the observed consistency across splits and as a tractable setup for random matrix theory analysis.

**Memorization, Generalization and Creativity in Diffusion**    The question of when diffusion models can generate genuinely novel samples matters both scientifically and for mitigating data leakage. From the score-matching perspective, if the learned score exactly matches that of the empirical data distribution, then the reverse process reproduces that empirical distribution, and thus does not create new samples beyond the training set (Kamb & Ganguli, 2024; Li et al., 2024a; Wang & Vastola, 2024b). Yet high-quality diffusion models routinely generate images that are not identical copies of images from the training set. Kamb & Ganguli (2024) take an important step toward reconciling this: when the score network is a simple CNN, its inductive biases (locality and translation equivariance) favor patch wise composition, enabling global samples that are novel while remaining locally consistent "mosaics." Similarly, Wang & Pehlevan (2025) observed that score networks with different architectural constraints learn different approximations of the dataset and therefore generalize differently: e.g., linear networks learn the Gaussian approximation, and circular convolutional networks learn the stationary Gaussian process approximation. Finn et al. (2025) provided evidence that adding a final self-attention layer promotes global consistency across distant regions, organizing locally plausible features into coherent layouts that move beyond purely patch-level mosaics. This result is consistent with preliminary observations by Kamb & Ganguli (2024) regarding cases in which their purely convolutional models fail to generate coherent images, while models including attention succeed. Related theoretical work further probes why well-trained diffusion models can generalize despite apparent memorization pressures (Bonnaire et al., 2025; Vastola, 2025; Chen, 2025). These results suggest that departures from exact empirical-score fitting—mediated by inductive biases (both architectural and training dynamics) can explain how diffusion models avoid pure memorization while maintaining visual plausibility (Ambrogioni, 2023).

# B. Extended Results and Figures

## B.1. Extended visual examples for the motivating observation

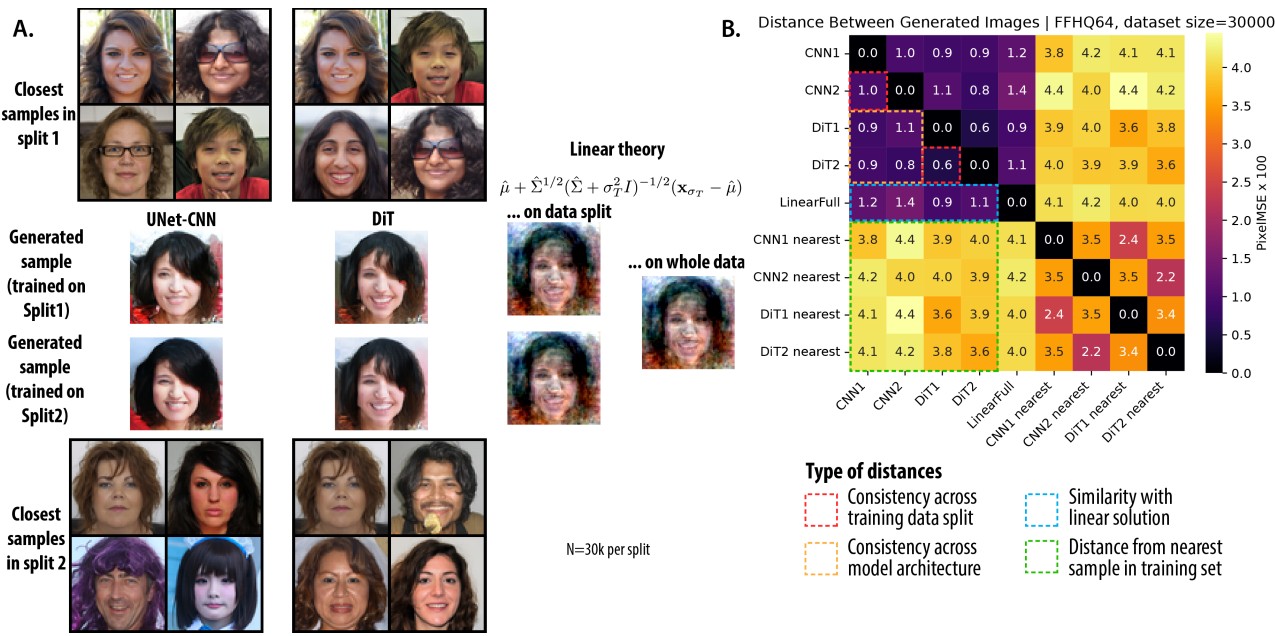

*Figure 6.* **Motivating observation and the linear theory for FFHQ64 dataset**. Similar format to Fig. 1, but for FFHQ64 dataset. **A.** Examples of generated samples from the same noise seed, for UNet, DiT, and a linear denoiser on split 1 and split 2 of the data, each with 30k non-overlapping samples. The closest 4 samples in the corresponding training set are shown above and below the generated sample. One can appreciate the visual similarity of samples generated from models trained on separate splits, even with different neural architectures, and also with the linear denoiser on each split. Admittedly, the generated outcomes of linear denoisers at 64-pixel resolution look worse, especially around edges, showing signatures of non-Gaussian statistics, as (Wang & Vastola, 2024b) has pointed out. **B.** Quantification of **A**, paired image distances (MSE) averaging from 512 initial noises.

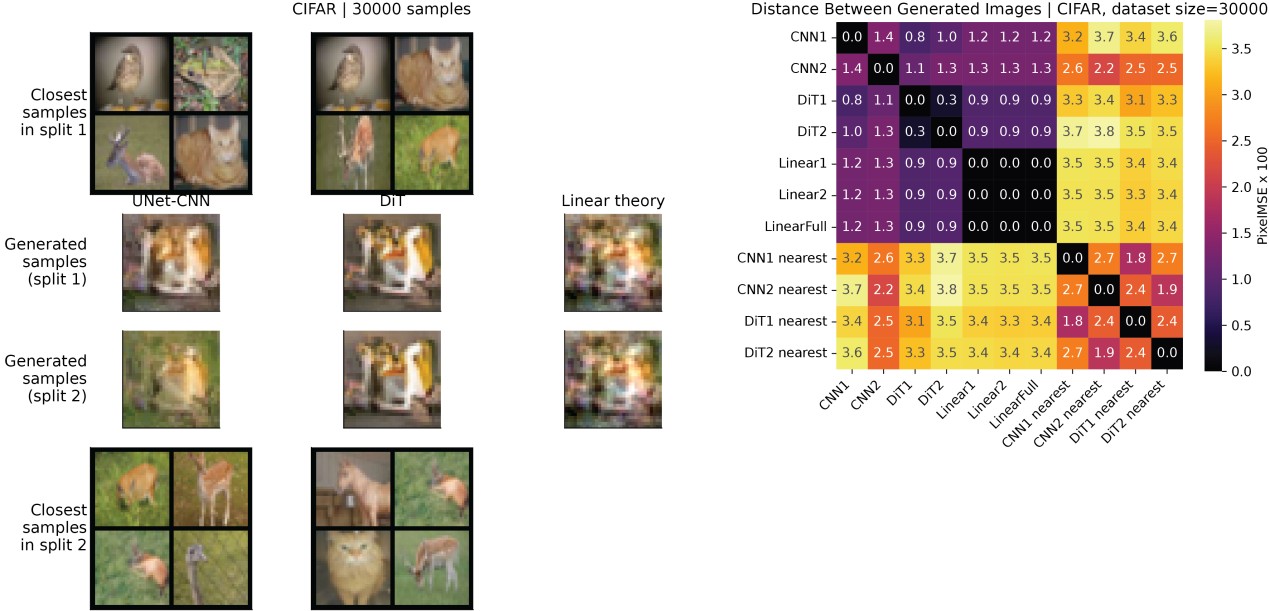

*Figure 7.* **Motivating observation and the linear theory for CIFAR10 dataset**. Similar format to Fig. 1. **Left.** Generated samples from DNN and linear theory from initial noise seed 2. **Right.** Paired image distance MSE averaging from 1000 initial noises.

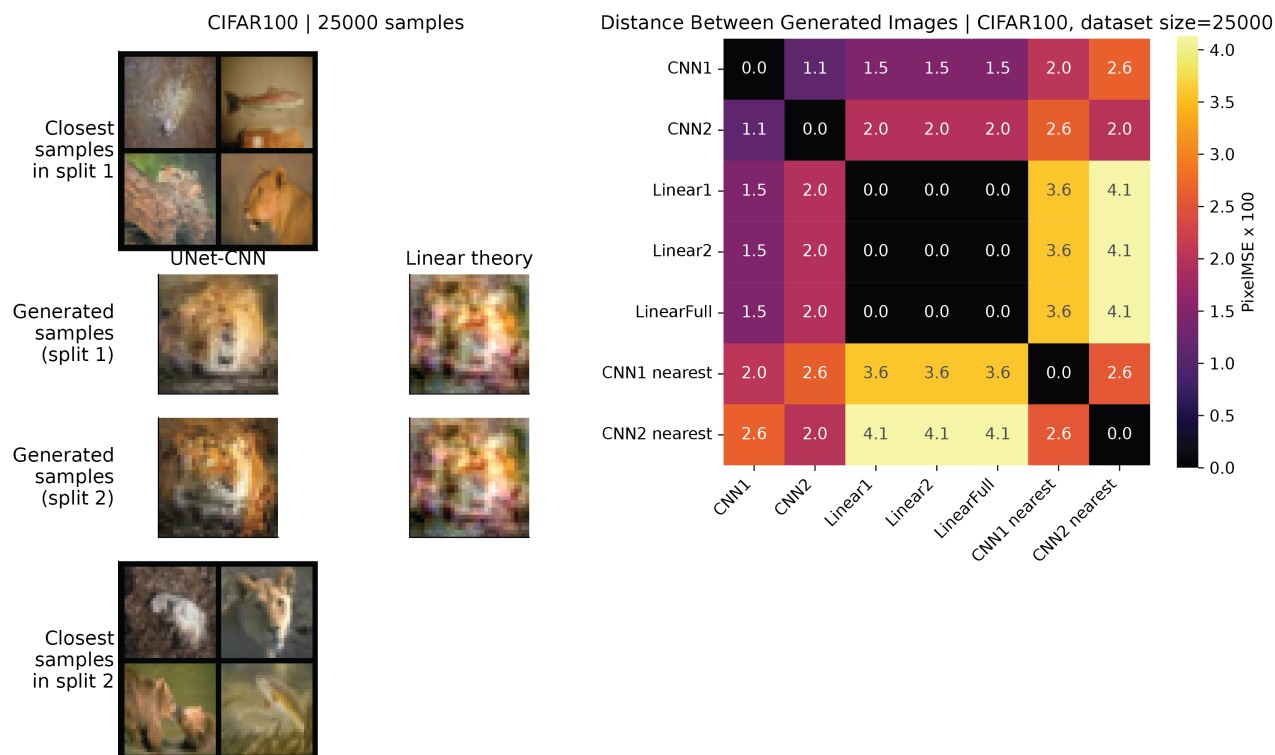

*Figure 8.* **Motivating observation and the linear theory for CIFAR100 dataset**. Similar format to Fig. 1. **Left.** Generated samples from DNN and linear theory from initial noise seed 2. **Right.** Paired image distance MSE averaging from 1000 initial noises.

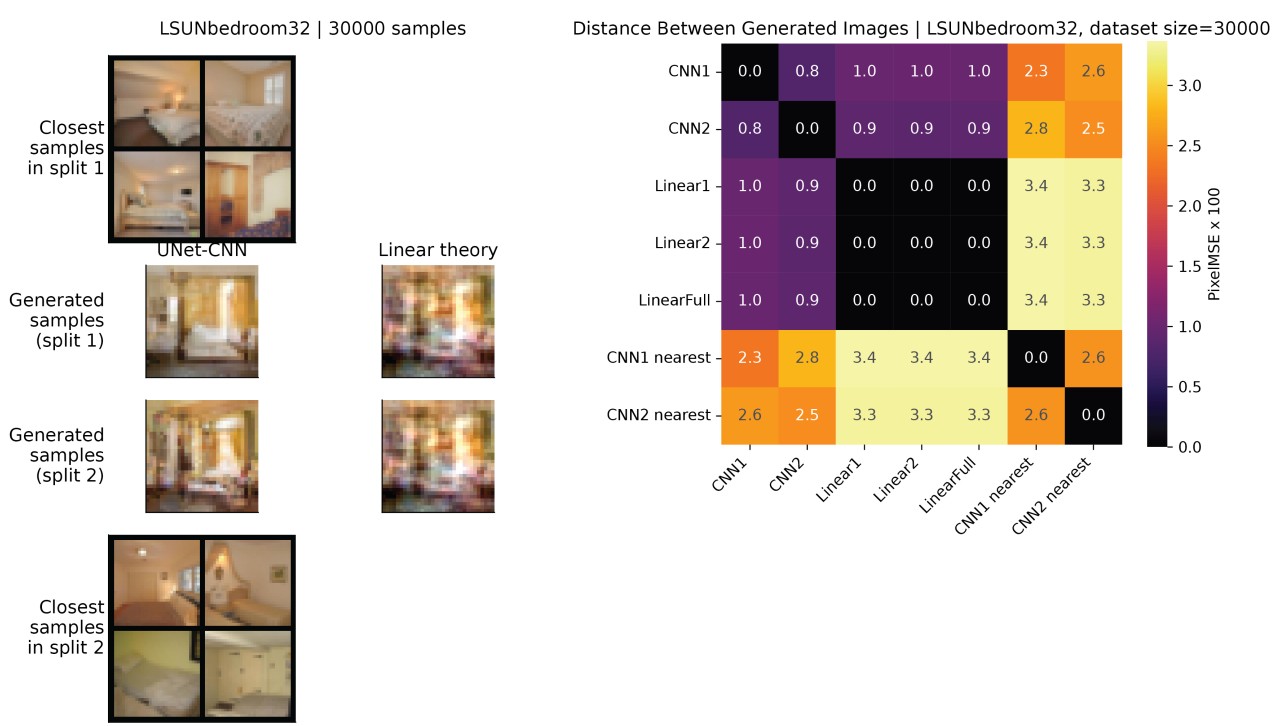

*Figure 9.* **Motivating observation and the linear theory for LSUN bedroom dataset (32 pixel)**. Similar format to Fig. 1. **Left.** Generated samples from DNN and linear theory from initial noise seed 2. **Right.** Paired image distance MSE averaging from 1000 initial noises.

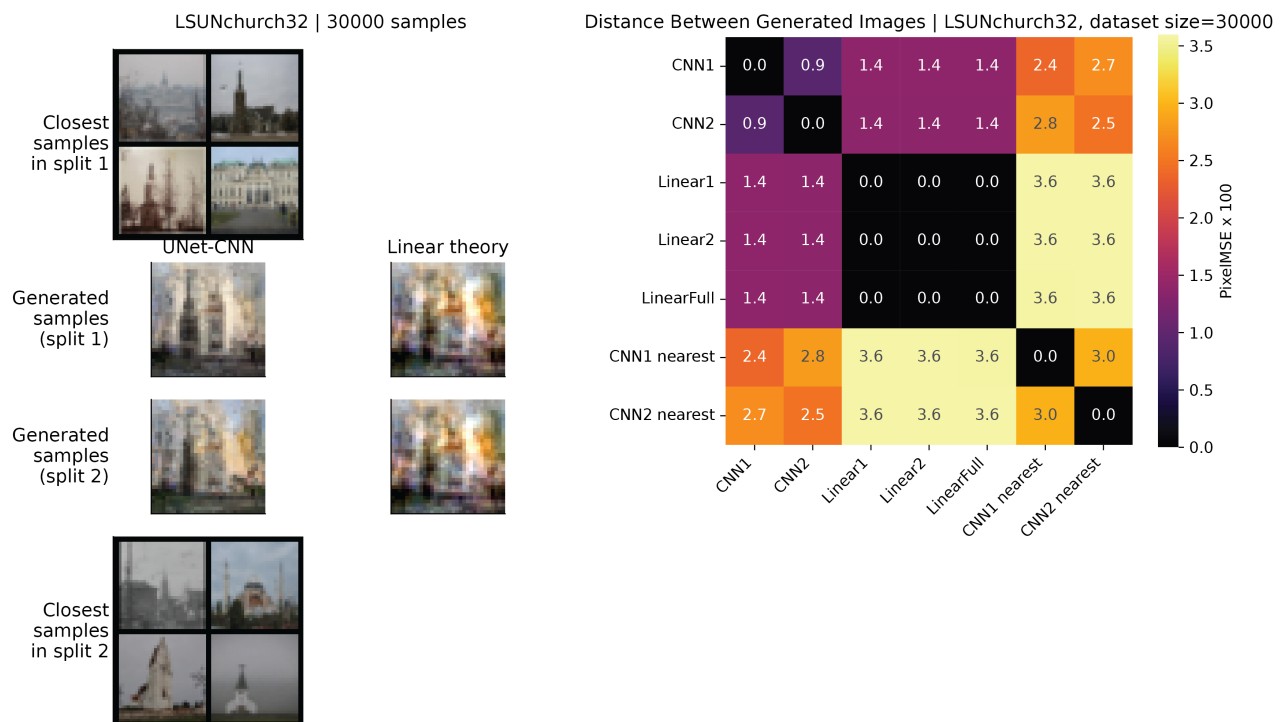

*Figure 10.* **Motivating observation and the linear theory for LSUN church dataset (32 pixel)**. Similar format to Fig. 1. **Left.** Generated samples from DNN and linear theory from initial noise seed 2. **Right.** Paired image distance MSE averaging from 1000 initial noises.

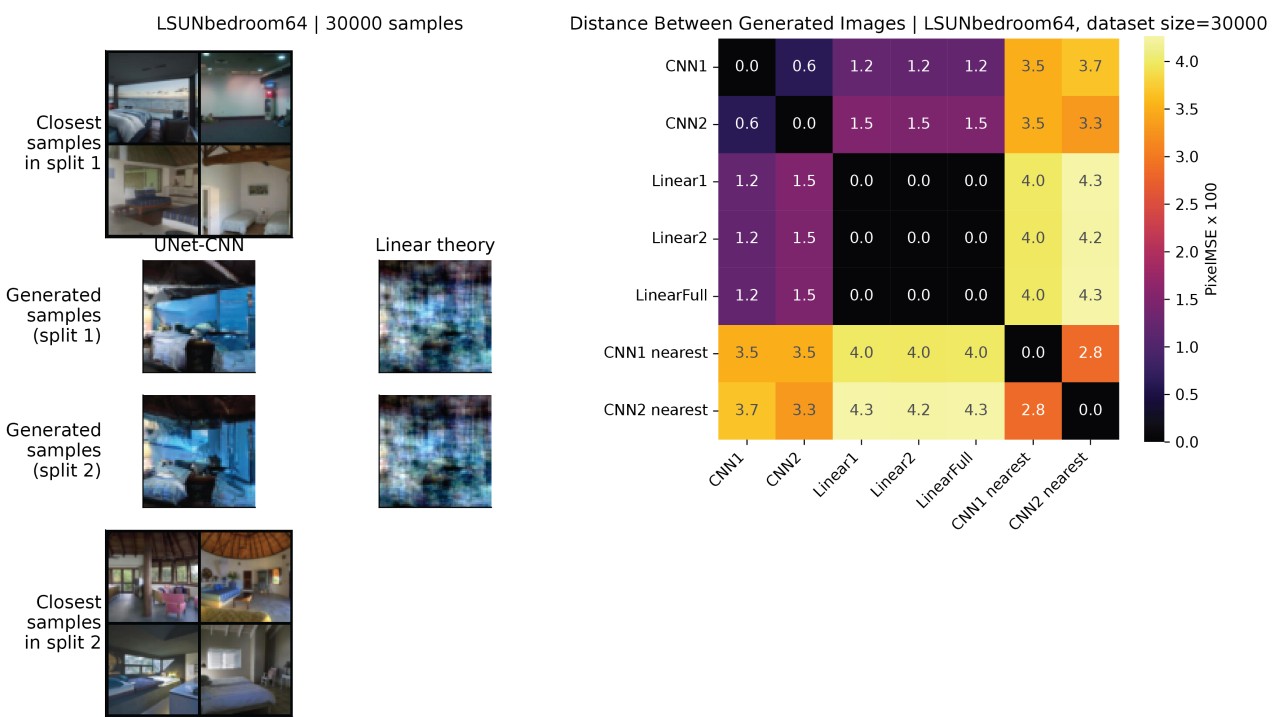

*Figure 11.* **Motivating observation and the linear theory for LSUN bedroom dataset (64 pixel)**. Similar format to Fig. 1. **Left.** Generated samples from DNN and linear theory from initial noise seed 2. **Right.** Paired image distance MSE averaging from 1000 initial noises.

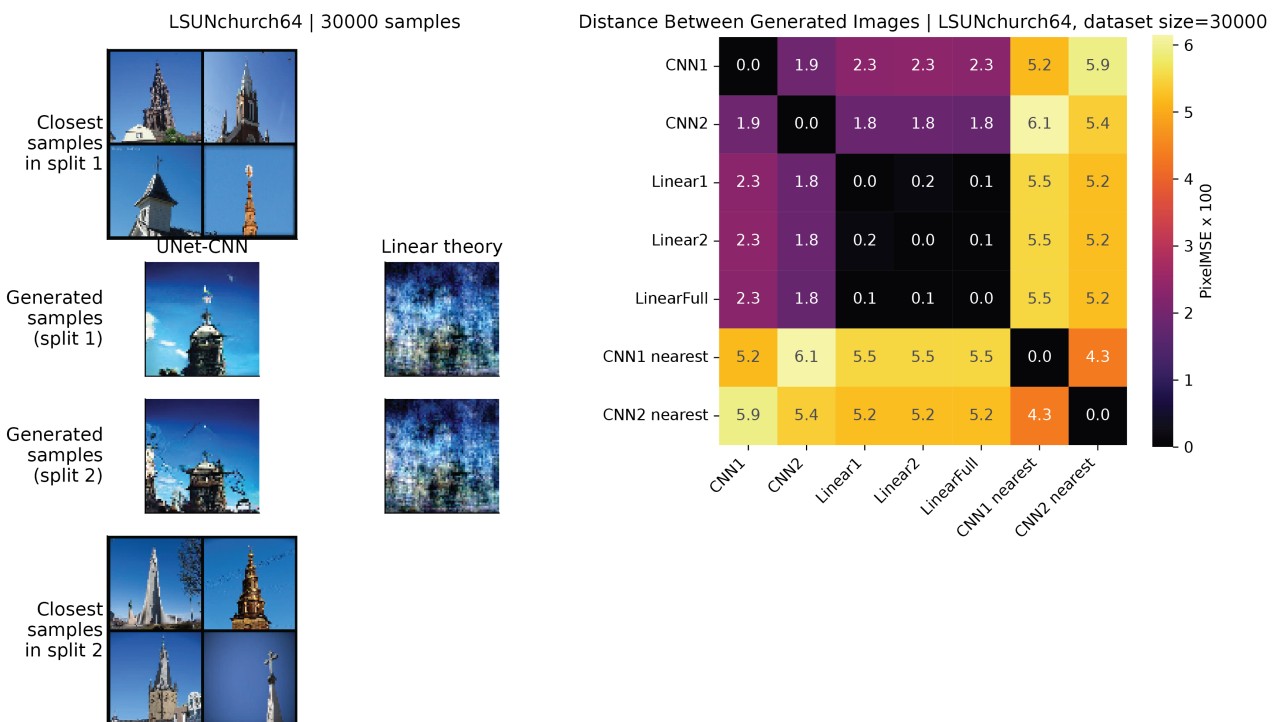

*Figure 12.* **Motivating observation and the linear theory for LSUN church dataset (64 pixel).** Similar format to Fig. 1. **Left.** Generated samples from DNN and linear theory from initial noise seed 2. **Right.** Paired image distance MSE averaging from 1000 initial noises.

## B.2. Counterfactual moment manipulation: when the first two moments do not match

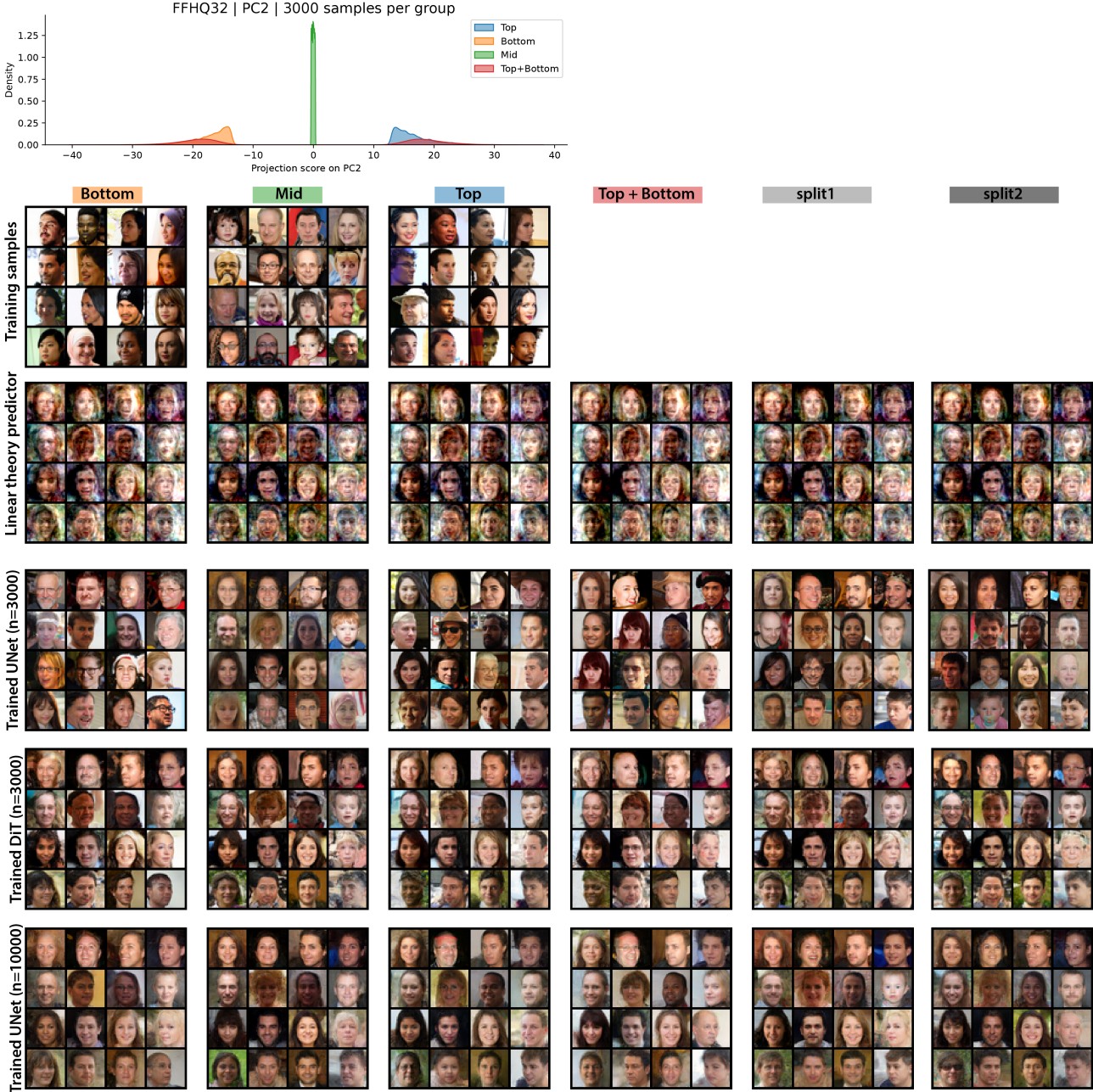

*Figure 13.* **Visual examples of counterfactual dataset splits with controlled mean and covariance. Top**: Kernel density estimates of projection scores along the second principal component (PC2) of the training data. Stratified subsets (`bottom`, `mid`, `top`) each contain 3,000 samples, while the combined `top+bottom` split is constructed by combining half of the data from the top and half from the bottom PC2 extremes, also totaling 3,000 samples. **Columns**: Different dataset splits, including PC2-stratified splits (`bottom`, `mid`, `top`), the combined `top+bottom` split, and two random non-overlapping control splits (`split1`, `split2`). **Rows**: Different sampling mechanisms, including representative training samples from each split; predictions from the linear theory; samples generated by CNN-UNet diffusion models trained on $N{=}3000$ samples; samples generated by DiT-based diffusion models trained on $N{=}3000$ samples; and samples generated by a CNN-UNet diffusion model trained on $N{=}10000$ samples (note that this model is trained on a larger dataset constructed using the same splitting mechanism). All generated samples use matched initial noise realizations to enable direct qualitative comparison across training splits. This counterfactual construction induces controlled differences in the mean and/or covariance along the PC2 direction, yielding visibly larger variations between statistically manipulated splits (especially between top and bottom) than between random control splits. This result is systematically quantified in Fig. 14.

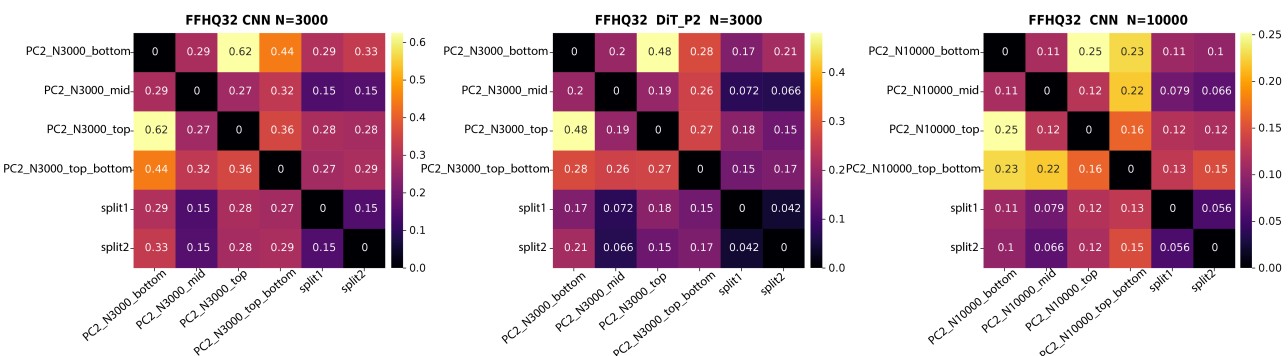

*Figure 14.* **Quantification of counterfactual dataset splits manipulations. Sample consistency between diffusion models trained on splits with mean and covariance manipulations.** Each panel shows a heatmap of the pixel-wise MSE between sample tensors generated from identical random seeds by diffusion models trained on different dataset splits (labels). Labels indicate the data splits of FFHQ32 dataset, including splits by projection along the second principal component (PC2; top, mid, bottom) and random non-overlapping splits (split1, split2). **Left**: CNN, training set size $N=3000$. **Middle**: DiT, $N=3000$. **Right**: CNN, $N=10000$. Notably, diffusion models trained on dataset splits with manipulated moments exhibit reduced sample consistency compared to models trained on two random i.i.d. splits, highlighting the importance of matching the first two moments for consistent diffusion model generation across training splits.

## B.3. Additional validation with linear denoisers

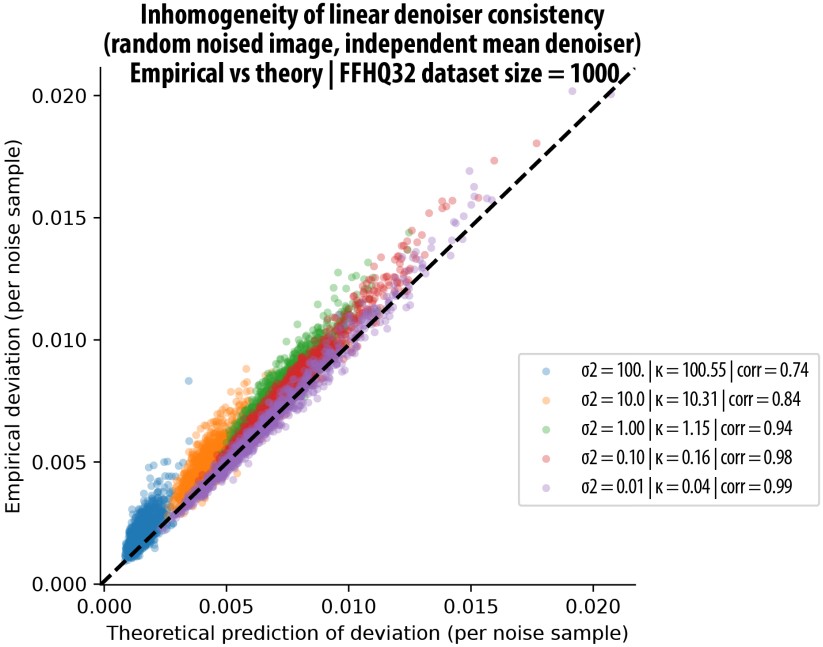

*Figure 15.* **Point by point prediction of denoiser consistency. (FFHQ32 dataset,** $n = 1000$**)** Each dot denotes one noised image sample, x-axis shows the theoretical prediction from Eq. 7, after marginalizing over $\mathbf{v}$; y-axis shows the empirical measurement of their MSE after training two linear denoiser on non-overlapping data splits. We note that, the RMT theory prediction is more precise for lower noise scales; at higher noise scales, we think the effect of different empirical means $\hat{\boldsymbol{\mu}}$ kicks in, resulting in deviation from the theory that only considers $\hat{\boldsymbol{\Sigma}}$.

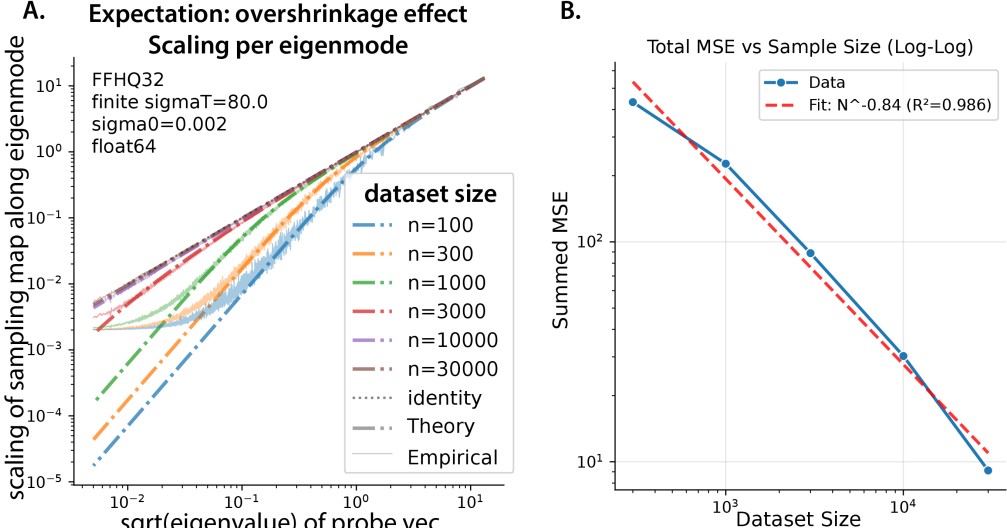

*Figure 16.* **Finite sample effect on diffusion sampling map. (extended) A.** *Overshrinkage of expectation.* The expected scaling along PC $\mathbf{u}_k^\top \hat{\boldsymbol{\Sigma}}^{1/2} \mathbf{u}_k$ of empirical sampling map compared to the ideal scaling $\sqrt{\lambda_k}$, here we used $\sigma_0 = 0.002$ for empirical matrix computation. The $\sigma_0$ is smallest noise scale that probability flow ODE integration stops, for numerical reasons. This floors the smallest scaling factor it could generate, making the mismatch with theory at the low eigen space. **B.** Overall MSE scaling with respect to dataset size, roughly scales at $1/n$ at large data, but the scaling is shallower at smaller data scale.

## B.4. Additional validation with deep networks

### B.4.1. NEAREST-NEIGHBOR DISTANCES ACROSS DATASET SIZES

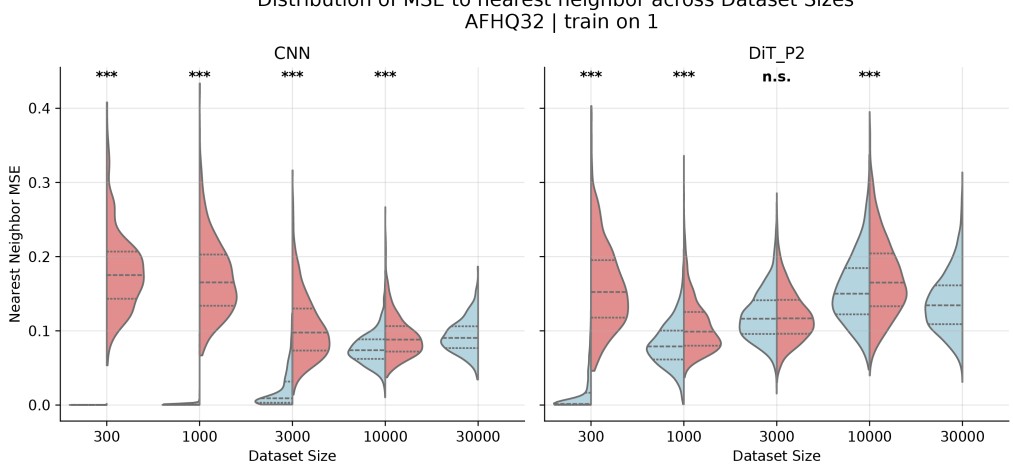

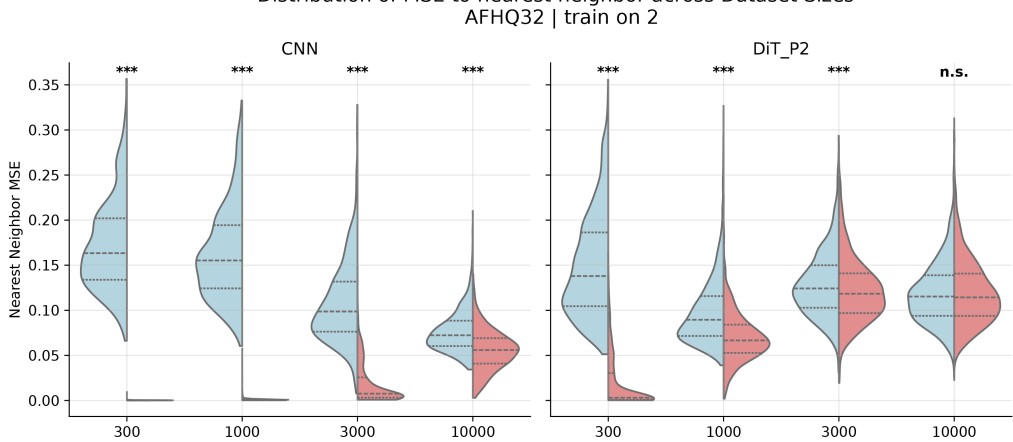

*Figure 17.* **DNN validation experiments (AFHQ32), nearest neighbor in training and control set**

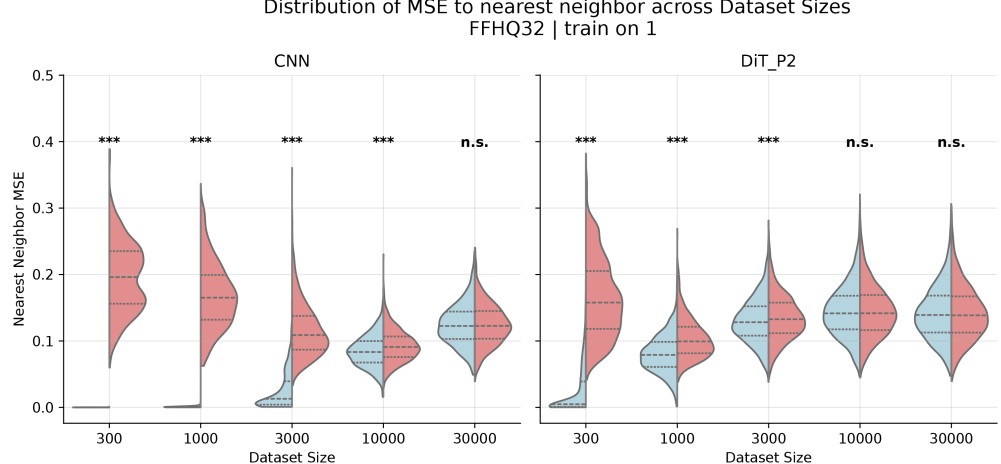

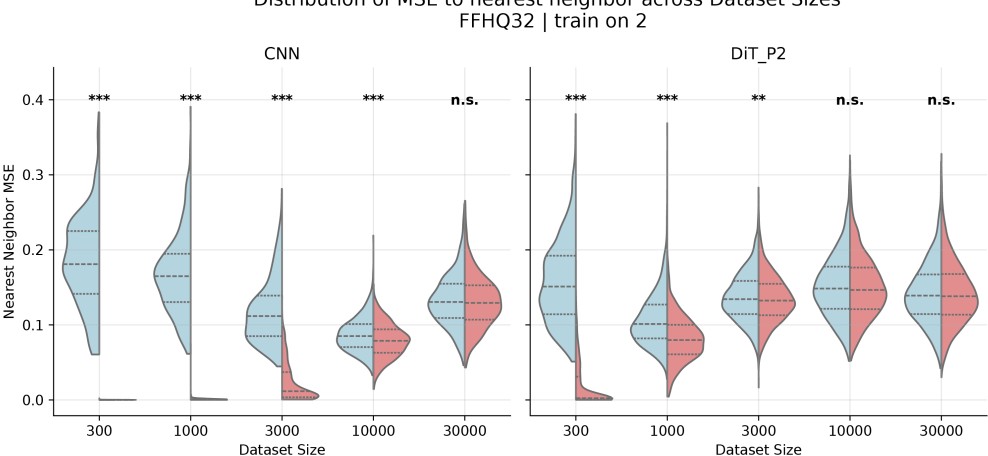

*Figure 18.* **DNN validation experiments (FFHQ32), nearest neighbor in training and control set**

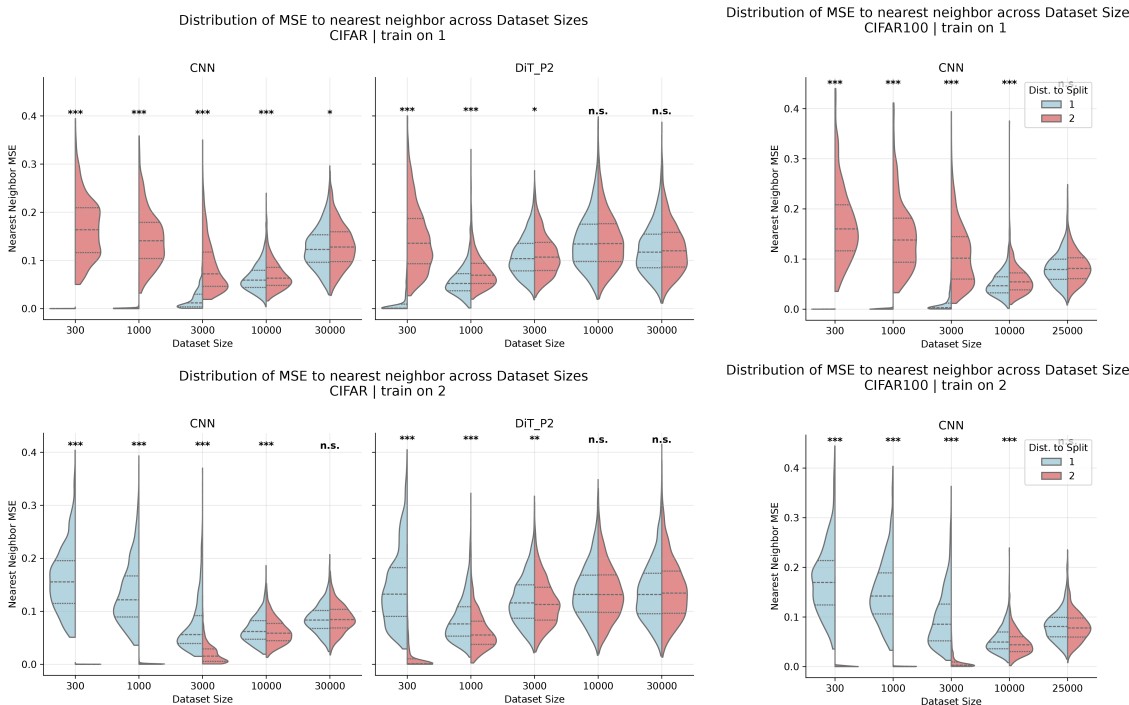

*Figure 19.* **DNN validation experiments (CIFAR10 and CIFAR100), nearest neighbor in training and control set**

Distribution of MSE to nearest neighbor across Dataset Sizes
LSUNbedroom32 | train on 1

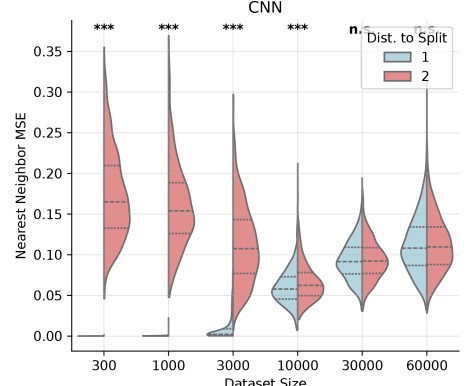

Distribution of MSE to nearest neighbor across Dataset Sizes
LSUNbedroom64 | train on 1

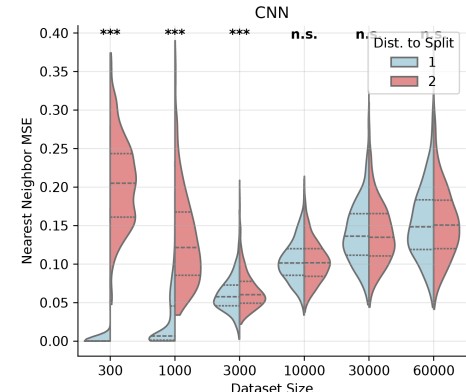

Distribution of MSE to nearest neighbor across Dataset Sizes
LSUNbedroom32 | train on 2

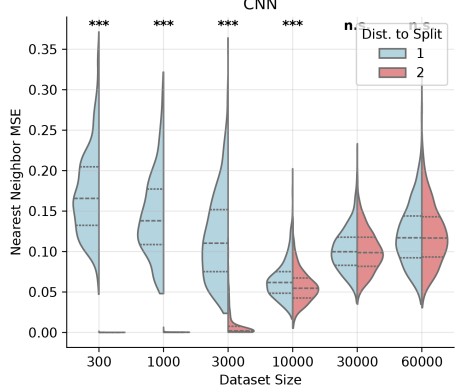

Distribution of MSE to nearest neighbor across Dataset Sizes
LSUNbedroom64 | train on 2

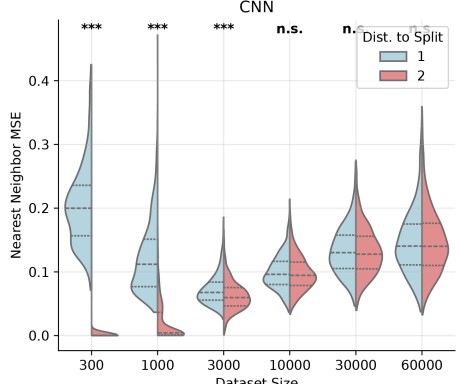

*Figure 20.* **DNN validation experiments (LSUN bedroom 32 and 64), nearest neighbor in training and control set**

Distribution of MSE to nearest neighbor across Dataset Sizes
LSUNchurch32 | train on 1

Distribution of MSE to nearest neighbor across Dataset Sizes
LSUNchurch64 | train on 1

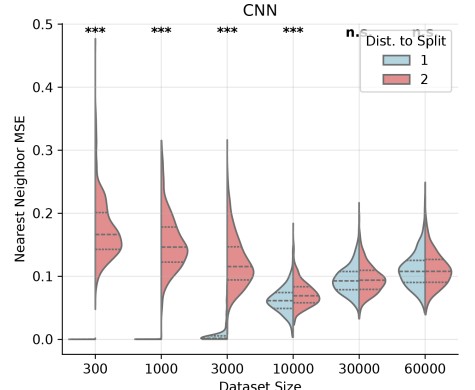 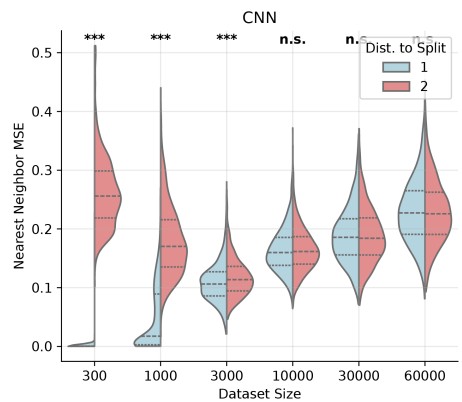

Distribution of MSE to nearest neighbor across Dataset Sizes
LSUNchurch32 | train on 2

Distribution of MSE to nearest neighbor across Dataset Sizes
LSUNchurch64 | train on 2

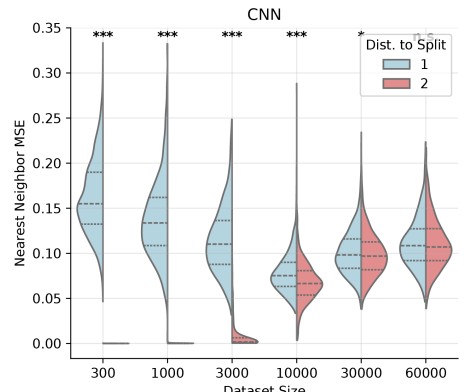 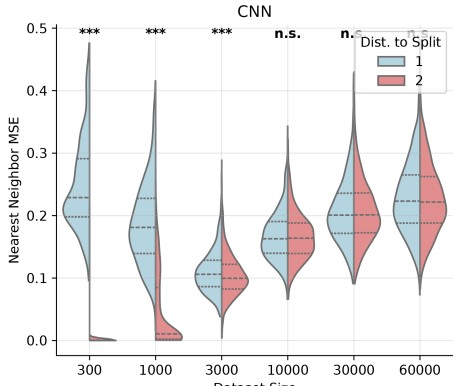

*Figure 21.* **DNN validation experiments (LSUN church 32 and 64), nearest neighbor in training and control set**

### B.4.2. CROSS-SPLIT CONSISTENCY ACROSS DATASET SIZES

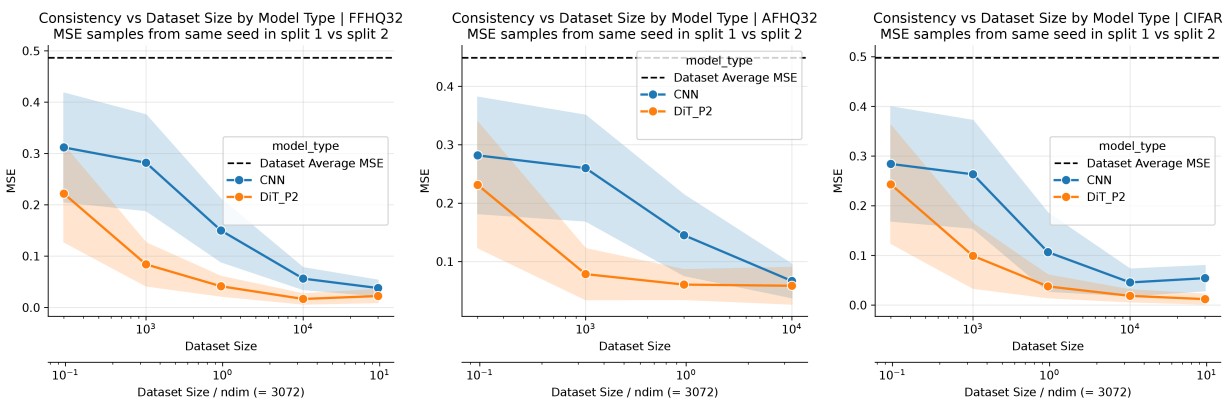

*Figure 22.* **Scaling of DNN generation consistency with dataset size (CNN and DiT).**

### B.4.3. DNN SAMPLES APPROACH THE GAUSSIAN PREDICTOR AT LARGER DATASET SIZES

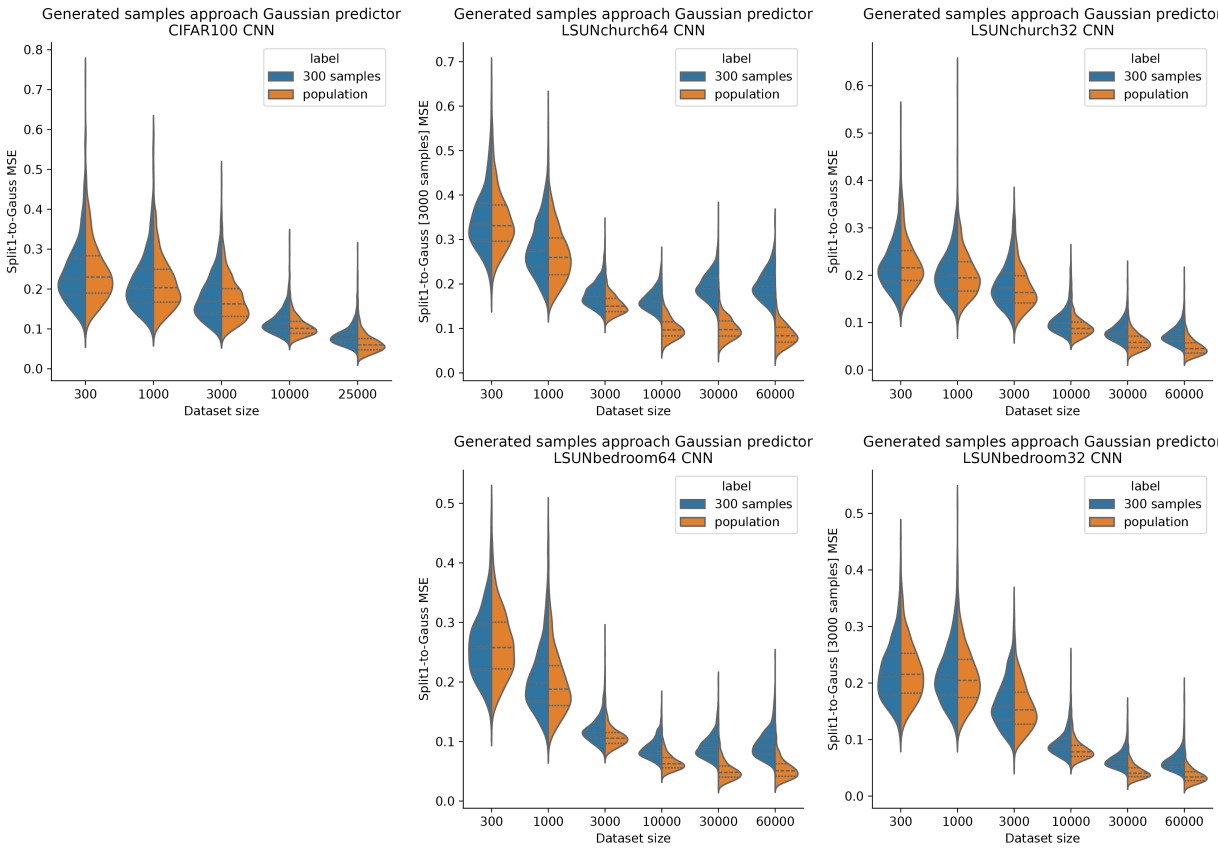

*Figure 23.* **DNN-generated samples approach the linear theory predictor (with finite-sample or population covariance).** With increasing dataset size $n$, the generated samples from DNNs (trained on split 1) with a fixed noise seed gradually approach the linear theory predictor using the same initial seed. The left violin plot shows the MSE from the linear predictor using empirical mean and covariance $\hat{\boldsymbol{\mu}}, \hat{\boldsymbol{\Sigma}}$ computed from only 300 samples; the right violin plot shows the MSE from the linear predictor using the population mean and covariance (whole dataset). The results are consistent across datasets.

### B.4.4. SPECTRAL STRUCTURE OF GENERATION VARIANCE AND DEVIATION

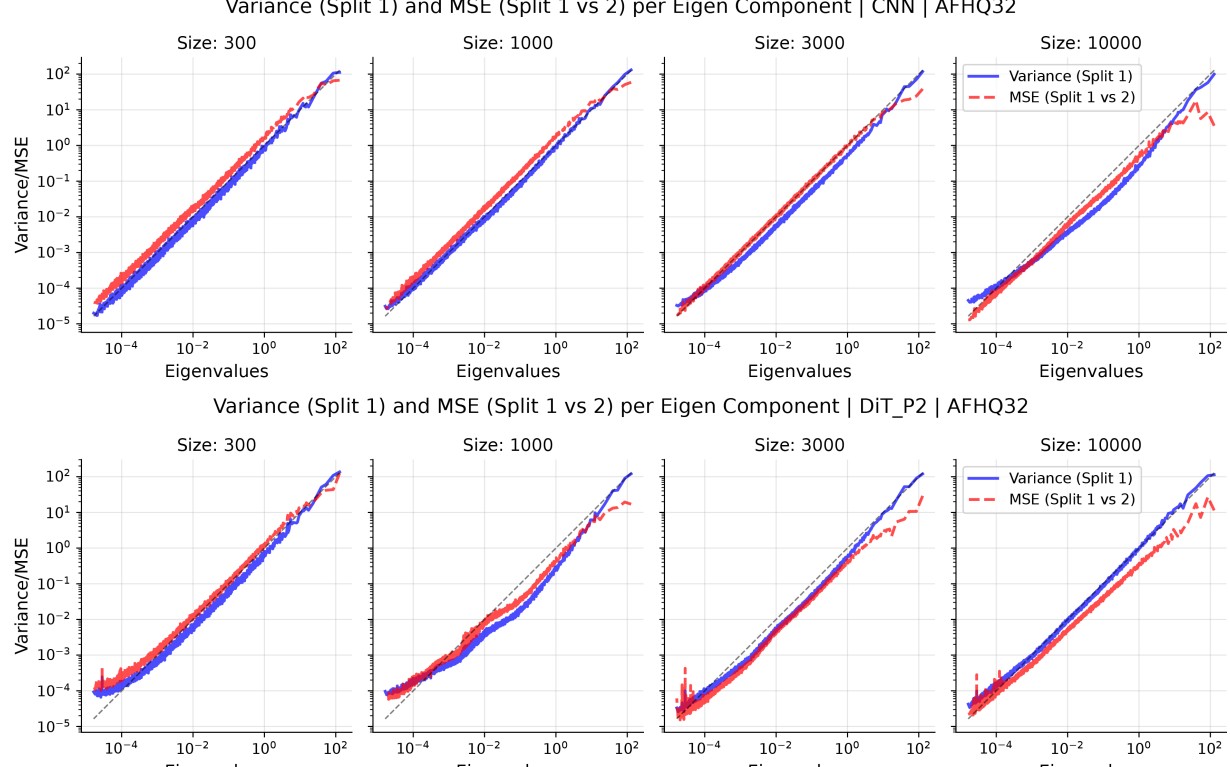

*Figure 24.* **DNN validation experiments, Anisotropy and overshrinking (AFHQ32)**

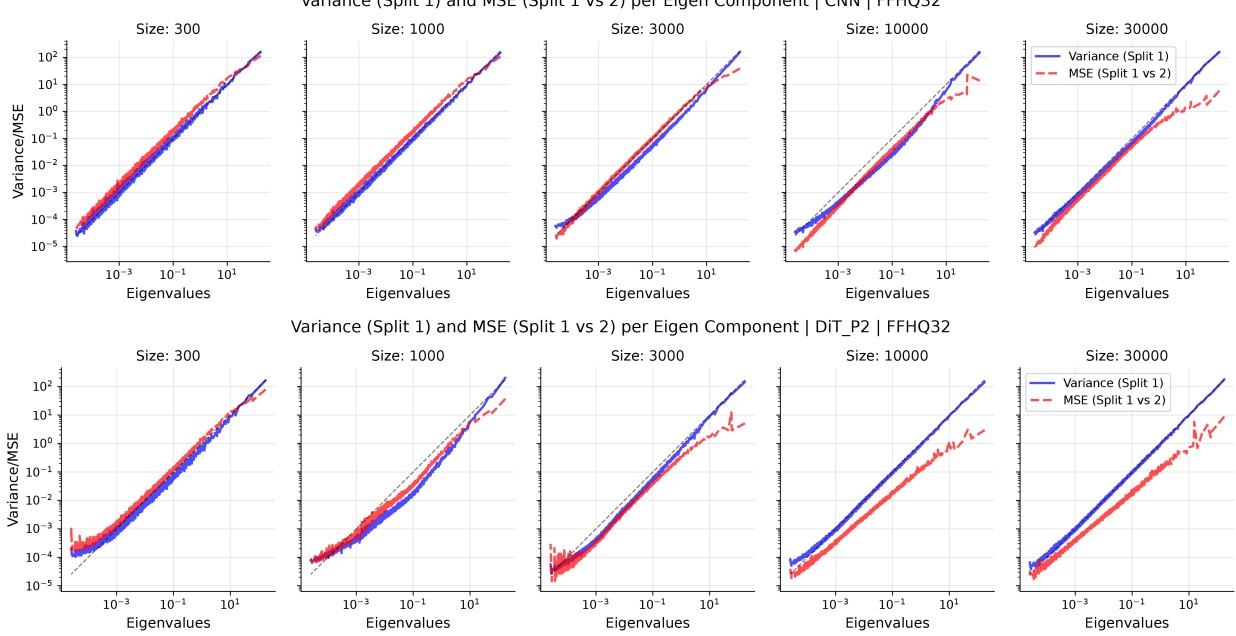

*Figure 25.* **DNN validation experiments, Anisotropy and overshrinking (FFHQ32)**

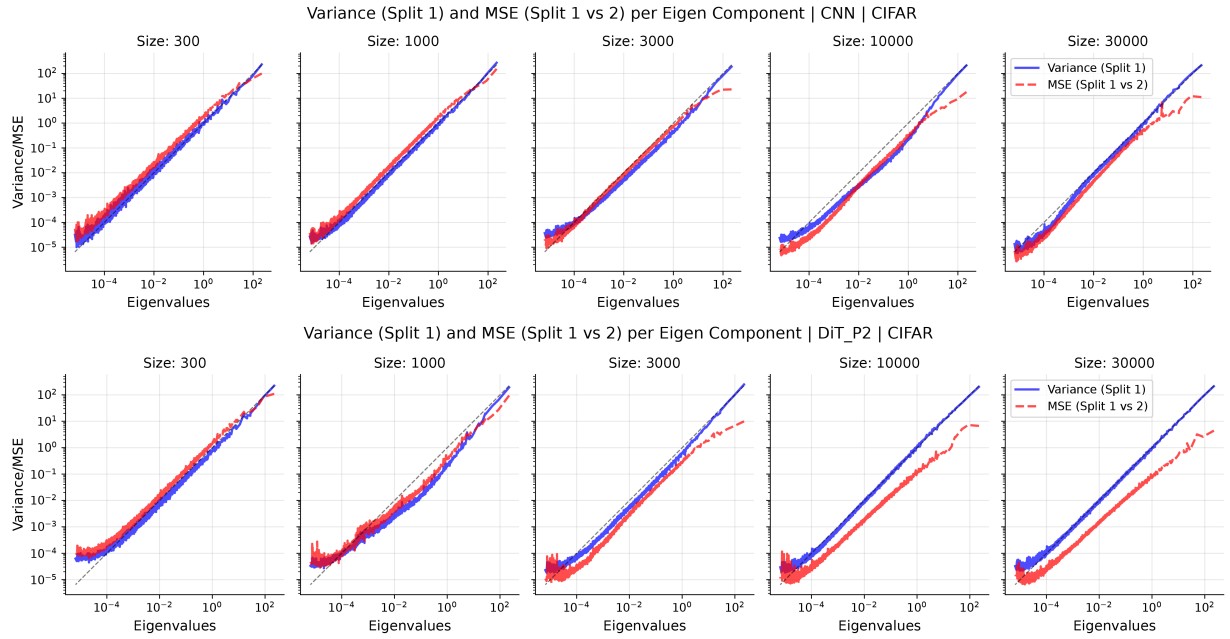

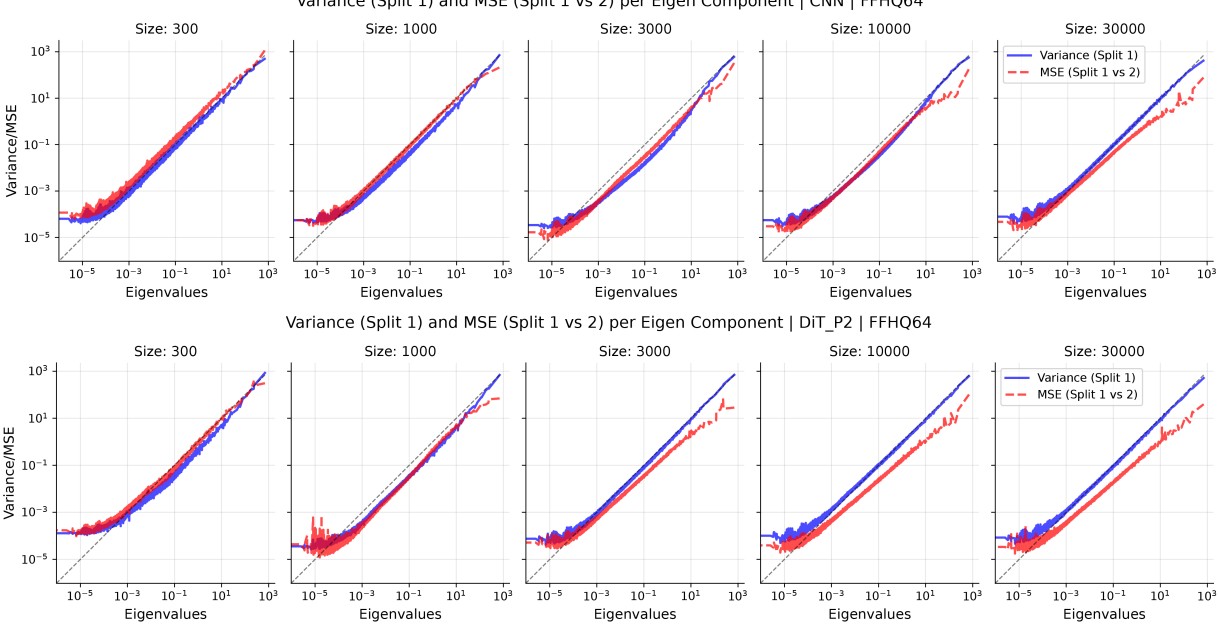

*Figure 26.* **DNN validation experiments, Anisotropy and overshrinking (CIFAR10)**

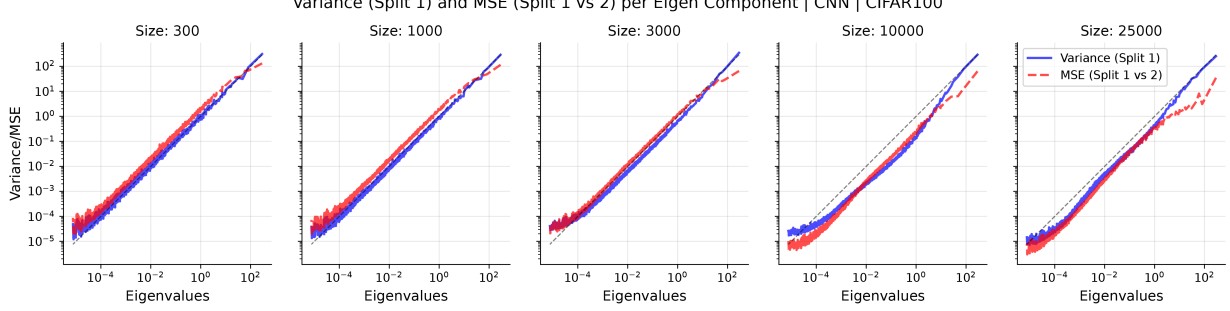

*Figure 27.* **DNN validation experiments, Anisotropy and overshrinking (FFHQ64)**

*Figure 28.* **DNN validation experiments, Anisotropy and overshrinking (CIFAR100)**

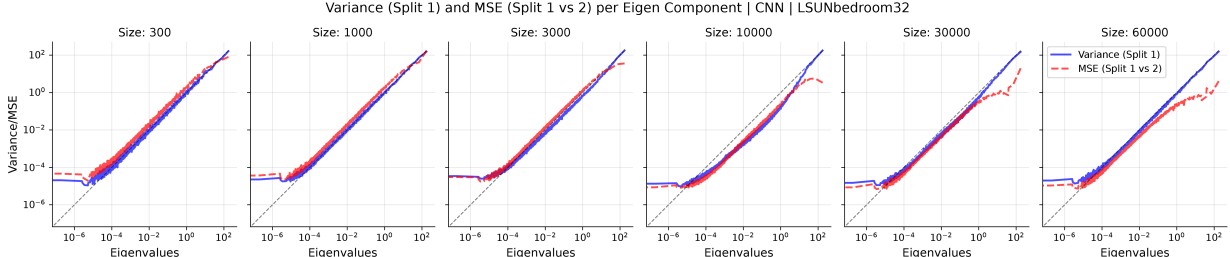

*Figure 29.* **DNN validation experiments, Anisotropy and overshrinking (LSUN bedroom 32)**

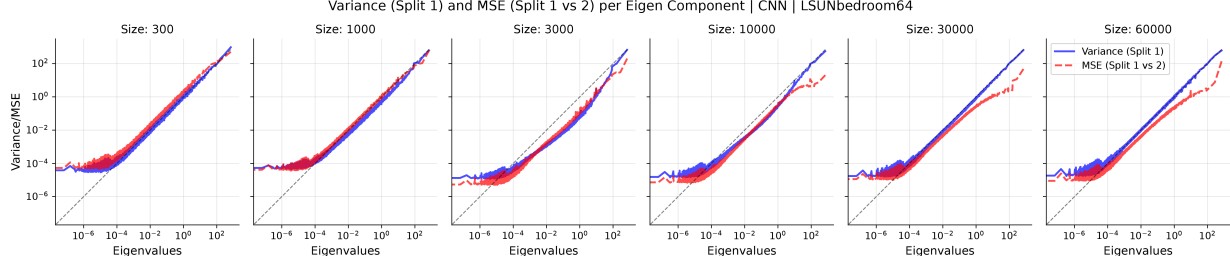

*Figure 30.* **DNN validation experiments, Anisotropy and overshrinking (LSUN bedroom 64)**

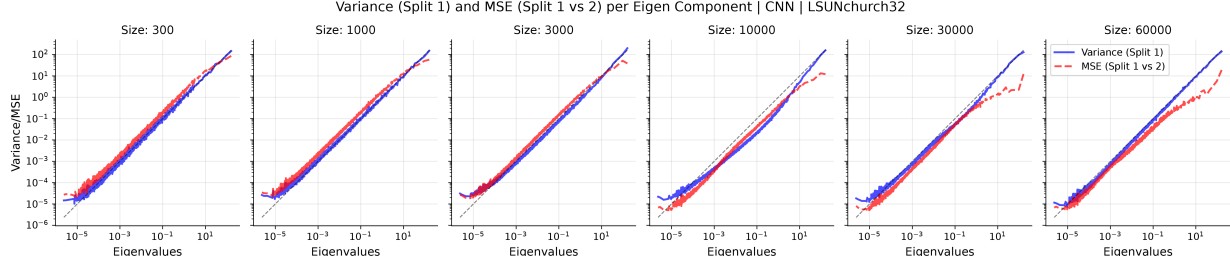

*Figure 31.* **DNN validation experiments, Anisotropy and overshrinking (LSUN church 32)**

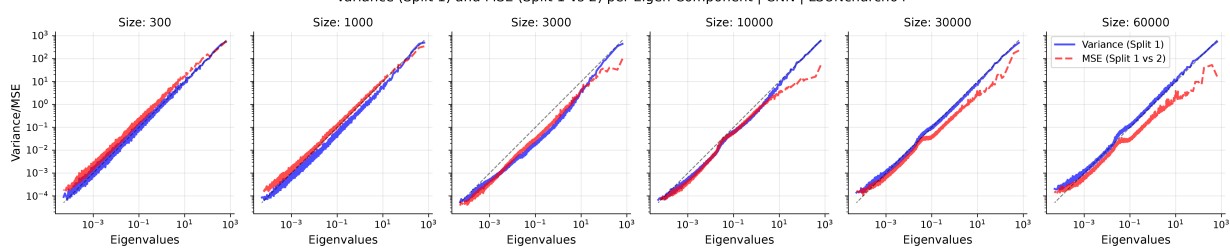

*Figure 32.* **DNN validation experiments, Anisotropy and overshrinking (LSUN church 64)**

### B.4.5. SPATIAL INHOMOGENEITY OF CROSS-SPLIT DISAGREEMENT

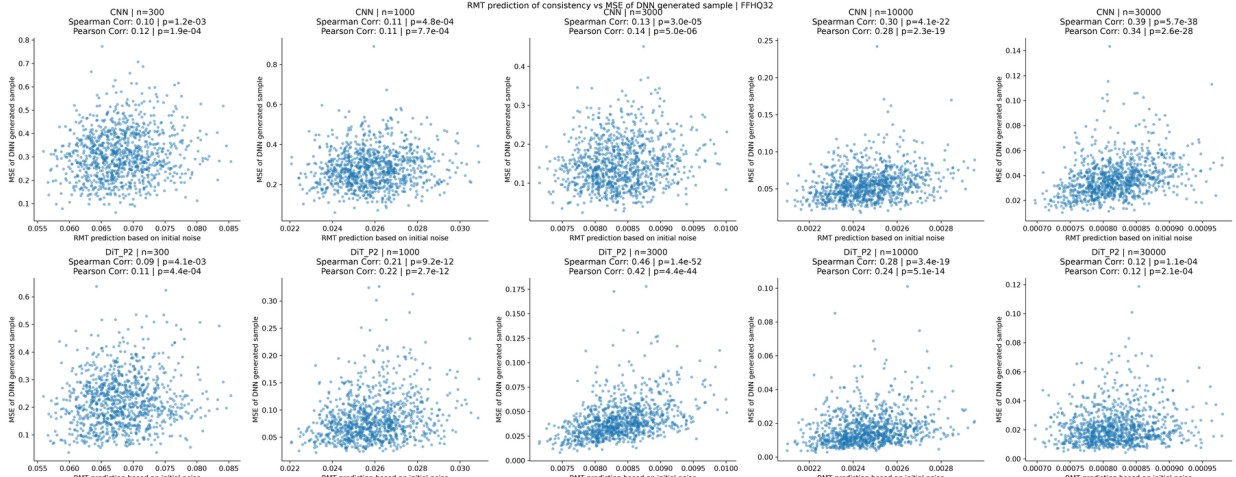

*Figure 33.* **DNN validation experiments, RMT predicting inhomogeneity (FFHQ32)**

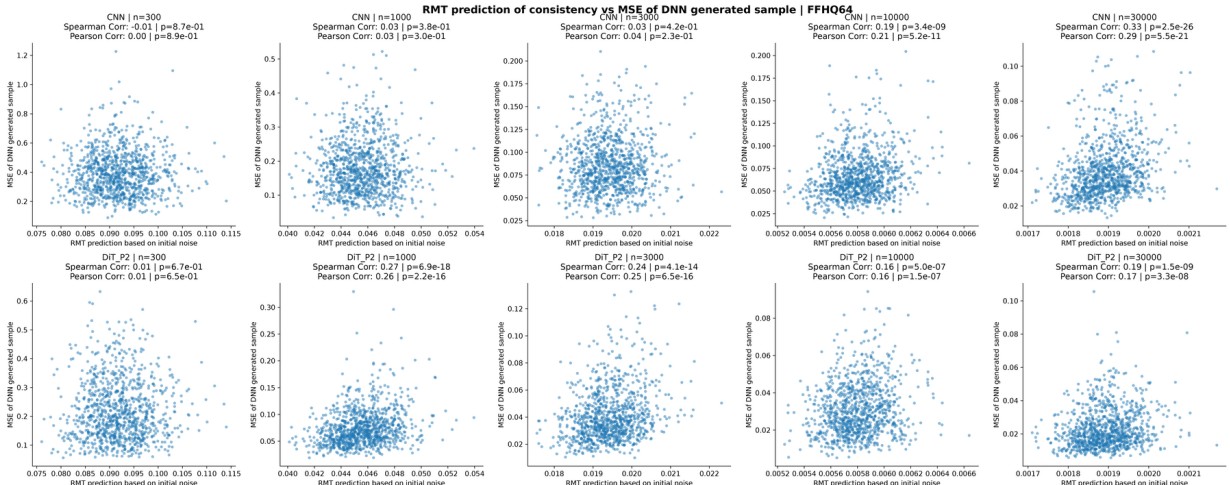

*Figure 34.* **DNN validation experiments, RMT predicting inhomogeneity (FFHQ64)**

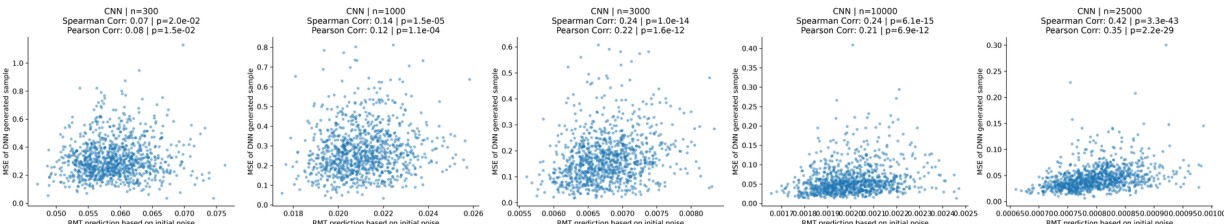

*Figure 35.* **DNN validation experiments, RMT predicting inhomogeneity (CIFAR100)**

B.4.6. ANISOTROPY OF THE INITIAL NOISE SPACE

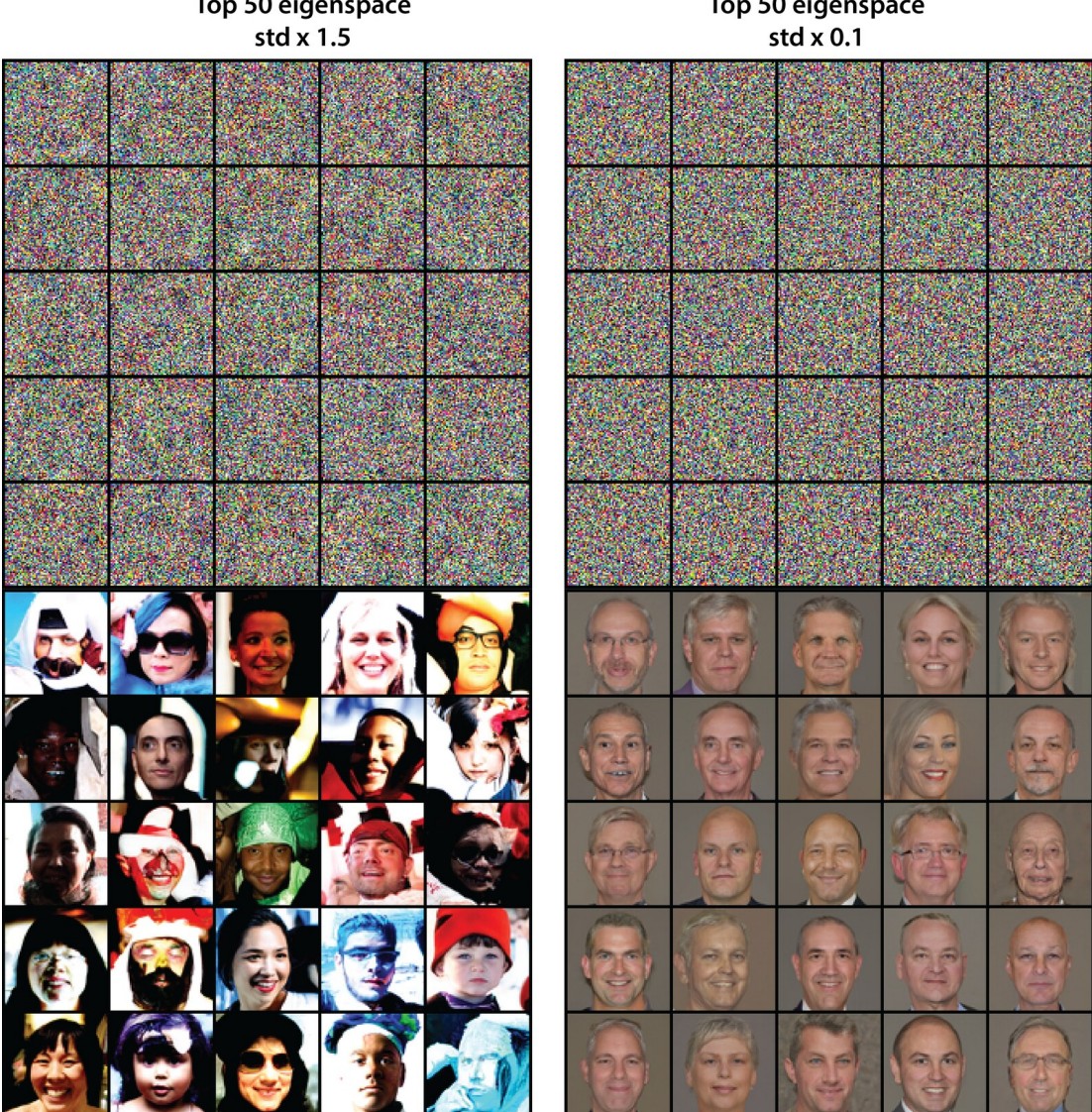

*Figure 36.* **Anisotropic structure of the initial noise space (CNN-UNet FFHQ64). Left:** Amplifying the initial-noise amplitude in the top-50 eigenspace. **Right:** Decreasing the initial-noise amplitude in the top-50 eigenspace. **Top**: initial noise; **Bottom**: generated samples (same noise seed). Increasing noise in the dominant eigendirections introduces more visual artifacts by amplifying the top eigen-structure of the generative map (Eq. 3). Conversely, reducing noise in these dimensions yields a cleanly segmented face against a gray background.

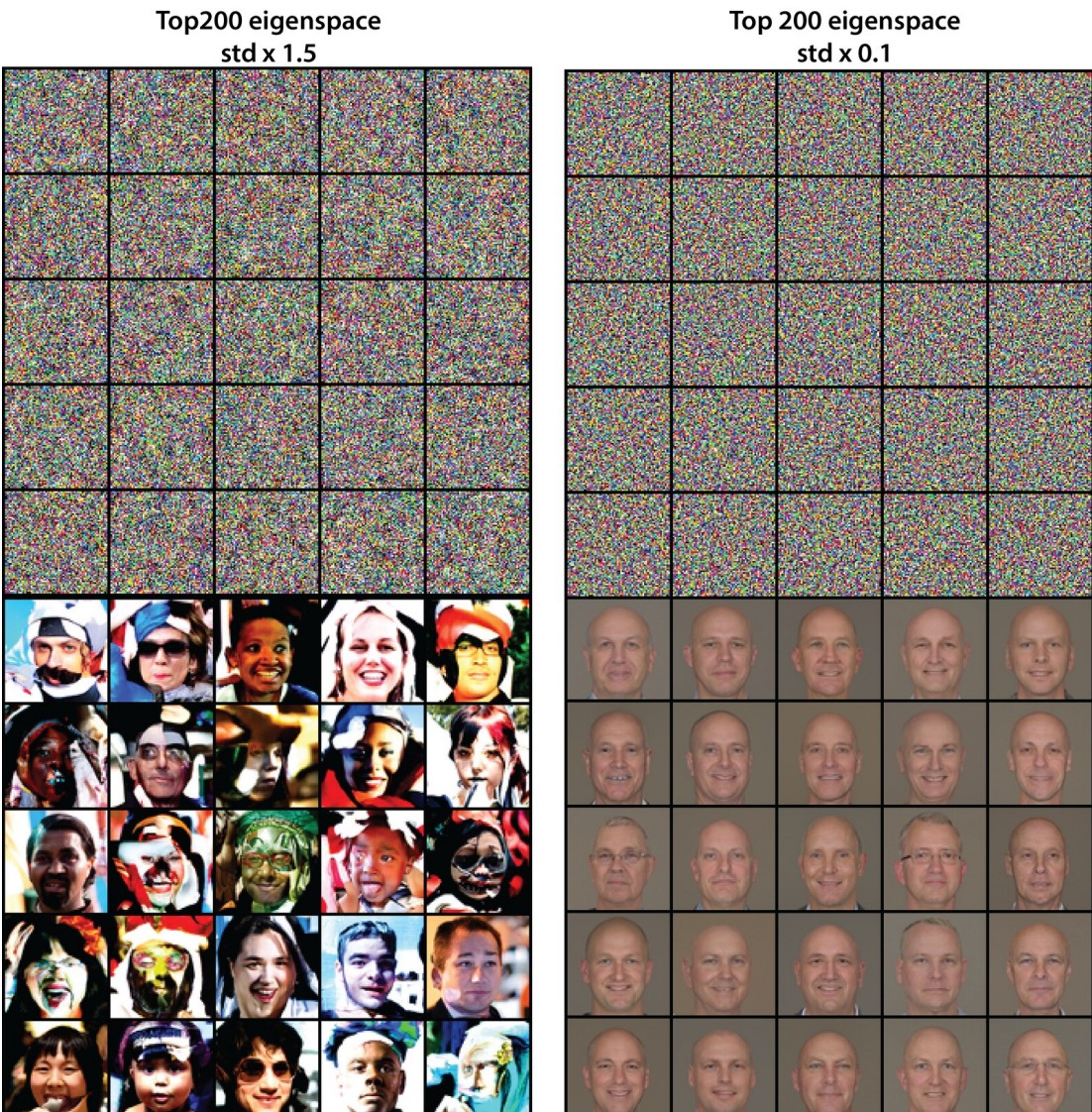

*Figure 37.* **Anisotropic structure of the initial noise space (CNN-UNet FFHQ64). Left:** Amplifying the initial-noise amplitude in the top-200 eigenspace. **Right:** Decreasing the initial-noise amplitude in the top-200 eigenspace. **Top**: initial noise; **Bottom**: generated samples (same noise seed). Similar to the top-50 case: stronger noise in leading eigendirections amplifies dominant structure, producing visible artifacts. Suppressing noise yields an even more homogeneous and simplified face, reflecting the removal of additional variation modes.

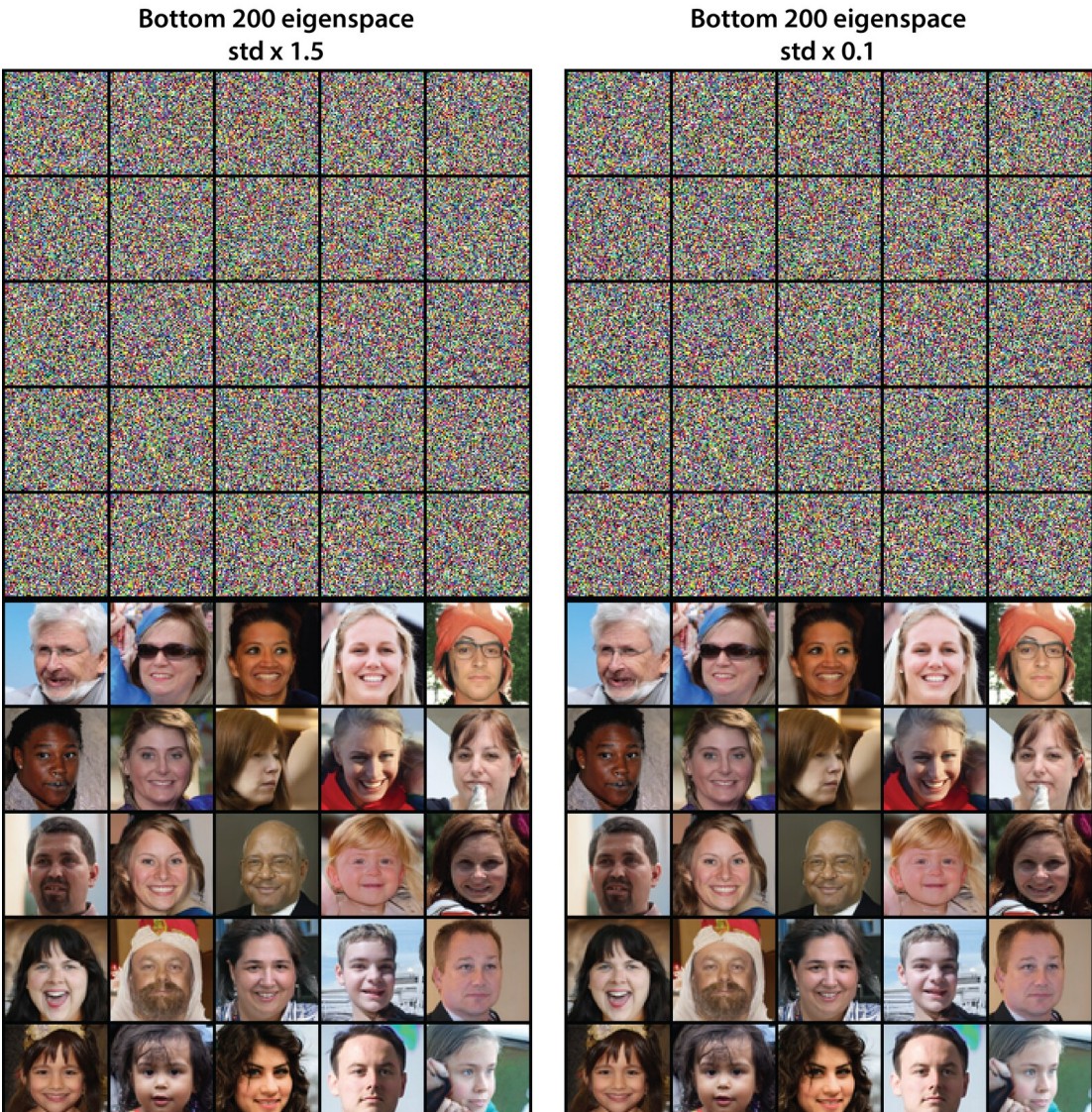

*Figure 38.* **Anisotropic structure of the initial noise space (CNN-UNet FFHQ64). Left:** Amplifying the initial-noise amplitude in the bottom-200 eigenspace. **Right:** Decreasing the initial-noise amplitude in the bottom-200 eigenspace. **Top**: initial noise; **Bottom**: generated samples (same noise seed). In contrast to perturbations along top eigendirections, manipulating the least-significant eigenspaces produces minimal perceptual impact on the generated image.

# C. Proofs and Derivations

## C.1. Deterministic equivalence relations

Here we collect the one-point and two-point deterministic equivalent relationships from (Atanasov et al., 2026b;a; Bach, 2024), rewritten in a unified notation.

**Setup**   Using similar notation as Bach (2024), we consider data matrix $X \in \mathbb{R}^{n \times d}$, where each row is an i.i.d. sample $\mathbf{x}_i$. The population covariance of these samples is denoted as $\mathbf{\Sigma}$. The key object of analysis is their empirical covariance

$$\hat{\mathbf{\Sigma}} = \frac{1}{n} X^\top X$$

We use $\gamma = d/n$ to denote the aspect ratio of $X$, i.e. the dimensionality of data over the number of data points. We use $\mathrm{Tr}$ to denote the normal trace of a matrix, and $\mathrm{tr}$ to denote the normalized trace, i.e. $\mathrm{tr}[M] = \frac{1}{d}\mathrm{Tr}[M]$ for a $d \times d$ matrix $M$.

**Self-consistency equation for the renormalized noise scale**   The spectral properties of a matrix are determined by the Stieltjes transform. We consider the Stieltjes transform of the kernel matrix $\frac{1}{n}XX^\top$, defined as $\hat{\varphi}(z) := \mathrm{Tr}[(XX^\top - nzI)^{-1}]$. In the large-matrix limit, the limiting variable $\varphi(z)$ satisfies the following self-consistent equation,

$$\frac{1}{\varphi(z)} + z = \gamma \int_0^\infty \frac{s d\mu(s)}{1 + s\varphi(z)} \tag{11}$$

where $\mu(s)$ is the limiting spectral measure of the population covariance $\mathbf{\Sigma}$. This follows from the leave-one-out arguments in the Appendix of Bach (2024), as well as Bai et al. (2010); Ledoit & Péché (2011).

This can be translated to the self-consistent equation of the renormalized ridge variable $\kappa(z) := \frac{1}{\varphi(-z)}$, which is used throughout the paper,

$$\frac{1}{\varphi(-z)} - z = \gamma \int_0^\infty \frac{s d\mu(s)}{1 + s\varphi(-z)}$$

$$\kappa(z) - z = \gamma \int_0^\infty \frac{s d\mu(s)}{1 + s\frac{1}{\kappa(z)}}$$

$$\kappa(z) - z = \gamma \kappa(z) \int_0^\infty \frac{s d\mu(s)}{\kappa(z) + s}$$

$$z = \kappa(z) \left[ 1 - \gamma \int_0^\infty \frac{s d\mu(s)}{\kappa(z) + s} \right]$$

Practically, when solving such equations, given a finite size population covariance matrix, the integral over the spectral measure can be represented as normalized trace, leading to the Silverstein equation (Eq. 4).

$$\kappa(\lambda) - \lambda = \gamma \kappa(\lambda) \, \mathrm{tr}[\mathbf{\Sigma}(\mathbf{\Sigma} + \kappa(\lambda)I)^{-1}] \tag{Silverstein}$$

The self-consistent solution to this equation obeys $\kappa(\lambda) \geq \lambda$ for all $\lambda \geq 0$, as can be seen from the fact that the right-hand-side is non-negative.

**Degrees-of-freedom functions**   We define the degrees-of-freedom functions with unnormalized trace, following the convention in Bach (2024), unlike Atanasov et al. (2026a), which used normalized trace.

$$\mathrm{df}_1(\lambda) := \mathrm{Tr}[\mathbf{\Sigma}(\mathbf{\Sigma} + \lambda I)^{-1}] \tag{12}$$

$$\mathrm{df}_2(\lambda) := \mathrm{Tr}[\mathbf{\Sigma}^2(\mathbf{\Sigma} + \lambda I)^{-2}]. \tag{13}$$

Note that

$$\mathrm{df}_2(\lambda) - \mathrm{df}_1(\lambda) = \mathrm{Tr}[\mathbf{\Sigma}^2(\mathbf{\Sigma} + \lambda I)^{-2}] - \mathrm{Tr}[\mathbf{\Sigma}(\mathbf{\Sigma} + \lambda I)^{-1}]$$

$$= \mathrm{Tr}[(\mathbf{\Sigma}(\mathbf{\Sigma} + \lambda I)^{-1} - I)\mathbf{\Sigma}(\mathbf{\Sigma} + \lambda I)^{-1}]$$

$$= -\lambda \, \mathrm{Tr}[\mathbf{\Sigma}(\mathbf{\Sigma} + \lambda I)^{-2}]$$

$$\leq 0.$$

Note that both $\mathrm{df}_2(\lambda), \mathrm{df}_1(\lambda)$ are smaller than the number of non-zero eigenvalues of $\mathbf{\Sigma}$, i.e. $\mathrm{rank}(\mathbf{\Sigma})$. Thus, we have the chain of inequalities

$$0 \leq \mathrm{df}_2(\lambda) \leq \mathrm{df}_1(\lambda) \leq \mathrm{rank}(\mathbf{\Sigma}) \leq d.$$

We will also use the mixed degrees-of-freedom function

$$\mathrm{df}_2(\lambda, \lambda') := \mathrm{Tr}[\mathbf{\Sigma}^2(\mathbf{\Sigma} + \lambda I)^{-1}(\mathbf{\Sigma} + \lambda' I)^{-1}]. \tag{14}$$

Clearly, $\mathrm{df}_2(\lambda, \lambda') \geq 0$. By the Cauchy-Schwarz inequality,

$$\mathrm{df}_2(\lambda, \lambda') = \mathrm{Tr}[\mathbf{\Sigma}^2(\mathbf{\Sigma} + \lambda I)^{-1}(\mathbf{\Sigma} + \lambda' I)^{-1}] \leq \sqrt{\mathrm{Tr}[\mathbf{\Sigma}^2(\mathbf{\Sigma} + \lambda I)^{-2}]\,\mathrm{Tr}[\mathbf{\Sigma}^2(\mathbf{\Sigma} + \lambda' I)^{-2}]} = \sqrt{\mathrm{df}_2(\lambda)\,\mathrm{df}_2(\lambda')}. \tag{15}$$

Now, by re-writing the Silverstein equation, we have that

$$\mathrm{df}_1(\kappa) = \left(1 - \frac{\lambda}{\kappa}\right) n < n, \tag{16}$$

where the inequality is strict if both $\lambda$ and $\kappa$ are strictly positive. This leads to an additional upper bound

$$\mathrm{df}_2(\kappa, \kappa') < n. \tag{17}$$

**Basic deterministic equivalences**   Following Proposition 1 of Bach (2024), we use the shorthand $\kappa(z) := 1/\varphi(-z)$ to express the deterministic equivalences in the more convenient forms below. In what follows, $A$ and $B$ are test matrices of bounded spectral norm. For the resolvent, we have:

$$\mathrm{Tr}\big[A\,(\hat{\mathbf{\Sigma}} + \lambda I)^{-1}\big] \asymp \frac{\kappa(\lambda)}{\lambda}\,\mathrm{Tr}\big[A(\mathbf{\Sigma} + \kappa(\lambda)I)^{-1}\big] \tag{18}$$

and

$$
\begin{aligned}
\mathrm{Tr}\big[A(\hat{\mathbf{\Sigma}} + \lambda I)^{-1}B(\hat{\mathbf{\Sigma}} + \lambda I)^{-1}\big] \\
\asymp \frac{\kappa(\lambda)^2}{\lambda^2}\,\mathrm{Tr}\big[A\,(\mathbf{\Sigma} + \kappa(\lambda)I)^{-1}\,B\,(\mathbf{\Sigma} + \kappa(\lambda)I)^{-1}\big] \\
+ \frac{\kappa(\lambda)^2}{\lambda^2}\frac{1}{n - \mathrm{df}_2(\kappa(\lambda))}\,\mathrm{Tr}\big[A\,(\mathbf{\Sigma} + \kappa(\lambda)I)^{-2}\mathbf{\Sigma}\big]\,\mathrm{Tr}\big[B\,(\mathbf{\Sigma} + \kappa(\lambda)I)^{-2}\mathbf{\Sigma}\big].
\end{aligned}
\tag{19}
$$

Equivalently,

$$\mathrm{Tr}\big[A\hat{\mathbf{\Sigma}}(\hat{\mathbf{\Sigma}} + \lambda I)^{-1}\big] \asymp \mathrm{Tr}\big[A\,\mathbf{\Sigma}\,(\mathbf{\Sigma} + \kappa(\lambda)I)^{-1}\big] \tag{20}$$

and

$$
\begin{aligned}
\mathrm{Tr}\big[A\,\hat{\mathbf{\Sigma}}\,(\hat{\mathbf{\Sigma}} + \lambda I)^{-1}\,B\,\hat{\mathbf{\Sigma}}\,(\hat{\mathbf{\Sigma}} + \lambda I)^{-1}\big] \\
\asymp \mathrm{Tr}\big[A\,\mathbf{\Sigma}\,(\mathbf{\Sigma} + \kappa(\lambda)I)^{-1}\,B\,\mathbf{\Sigma}\,(\mathbf{\Sigma} + \kappa(\lambda)I)^{-1}\big] \\
+ \frac{\kappa^2(\lambda)}{n - \mathrm{df}_2(\kappa(\lambda))}\,\mathrm{Tr}\big[A\,(\mathbf{\Sigma} + \kappa(\lambda)I)^{-2}\mathbf{\Sigma}\big]\,\mathrm{Tr}\big[B\,(\mathbf{\Sigma} + \kappa(\lambda)I)^{-2}\mathbf{\Sigma}\big]
\end{aligned}
\tag{21}
$$

where $\kappa(\lambda)$ can be solved from the self-consistent equation (4) above. Note that given the unnormalized trace, the equivalence $\asymp$ shall be understood through convergence of ratio.

**Two-point equivalence for resolvents with different arguments**   This can be further generalized to equivalence with two "Ridge" variables $\lambda, \lambda'$,

$$\mathrm{Tr}\left[A\hat{\mathbf{\Sigma}}(\lambda + \hat{\mathbf{\Sigma}})^{-1}B\hat{\mathbf{\Sigma}}(\lambda' + \hat{\mathbf{\Sigma}})^{-1}\right] \asymp \mathrm{Tr}\left[AT_{\mathbf{\Sigma}}BT'_{\mathbf{\Sigma}}\right] + \tag{22}$$

$$\frac{\kappa\kappa'}{n - \mathrm{df}_2(\kappa, \kappa')}\,\mathrm{Tr}\left[AG_{\mathbf{\Sigma}}\mathbf{\Sigma}G'_{\mathbf{\Sigma}}\right]\,\mathrm{Tr}\left[G'_{\mathbf{\Sigma}}\mathbf{\Sigma}G_{\mathbf{\Sigma}}B\right]$$

where $T_{\mathbf{\Sigma}} := \mathbf{\Sigma}(\mathbf{\Sigma} + \kappa)^{-1}, T'_{\mathbf{\Sigma}} := \mathbf{\Sigma}(\mathbf{\Sigma} + \kappa')^{-1}, G_{\mathbf{\Sigma}} := (\mathbf{\Sigma} + \kappa)^{-1}, G'_{\mathbf{\Sigma}} := (\mathbf{\Sigma} + \kappa')^{-1}$. and $\mathrm{df}_2(\kappa, \kappa') := \mathrm{Tr}[\mathbf{\Sigma}^2 G_{\mathbf{\Sigma}}G'_{\mathbf{\Sigma}}]$. When $\kappa = \kappa'$ it recovers Eq. 21.

As a brief note for derivation, this follows from the deterministic equivalence for free product of matrices $A * B$ presented in Appendix A of Atanasov et al. (2026a). Let $A = \mathbf{\Sigma}$ be population covariance, $B = \frac{1}{n}ZZ^T$ be whitened data covariance, then $A * B = \hat{\mathbf{\Sigma}}$. Thus,

$$\hat{\mathbf{\Sigma}}(\lambda + \hat{\mathbf{\Sigma}})^{-1}M\hat{\mathbf{\Sigma}}(\lambda' + \hat{\mathbf{\Sigma}})^{-1} \asymp T_{\mathbf{\Sigma}}MT'_{\mathbf{\Sigma}} + \kappa\kappa' G_{\mathbf{\Sigma}}\mathbf{\Sigma}G'_{\mathbf{\Sigma}}\frac{\mathrm{Tr}\left[G'_{\mathbf{\Sigma}}\mathbf{\Sigma}G_{\mathbf{\Sigma}}M\right]}{n - \mathrm{df}_2(\kappa, \kappa')} \tag{23}$$

Note that $q$ in their notation corresponds to our $\gamma$ and that their definition of df used normalized trace. This two-point equivalence was derived in Atanasov et al. (2026a; 2025) using a diagrammatic moment-method argument; it could also be derived by extending the leave-one-out arguments used by Bach (2024) to prove the deterministic equivalents with a single $\lambda$ listed above.

## C.2. Proof of the denoiser expectation equivalence (Proposition 4.1)

**Proposition C.1** (Main result, deterministic equivalence of the expectation of score and denoiser). *The optimal linear score and denoiser using empirical covariance has the following deterministic equivalence.*

$$\mathbb{E}_{\hat{\boldsymbol{\Sigma}}}\left[\mathbf{v}^\top \mathbf{D}^*_{\hat{\boldsymbol{\Sigma}}}(\mathbf{x};\sigma)\right] \asymp \mathbf{v}^\top \mathbf{x} + \mathbf{v}^\top \kappa(\sigma^2)(\boldsymbol{\Sigma} + \kappa(\sigma^2)I)^{-1}(\boldsymbol{\mu} - \mathbf{x})$$

$$= \mathbf{v}^\top \boldsymbol{\mu} + \mathbf{v}^\top \boldsymbol{\Sigma}(\boldsymbol{\Sigma} + \kappa(\sigma^2)I)^{-1}(\mathbf{x} - \boldsymbol{\mu})$$

$$\mathbb{E}_{\hat{\boldsymbol{\Sigma}}}\left[\mathbf{v}^\top \mathbf{s}^*_{\hat{\boldsymbol{\Sigma}}}(\mathbf{x};\sigma)\right] \asymp \frac{\kappa(\sigma^2)}{\sigma^2}\mathbf{v}^\top(\boldsymbol{\Sigma} + \kappa(\sigma^2)I)^{-1}(\boldsymbol{\mu} - \mathbf{x})$$

*Proof.* Per assumption, assume the sample mean $\hat{\boldsymbol{\mu}} = \boldsymbol{\mu}$, consider only the effect of empirical covariance $\hat{\boldsymbol{\Sigma}}$,

$$\mathbf{D}^*_{\hat{\boldsymbol{\Sigma}}}(\mathbf{x};\sigma) = \mathbf{x} + \sigma^2(\hat{\boldsymbol{\Sigma}} + \sigma^2 I)^{-1}(\boldsymbol{\mu} - \mathbf{x})$$

Using the deterministic equivalence Eq. 18,20, in the sense that the trace with any independent matrix converge in ratio at limit.

$$(\hat{\boldsymbol{\Sigma}} + \sigma^2 I)^{-1} \asymp \frac{\kappa(\sigma^2)}{\sigma^2}(\boldsymbol{\Sigma} + \kappa(\sigma^2)I)^{-1}$$

$$\hat{\boldsymbol{\Sigma}}(\hat{\boldsymbol{\Sigma}} + \sigma^2 I)^{-1} \asymp \boldsymbol{\Sigma}(\boldsymbol{\Sigma} + \kappa(\sigma^2)I)^{-1}$$

Then, given a fixed measurement vector $\mathbf{v}$, and a noised input $\mathbf{x}$, the projection of score onto a vector can be framed as trace. The equivalence reads,

$$\mathbb{E}_{\hat{\boldsymbol{\Sigma}}}\left[\mathbf{v}^\top \mathbf{s}^*_{\hat{\boldsymbol{\Sigma}}}(\mathbf{x};\sigma)\right] = \mathbb{E}_{\hat{\boldsymbol{\Sigma}}}\left[\mathbf{v}^\top(\hat{\boldsymbol{\Sigma}} + \sigma^2 I)^{-1}(\boldsymbol{\mu} - \mathbf{x})\right]$$

$$= \mathbb{E}_{\hat{\boldsymbol{\Sigma}}}\mathrm{Tr}\left[(\hat{\boldsymbol{\Sigma}} + \sigma^2 I)^{-1}(\boldsymbol{\mu} - \mathbf{x})\mathbf{v}^\top\right]$$

$$\asymp \frac{\kappa(\sigma^2)}{\sigma^2}\mathrm{Tr}\left[(\boldsymbol{\Sigma} + \kappa(\sigma^2)I)^{-1}(\boldsymbol{\mu} - \mathbf{x})\mathbf{v}^\top\right]$$

$$= \frac{\kappa(\sigma^2)}{\sigma^2}\mathbf{v}^\top(\boldsymbol{\Sigma} + \kappa(\sigma^2)I)^{-1}(\boldsymbol{\mu} - \mathbf{x})$$

Similarly, use the other equivalence, the denoiser projection has equivalence,

$$\mathbb{E}_{\hat{\boldsymbol{\Sigma}}}\left[\mathbf{v}^\top \mathbf{D}^*_{\hat{\boldsymbol{\Sigma}}}(\mathbf{x};\sigma)\right] = \mathbf{v}^\top \boldsymbol{\mu} + \mathbb{E}_{\hat{\boldsymbol{\Sigma}}}\left[\mathbf{v}^\top \hat{\boldsymbol{\Sigma}}(\hat{\boldsymbol{\Sigma}} + \sigma^2 I)^{-1}(\mathbf{x} - \boldsymbol{\mu})\right]$$

$$\asymp \mathbf{v}^\top \boldsymbol{\mu} + \mathbf{v}^\top \boldsymbol{\Sigma}(\boldsymbol{\Sigma} + \kappa(\sigma^2)I)^{-1}(\mathbf{x} - \boldsymbol{\mu})$$

$$= \mathbf{v}^\top \mathbf{D}^*_{\boldsymbol{\Sigma}}(\mathbf{x};\kappa^{1/2})$$

Thus, in the expectation sense, the effect of empirical data covariance (finite data) on the denoiser, is equivalent to renormalizing and increasing the effective noise scale $\sigma^2 \to \kappa(\sigma^2)$, similar to adding an adaptive Ridge regularization. □

**Interpretation** Measuring the deviation of the empirical covariance denoiser from the population covariance denoiser, at the same noise scale,

$$\mathbb{E}_{\hat{\boldsymbol{\Sigma}}}\left[\mathbf{v}^\top\left(\mathbf{D}^*_{\hat{\boldsymbol{\Sigma}}}(\mathbf{x};\sigma) - \mathbf{D}^*_{\boldsymbol{\Sigma}}(\mathbf{x};\sigma)\right)\right]$$

$$\asymp \mathbf{v}^\top\left[\kappa(\sigma^2)(\boldsymbol{\Sigma} + \kappa(\sigma^2)I)^{-1} - \sigma^2(\boldsymbol{\Sigma} + \sigma^2 I)^{-1}\right](\boldsymbol{\mu} - \mathbf{x})$$

Using push through identity $A^{-1} - B^{-1} = A^{-1}(B - A)B^{-1}$,

$$\kappa(\boldsymbol{\Sigma} + \kappa I)^{-1} - \sigma^2(\boldsymbol{\Sigma} + \sigma^2 I)^{-1}$$

$$= \kappa\sigma^2(\boldsymbol{\Sigma} + \kappa I)^{-1}(\boldsymbol{\Sigma} + \sigma^2 I)^{-1}\left(\frac{1}{\sigma^2}(\boldsymbol{\Sigma} + \sigma^2 I) - \frac{1}{\kappa}(\boldsymbol{\Sigma} + \kappa I)\right)$$

$$= (\kappa(\sigma^2) - \sigma^2)\boldsymbol{\Sigma}(\boldsymbol{\Sigma} + \kappa I)^{-1}(\boldsymbol{\Sigma} + \sigma^2 I)^{-1}$$

We can represent the deviation as a resolvent product. This makes it clear that the deviation is proportional to the effect of renormalization $(\kappa(\sigma^2) - \sigma^2)$.

$$\mathbb{E}_{\hat{\boldsymbol{\Sigma}}}\left[\mathbf{v}^\top\left(\mathbf{D}^*_{\hat{\boldsymbol{\Sigma}}}(\mathbf{x};\sigma) - \mathbf{D}^*_{\boldsymbol{\Sigma}}(\mathbf{x};\sigma)\right)\right]$$

$$\asymp (\kappa(\sigma^2) - \sigma^2)\mathbf{v}^\top\boldsymbol{\Sigma}(\boldsymbol{\Sigma} + \kappa I)^{-1}(\boldsymbol{\Sigma} + \sigma^2 I)^{-1}(\boldsymbol{\mu} - \mathbf{x})$$

Setting the measurement vector along population eigenvector $\mathbf{u}_k$, with eigenvalue $\lambda_k$, then the deviation reads

$$\mathbb{E}_{\hat{\boldsymbol{\Sigma}}}\left[\mathbf{u}_k^\top\left(\mathbf{D}_{\hat{\boldsymbol{\Sigma}}}^*(\mathbf{x};\sigma) - \mathbf{D}_{\boldsymbol{\Sigma}}^*(\mathbf{x};\sigma)\right)\right] \asymp \frac{\lambda_k(\kappa - \sigma^2)}{(\lambda_k + \sigma^2)(\lambda_k + \kappa)}\mathbf{u}_k^\top(\boldsymbol{\mu} - \mathbf{x})$$

It's easy to see the deviation affects the lower eigenspace more. Viewed as a function of $\lambda_k$, the coefficient

$$f(\lambda_k) = \frac{\lambda_k(\kappa - \sigma^2)}{(\lambda_k + \sigma^2)(\lambda_k + \kappa)}$$

is unimodal, peaking at the geometric mean of the two scales, $\lambda_k^\star = \sqrt{\sigma^2\kappa}$, where it attains

$$f(\lambda_k^\star) = \frac{\kappa - \sigma^2}{\left(\sigma + \sqrt{\kappa}\right)^2} = \frac{\sqrt{\kappa} - \sigma}{\sqrt{\kappa} + \sigma}.$$

Since $\kappa(\sigma^2) > \sigma^2$ (renormalization inflates the noise scale), this peak sits somewhat above $\sigma^2$ but still in the low-to-moderate part of the spectrum, and the weight decays to zero on both sides:

- For top eigenmodes ($\lambda_k \gg \kappa$), $f(\lambda_k) \approx (\kappa - \sigma^2)/\lambda_k \to 0$: large-variance directions are essentially unaffected, because the resolvents $(\boldsymbol{\Sigma} + \kappa I)^{-1}$ and $(\boldsymbol{\Sigma} + \sigma^2 I)^{-1}$ both act like $\lambda_k^{-1}$ and the additive shifts $\kappa, \sigma^2$ are negligible relative to $\lambda_k$.

- For very small eigenmodes ($\lambda_k \to 0$), $f(\lambda_k) \to 0$ as well, suppressed by the leading $\lambda_k$ factor.

- In between, modes with $\lambda_k$ comparable to $\sqrt{\sigma^2\kappa}$—the lower, signal-to-noise-marginal part of the spectrum—receive the largest correction, of magnitude $(\sqrt{\kappa} - \sigma)/(\sqrt{\kappa} + \sigma)$.

Thus the deviation is the most salient when $\kappa$ deviates from $\sigma^2$; and along the spectrum, it peaked at the eigenmodes with variance around $\sigma^2$ and $\kappa$.

### C.3. Proof of the denoiser fluctuation equivalence (Proposition 4.2)

**Proposition C.2** (Main result, deterministic equivalence of the denoiser variance). *Assuming $\hat{\boldsymbol{\mu}} = \boldsymbol{\mu}$, across dataset realizations of size $n$, the variance of the optimal empirical linear denoiser at point $\mathbf{x}$ in direction $\mathbf{v}$, given by $\mathbf{v}^\top \mathcal{S}_D(\mathbf{x})\mathbf{v}$, admits the following deterministic equivalence.*

$$\mathbf{v}^\top \mathcal{S}_D(\mathbf{x})\mathbf{v} = \mathrm{Var}_{\hat{\boldsymbol{\Sigma}}}[\mathbf{v}^\top \mathbf{D}_{\hat{\boldsymbol{\Sigma}}}^*(\mathbf{x};\sigma)] \tag{24}$$

$$\asymp \frac{\kappa(\sigma^2)^2}{n - \mathrm{df}_2\big(\kappa(\sigma^2)\big)} \underbrace{\left(\mathbf{v}^\top\left(\boldsymbol{\Sigma} + \kappa(\sigma^2)I\right)^{-2}\boldsymbol{\Sigma}\mathbf{v}\right)}_{anisotropy:\ \diamond(\mathbf{v},\kappa,\boldsymbol{\Sigma})} \underbrace{\left((\mathbf{x} - \boldsymbol{\mu})^\top\left(\boldsymbol{\Sigma} + \kappa(\sigma^2)I\right)^{-2}\boldsymbol{\Sigma}(\mathbf{x} - \boldsymbol{\mu})\right)}_{inhomogeneity:\ \diamond(\mathbf{x}-\boldsymbol{\mu},\kappa,\boldsymbol{\Sigma})}$$

*Proof.* Next, we examine the covariance of denoiser due to dataset realization, the score variance reads,

$$\mathcal{S}_s := Cov_{\hat{\boldsymbol{\Sigma}}}[\mathbf{s}_{\hat{\boldsymbol{\Sigma}}}^*(\mathbf{x};\sigma)] = \mathbb{E}_{\hat{\boldsymbol{\Sigma}}}\mathbf{s}_{\hat{\boldsymbol{\Sigma}}}^*(\mathbf{x};\sigma)\mathbf{s}_{\hat{\boldsymbol{\Sigma}}}^*(\mathbf{x};\sigma)^\top - \left(\mathbb{E}_{\hat{\boldsymbol{\Sigma}}}\mathbf{s}_{\hat{\boldsymbol{\Sigma}}}^*(\mathbf{x};\sigma)\right)\left(\mathbb{E}_{\hat{\boldsymbol{\Sigma}}}\mathbf{s}_{\hat{\boldsymbol{\Sigma}}}^*(\mathbf{x};\sigma)\right)^\top$$

$$= \mathbb{E}_{\hat{\boldsymbol{\Sigma}}}\left[(\hat{\boldsymbol{\Sigma}} + \sigma^2 I)^{-1}(\boldsymbol{\mu} - \mathbf{x})(\boldsymbol{\mu} - \mathbf{x})^\top(\hat{\boldsymbol{\Sigma}} + \sigma^2 I)^{-1}\right] -$$

$$\mathbb{E}_{\hat{\boldsymbol{\Sigma}}}\left[(\hat{\boldsymbol{\Sigma}} + \sigma^2 I)^{-1}(\boldsymbol{\mu} - \mathbf{x})\right]\mathbb{E}_{\hat{\boldsymbol{\Sigma}}}\left[(\boldsymbol{\mu} - \mathbf{x})^\top(\hat{\boldsymbol{\Sigma}} + \sigma^2 I)^{-1}\right]$$

$$= \mathbb{E}_{\hat{\boldsymbol{\Sigma}}}\left[(\hat{\boldsymbol{\Sigma}} + \sigma^2 I)^{-1}(\boldsymbol{\mu} - \mathbf{x})(\boldsymbol{\mu} - \mathbf{x})^\top(\hat{\boldsymbol{\Sigma}} + \sigma^2 I)^{-1}\right] -$$

$$\mathbb{E}_{\hat{\boldsymbol{\Sigma}}}\left[(\hat{\boldsymbol{\Sigma}} + \sigma^2 I)^{-1}\right](\boldsymbol{\mu} - \mathbf{x})(\boldsymbol{\mu} - \mathbf{x})^\top\mathbb{E}_{\hat{\boldsymbol{\Sigma}}}\left[(\hat{\boldsymbol{\Sigma}} + \sigma^2 I)^{-1}\right]$$

Note that the variance of denoiser and that of score has the simple scaling relationship, so we just need to study the score.

$$\mathcal{S}_{\mathbf{D}} = \sigma^4 \mathcal{S}_s$$

We are interested in the variance of score vector along a fixed probe vector $\mathbf{v}$,

$$\mathbf{v}^\top \mathcal{S}_s \mathbf{v} = Var_{\hat{\mathbf{\Sigma}}}[\mathbf{v}^\top \mathbf{s}^*_{\hat{\mathbf{\Sigma}}}(\mathbf{x}; \sigma)]$$

$$= \mathbb{E}_{\hat{\mathbf{\Sigma}}}\left[\mathbf{v}^\top (\hat{\mathbf{\Sigma}} + \sigma^2 I)^{-1}(\boldsymbol{\mu} - \mathbf{x})(\boldsymbol{\mu} - \mathbf{x})^\top (\hat{\mathbf{\Sigma}} + \sigma^2 I)^{-1}\mathbf{v}\right] - \left(\mathbf{v}^\top \mathbb{E}_{\hat{\mathbf{\Sigma}}}\left[(\hat{\mathbf{\Sigma}} + \sigma^2 I)^{-1}\right](\boldsymbol{\mu} - \mathbf{x})\right)^2$$

$$= \underbrace{\mathbb{E}_{\hat{\mathbf{\Sigma}}} \operatorname{Tr}\left[\mathbf{v}\mathbf{v}^\top (\hat{\mathbf{\Sigma}} + \sigma^2 I)^{-1}(\boldsymbol{\mu} - \mathbf{x})(\boldsymbol{\mu} - \mathbf{x})^\top (\hat{\mathbf{\Sigma}} + \sigma^2 I)^{-1}\right]}_{\text{2nd moment}} - \underbrace{\left(\mathbb{E}_{\hat{\mathbf{\Sigma}}} \operatorname{Tr}\left[(\hat{\mathbf{\Sigma}} + \sigma^2 I)^{-1}(\boldsymbol{\mu} - \mathbf{x})\mathbf{v}^\top\right]\right)^2}_{\text{1st moment}}$$

The two terms can be tackled by one-point and two-point equivalence Eq. 19,18. Abbreviating $A := \mathbf{v}\mathbf{v}^\top$, $B := (\boldsymbol{\mu} - \mathbf{x})(\boldsymbol{\mu} - \mathbf{x})^\top$, $z := \sigma^2$.

$$\operatorname{Tr}\left[A\,(\hat{\mathbf{\Sigma}} + zI)^{-1}\right] \sim \frac{\kappa(z)}{z} \operatorname{Tr}\left[A(\mathbf{\Sigma} + \kappa(z)I)^{-1}\right]$$

$$\operatorname{Tr}\left[A(\hat{\mathbf{\Sigma}} + zI)^{-1} B(\hat{\mathbf{\Sigma}} + zI)^{-1}\right] \sim \frac{\kappa(z)^2}{z^2} \operatorname{Tr}\left[A\left(\mathbf{\Sigma} + \kappa(z)I\right)^{-1} B\left(\mathbf{\Sigma} + \kappa(z)I\right)^{-1}\right]$$
$$+ \frac{\kappa(z)^2}{z^2} \operatorname{Tr}\left[A\left(\mathbf{\Sigma} + \kappa(z)I\right)^{-2}\mathbf{\Sigma}\right] \operatorname{Tr}\left[B\left(\mathbf{\Sigma} + \kappa(z)I\right)^{-2}\mathbf{\Sigma}\right]\frac{1}{n - \operatorname{df}_2(\kappa(z))}$$

The 2nd moment term is equivalent to,

$$\operatorname{Tr}\left[A(\hat{\mathbf{\Sigma}} + zI)^{-1} B(\hat{\mathbf{\Sigma}} + zI)^{-1}\right]$$
$$\sim \frac{\kappa(z)^2}{z^2} \operatorname{Tr}\left[\mathbf{v}\mathbf{v}^\top \left(\mathbf{\Sigma} + \kappa(z)I\right)^{-1}(\boldsymbol{\mu} - \mathbf{x})(\boldsymbol{\mu} - \mathbf{x})^\top \left(\mathbf{\Sigma} + \kappa(z)I\right)^{-1}\right]$$
$$+ \frac{\kappa(z)^2}{z^2}\frac{1}{n - \operatorname{df}_2(\kappa(z))} \operatorname{Tr}\left[\mathbf{v}\mathbf{v}^\top \left(\mathbf{\Sigma} + \kappa(z)I\right)^{-2}\mathbf{\Sigma}\right] \operatorname{Tr}\left[(\boldsymbol{\mu} - \mathbf{x})(\boldsymbol{\mu} - \mathbf{x})^\top \left(\mathbf{\Sigma} + \kappa(z)I\right)^{-2}\mathbf{\Sigma}\right]$$
$$= \frac{\kappa(z)^2}{z^2}\left(\mathbf{v}^\top \left(\mathbf{\Sigma} + \kappa(z)I\right)^{-1}(\boldsymbol{\mu} - \mathbf{x})\right)^2$$
$$+ \frac{\kappa(z)^2}{z^2}\frac{1}{n - \operatorname{df}_2(\kappa(z))}\left(\mathbf{v}^\top \left(\mathbf{\Sigma} + \kappa(z)I\right)^{-2}\mathbf{\Sigma}\mathbf{v}\right)\left((\boldsymbol{\mu} - \mathbf{x})^\top \left(\mathbf{\Sigma} + \kappa(z)I\right)^{-2}\mathbf{\Sigma}(\boldsymbol{\mu} - \mathbf{x})\right)$$

The first moment term is equivalent to,

$$\operatorname{Tr}\left[(\hat{\mathbf{\Sigma}} + zI)^{-1}(\boldsymbol{\mu} - \mathbf{x})\mathbf{v}^\top\right] \sim \frac{\kappa(z)}{z} \operatorname{Tr}\left[\left(\mathbf{\Sigma} + \kappa(z)I\right)^{-1}(\boldsymbol{\mu} - \mathbf{x})\mathbf{v}^\top\right]$$
$$= \frac{\kappa(z)}{z}\mathbf{v}^\top \left(\mathbf{\Sigma} + \kappa(z)I\right)^{-1}(\boldsymbol{\mu} - \mathbf{x})$$

Thus, combining the two terms, we obtain the variance of score at noised datapoint $\mathbf{x}$, along direction $\mathbf{v}$,

$$\mathbf{v}^\top \mathcal{S}_s(\mathbf{x})\mathbf{v} = Var_{\hat{\mathbf{\Sigma}}}[\mathbf{v}^\top \mathbf{s}^*_{\hat{\mathbf{\Sigma}}}(\mathbf{x}; \sigma)]$$
$$= \mathbb{E}_{\hat{\mathbf{\Sigma}}} \operatorname{Tr}\left[\mathbf{v}\mathbf{v}^\top (\hat{\mathbf{\Sigma}} + \sigma^2 I)^{-1}(\boldsymbol{\mu} - \mathbf{x})(\boldsymbol{\mu} - \mathbf{x})^\top (\hat{\mathbf{\Sigma}} + \sigma^2 I)^{-1}\right]$$
$$- \left(\mathbb{E}_{\hat{\mathbf{\Sigma}}} \operatorname{Tr}\left[(\hat{\mathbf{\Sigma}} + \sigma^2 I)^{-1}(\boldsymbol{\mu} - \mathbf{x})\mathbf{v}^\top\right]\right)^2$$
$$\sim \frac{\kappa(z)^2}{z^2}\left(\mathbf{v}^\top \left(\mathbf{\Sigma} + \kappa(z)I\right)^{-1}(\boldsymbol{\mu} - \mathbf{x})\right)^2$$
$$+ \frac{\kappa(z)^2}{z^2}\left(\mathbf{v}^\top \left(\mathbf{\Sigma} + \kappa(z)I\right)^{-2}\mathbf{\Sigma}\mathbf{v}\right)\left((\boldsymbol{\mu} - \mathbf{x})^\top \left(\mathbf{\Sigma} + \kappa(z)I\right)^{-2}\mathbf{\Sigma}(\boldsymbol{\mu} - \mathbf{x})\right)\frac{1}{n - \operatorname{df}_2(\kappa(z))}$$
$$- \left(\frac{\kappa(z)}{z}\mathbf{v}^\top \left(\mathbf{\Sigma} + \kappa(z)I\right)^{-1}(\boldsymbol{\mu} - \mathbf{x})\right)^2$$
$$= \frac{1}{n - \operatorname{df}_2(\kappa(z))}\frac{\kappa(z)^2}{z^2}\left(\mathbf{v}^\top \left(\mathbf{\Sigma} + \kappa(z)I\right)^{-2}\mathbf{\Sigma}\mathbf{v}\right)\left((\boldsymbol{\mu} - \mathbf{x})^\top \left(\mathbf{\Sigma} + \kappa(z)I\right)^{-2}\mathbf{\Sigma}(\boldsymbol{\mu} - \mathbf{x})\right)$$
$$(z \mapsto \sigma^2) = \frac{1}{n - \operatorname{df}_2(\kappa(\sigma^2))}\frac{\kappa(\sigma^2)^2}{\sigma^4}\left(\mathbf{v}^\top \left(\mathbf{\Sigma} + \kappa(\sigma^2)I\right)^{-2}\mathbf{\Sigma}\mathbf{v}\right)\left((\boldsymbol{\mu} - \mathbf{x})^\top \left(\mathbf{\Sigma} + \kappa(\sigma^2)I\right)^{-2}\mathbf{\Sigma}(\boldsymbol{\mu} - \mathbf{x})\right)$$

Per simple scaling, the variance of denoisers reads,

$$\mathbf{v}^\top \mathcal{S}_D(\mathbf{x})\mathbf{v} = \sigma^4 \mathbf{v}^\top \mathcal{S}_s(\mathbf{x})\mathbf{v}$$

$$\sim \frac{\kappa(\sigma^2)^2}{n - \mathrm{df}_2\big(\kappa(\sigma^2)\big)} \underbrace{\left(\mathbf{v}^\top \big(\mathbf{\Sigma} + \kappa(\sigma^2)I\big)^{-2}\mathbf{\Sigma}\mathbf{v}\right)}_{\diamond(\mathbf{v},\kappa,\mathbf{\Sigma})} \underbrace{\left((\boldsymbol{\mu} - \mathbf{x})^\top \big(\mathbf{\Sigma} + \kappa(\sigma^2)I\big)^{-2}\mathbf{\Sigma}(\boldsymbol{\mu} - \mathbf{x})\right)}_{\diamond(\boldsymbol{\mu}-\mathbf{x},\kappa,\mathbf{\Sigma})}$$

$\square$

### C.3.1. INTERPRETATION AND DERIVATIONS

**Dependence on the probe direction $\mathbf{v}$**  This dependency on $\mathbf{v}$ describes the *anisotropy of uncertainty*, or the variance of the score/denoiser prediction along different directions.

$$\diamond(\mathbf{v},\kappa,\mathbf{\Sigma}) := \mathbf{v}^\top \big(\mathbf{\Sigma} + \kappa(\sigma^2)I\big)^{-2}\mathbf{\Sigma}\mathbf{v}$$

$$= \mathbf{v}^\top U \frac{\Lambda}{(\Lambda + \kappa(\sigma^2))^2} U^\top \mathbf{v}$$

By assumption, the probe vector $\mathbf{v}$ is a unit vector. Then this dependency is determined by the diagonal matrix $\frac{\Lambda}{(\Lambda+\kappa(\sigma^2))^2} = \mathrm{diag}\left(\frac{\lambda_k}{(\lambda_k+\kappa(\sigma^2))^2}\right)$ and the alignment between $\mathbf{v}$ and the eigenbasis $U$.

Consider when the probing vector is aligned exactly with the $k$-th eigenvector $\mathbf{u}_k$, this term reads

$$\diamond(\mathbf{u}_k,\kappa,\mathbf{\Sigma}) = \mathbf{u}_k^\top \big(\mathbf{\Sigma} + \kappa(\sigma^2)I\big)^{-2}\mathbf{\Sigma}\mathbf{u}_k$$

$$= \frac{\lambda_k}{(\lambda_k + \kappa(\sigma^2))^2}$$

$$=: \chi(\lambda_k, \kappa(\sigma^2))$$

We can discuss the different regimes of $\chi(\lambda_k, \kappa)$ depending on $\lambda_k$ and $\kappa(\sigma^2)$

- High noise regime $\lambda_k \ll \kappa$: $\chi(\lambda_k, \kappa) \approx \frac{\lambda_k}{\kappa^2}$, so $\chi(\lambda_k, \kappa)$ increases with $\lambda_k$. Higher variance directions have larger uncertainties.

- Low noise regime $\lambda_k \gg \kappa$: $\chi(\lambda_k, \kappa) \approx \frac{1}{\lambda_k}$, so $\chi(\lambda_k, \kappa)$ will decrease with $\lambda_k$. Lower variance directions have larger uncertainties.

- Regarding $\kappa$, $\chi(\lambda_k, \kappa)$ decreases monotonically with $\kappa$, i.e. the higher the noise scale, the smaller the variance.

- Regarding $\lambda_k$, $\chi(\lambda_k, \kappa)$ has one unique maximum, where $\arg\max_\lambda \chi(\lambda, \kappa) = \kappa$, and $\max_\lambda \chi(\lambda, \kappa) = \frac{1}{4\kappa}$. Thus, it is a bell shaped function of $\lambda_k$. (Proof below.)

  - This shows that at different noise level or $\kappa(\sigma^2)$, there is always some direction with variance comparable to $\kappa(\sigma^2)$ which will have the largest variance!
  - Further, the largest variance will be inversely proportional to $\kappa(\sigma^2)$, i.e. generally larger at lower noise.

This result shows that the anisotropy of the uncertainty depends on the renormalized noise scale $\kappa(\sigma^2)$, and the maximal uncertainty is focused around the PC dimensions with variance similar to $\kappa(\sigma^2)$.

**Proof of unique maximum of** $\chi(\lambda, \kappa)$

*Proof.* Given

$$\chi(\lambda, \kappa) = \frac{\lambda}{(\lambda + \kappa)^2}$$

Then

$$\frac{d\chi(\lambda, \kappa)}{d\lambda} = \frac{(\lambda + \kappa)^2 - 2(\lambda + \kappa)\lambda}{(\lambda + \kappa)^4}$$

$$= \frac{\kappa - \lambda}{(\lambda + \kappa)^3}$$

Setting the gradient to zero yields the unique stationary point, $\kappa = \lambda$.
Given $\kappa, \lambda > 0$, we have the unique maximum w.r.t. $\lambda$.

$$\arg\max_{\lambda} \chi(\lambda, \kappa) = \kappa$$

$$\max_{\lambda} \chi(\lambda, \kappa) = \frac{1}{4\kappa}$$

□

**Dependence on the probe point x.** The dependency on probe point $\mathbf{x}$ tells us about the spatial inhomogeneity of the uncertainty.

$$\diamond(\mathbf{x} - \boldsymbol{\mu}, \kappa, \boldsymbol{\Sigma}) = (\mathbf{x} - \boldsymbol{\mu})^\top \left(\boldsymbol{\Sigma} + \kappa(\sigma^2)I\right)^{-2} \boldsymbol{\Sigma}(\mathbf{x} - \boldsymbol{\mu})$$

$$= (\mathbf{x} - \boldsymbol{\mu})^\top U \frac{\Lambda}{(\Lambda + \kappa(\sigma^2))^2} U^\top (\mathbf{x} - \boldsymbol{\mu})$$

$$= \sum_k \frac{\lambda_k}{(\lambda_k + \kappa(\sigma^2))^2} \left(\mathbf{u}_k^\top (\mathbf{x} - \boldsymbol{\mu})\right)^2$$

$$= \sum_k \chi(\lambda_k, \kappa(\sigma^2)) \left(\mathbf{u}_k^\top (\mathbf{x} - \boldsymbol{\mu})\right)^2$$

This is similar to the dependency above, except that now the argument $\mathbf{x} - \boldsymbol{\mu}$ is no longer unit normed, but any probing direction in the sample space.

Note, generally the noised sample $\mathbf{x}$ from a certain realization of dataset is distributed like $\mathcal{N}(\boldsymbol{\mu}, \hat{\boldsymbol{\Sigma}} + \sigma^2 I)$ (under Gaussian data assumption), so

$$\mathbf{v}^\top (\mathbf{x} - \boldsymbol{\mu}) \sim \mathcal{N}(0, \mathbf{v}^\top (\hat{\boldsymbol{\Sigma}} + \sigma^2 I)\mathbf{v})$$

Consider a probe point on the hyperelliptical shell defined by $\mathcal{N}(\boldsymbol{\mu}, \boldsymbol{\Sigma} + \sigma^2 I)$, then if the point falls on the line $\mathbf{x} = \boldsymbol{\mu} + c\mathbf{u}_k$. $\|\mathbf{x} - \boldsymbol{\mu}\|^2 = c^2 \approx \sigma^2 + \lambda_k$

Then

$$\diamond(\mathbf{x} - \boldsymbol{\mu}, \kappa, \boldsymbol{\Sigma}) = \diamond(c\mathbf{u}_k, \kappa, \boldsymbol{\Sigma})$$

$$= c^2 \frac{\lambda_k}{(\lambda_k + \kappa(\sigma^2))^2}$$

$$\approx \frac{(\lambda_k + \sigma^2)\lambda_k}{(\lambda_k + \kappa(\sigma^2))^2}$$

$$= (\sigma^2 + \lambda_k) \chi(\lambda_k, \kappa(\sigma^2))$$

$$= \xi(\lambda_k, \sigma^2)$$

- **High noise regime**, $\kappa > \sigma^2 \gg \lambda$, then $\xi(\lambda, \sigma^2) \approx \frac{\sigma^2 \lambda}{\kappa^2(\sigma^2)} < \frac{\lambda}{\kappa(\sigma^2)} \ll 1$, which scale linearly with PC variance $\lambda$, higher the PC, larger the variance.

- **Low noise regime**, $\lambda \gg \kappa > \sigma^2$, then $\xi(\lambda, \sigma^2) \approx 1$. Then all points on the ellipsoid have large variance.

- At any fixed $\sigma^2$, this function increases monotonically with $\lambda_k$.

- The score or denoiser variance is larger when the probing point $\boldsymbol{\mu} + c\mathbf{u}_k$ is deviating along those higher variance directions $\mathbf{u}_k$.
- When the probing point is deviating along low variance directions, the variance is lower.

---

**Derivation of properties of $\xi(\lambda, \sigma^2)$**

$$\xi(\lambda, \sigma^2) = \frac{(\sigma^2 + \lambda)\lambda}{(\lambda + \kappa(\sigma^2))^2}$$

Derivative

$$
\begin{aligned}
\frac{d\xi(\lambda, \sigma^2)}{d\lambda} &= \frac{(\sigma^2 + 2\lambda)(\lambda + \kappa(\sigma^2))^2 - 2(\lambda + \kappa(\sigma^2))(\sigma^2 + \lambda)\lambda}{(\lambda + \kappa(\sigma^2))^4} \\
&= \frac{(\sigma^2 + 2\lambda)(\lambda + \kappa(\sigma^2)) - 2(\sigma^2 + \lambda)\lambda}{(\lambda + \kappa(\sigma^2))^3} \\
&= \frac{(\sigma^2 + 2\lambda)\kappa(\sigma^2) - \lambda\sigma^2}{(\lambda + \kappa(\sigma^2))^3} \\
&= \frac{\sigma^2\kappa(\sigma^2) + (2\kappa(\sigma^2) - \sigma^2)\lambda}{(\lambda + \kappa(\sigma^2))^3}
\end{aligned}
$$

Note that through the self-consistent equation $\kappa(\sigma^2) - \sigma^2 > 0$, thus $\frac{d\xi(\lambda,\sigma^2)}{d\lambda} > 0, \forall \lambda$. The function is monotonically increasing for $\lambda$.

Given that $\kappa(\sigma^2) > \sigma^2 > 0$, we have bounds

$$\xi(\lambda, \sigma^2) = \frac{(\sigma^2 + \lambda)\lambda}{(\lambda + \kappa(\sigma^2))^2} < \frac{\lambda}{\lambda + \kappa(\sigma^2)} < 1$$

---

**Overall scaling with sample size**  Finally, we marginalize over space and direction, obtaining an overall quantification of consistency of denoiser, and study its scaling property.

First, *marginalizing (summing) all directions*, we have

$$
\begin{aligned}
\sum_k \diamond(\mathbf{u}_k, \kappa, \boldsymbol{\Sigma}) &= \sum_k \mathbf{u}_k^\top \big(\boldsymbol{\Sigma} + \kappa(\sigma^2)I\big)^{-2}\boldsymbol{\Sigma}\mathbf{u}_k \\
&= \mathrm{Tr}\Big[\big(\boldsymbol{\Sigma} + \kappa(\sigma^2)I\big)^{-2}\boldsymbol{\Sigma}\Big]
\end{aligned}
$$

This can be further abbreviated as following,

$$
\begin{aligned}
\sum_k \diamond(\mathbf{u}_k, \kappa, \boldsymbol{\Sigma}) &= \mathrm{Tr}\Big[\big(\boldsymbol{\Sigma} + \kappa(\sigma^2)I\big)^{-2}\boldsymbol{\Sigma}I\Big] \\
&= \mathrm{Tr}\Big[\big(\boldsymbol{\Sigma} + \kappa(\sigma^2)I\big)^{-2}\boldsymbol{\Sigma}\frac{1}{\kappa(\sigma^2)}(\boldsymbol{\Sigma} + \kappa(\sigma^2)I - \boldsymbol{\Sigma})\Big] \\
&= \frac{1}{\kappa(\sigma^2)}\Big(\mathrm{Tr}\Big[\big(\boldsymbol{\Sigma} + \kappa(\sigma^2)I\big)^{-1}\boldsymbol{\Sigma}\Big] - \mathrm{Tr}\Big[\big(\boldsymbol{\Sigma} + \kappa(\sigma^2)I\big)^{-2}\boldsymbol{\Sigma}^2\Big]\Big) \\
&= \frac{\mathrm{df}_1(\kappa) - \mathrm{df}_2(\kappa)}{\kappa}
\end{aligned}
$$

Next, *marginalize (averaging) over space*. Here we consider the noised distribution starting from the true target distribution $\mathbf{x} \sim p(\mathbf{x}; \sigma) = p_0(\mathbf{x}) * \mathcal{N}(0, \sigma^2 I)$. For us, the only thing that matters is the second moment, so for an arbitrary distribution we have,

$$\mathbb{E}_\mathbf{x}[(\mathbf{x} - \boldsymbol{\mu})(\mathbf{x} - \boldsymbol{\mu})^\top] = \boldsymbol{\Sigma} + \sigma^2 I$$

Thus,

$$
\begin{aligned}
\mathbb{E}_\mathbf{x}\diamond(\mathbf{x} - \boldsymbol{\mu}, \kappa, \boldsymbol{\Sigma}) &= \mathbb{E}_\mathbf{x}\Big[(\mathbf{x} - \boldsymbol{\mu})^\top\big(\boldsymbol{\Sigma} + \kappa(\sigma^2)I\big)^{-2}\boldsymbol{\Sigma}(\mathbf{x} - \boldsymbol{\mu})\Big] \\
&= \mathrm{Tr}\Big[\big(\boldsymbol{\Sigma} + \sigma^2 I\big)\big(\boldsymbol{\Sigma} + \kappa(\sigma^2)I\big)^{-2}\boldsymbol{\Sigma}\Big]
\end{aligned}
$$

This can also be abbreviated using degree of freedom,

$$
\begin{aligned}
\mathbb{E}_{\mathbf{x}} \diamond(\mathbf{x} - \boldsymbol{\mu}, \kappa, \boldsymbol{\Sigma}) &= \mathrm{Tr}\Big[(\boldsymbol{\Sigma} + \sigma^2 I)(\boldsymbol{\Sigma} + \kappa(\sigma^2)I)^{-2}\boldsymbol{\Sigma}\Big] \\
&= \mathrm{Tr}\Big[(\boldsymbol{\Sigma} + \kappa(\sigma^2)I)^{-2}\boldsymbol{\Sigma}^2\Big] + \sigma^2 \mathrm{Tr}\Big[(\boldsymbol{\Sigma} + \kappa(\sigma^2)I)^{-2}\boldsymbol{\Sigma}\Big] \\
&= \mathrm{df}_2(\kappa) + \frac{\sigma^2}{\kappa}(\mathrm{df}_1(\kappa) - \mathrm{df}_2(\kappa)) \\
&= \frac{\sigma^2}{\kappa}\mathrm{df}_1(\kappa) + (1 - \frac{\sigma^2}{\kappa})\mathrm{df}_2(\kappa)
\end{aligned}
$$

Thus, we have

$$
\begin{aligned}
\mathbb{E}_{\mathbf{x}} \sum_k \mathbf{u}_k^\top \mathcal{S}_D(\mathbf{x}) \mathbf{u}_k &\asymp \frac{\kappa(\sigma^2)^2}{n - \mathrm{df}_2\big(\kappa(\sigma^2)\big)} \sum_k \underbrace{\Big(\mathbf{u}_k^\top (\boldsymbol{\Sigma} + \kappa(\sigma^2)I)^{-2}\boldsymbol{\Sigma}\mathbf{u}_k\Big)}_{\diamond(\mathbf{v}, \kappa, \boldsymbol{\Sigma})} \mathbb{E}_{\mathbf{x}} \underbrace{\Big((\boldsymbol{\mu} - \mathbf{x})^\top (\boldsymbol{\Sigma} + \kappa(\sigma^2)I)^{-2}\boldsymbol{\Sigma}(\boldsymbol{\mu} - \mathbf{x})\Big)}_{\diamond(\boldsymbol{\mu} - \mathbf{x}, \kappa, \boldsymbol{\Sigma})} \\
&= \frac{\kappa(\sigma^2)^2}{n - \mathrm{df}_2\big(\kappa(\sigma^2)\big)} \mathrm{Tr}\Big[(\boldsymbol{\Sigma} + \kappa(\sigma^2)I)^{-2}\boldsymbol{\Sigma}\Big] \mathrm{Tr}\Big[(\boldsymbol{\Sigma} + \sigma^2 I)(\boldsymbol{\Sigma} + \kappa(\sigma^2)I)^{-2}\boldsymbol{\Sigma}\Big] \\
&= \frac{\kappa(\sigma^2)^2}{n - \mathrm{df}_2\big(\kappa(\sigma^2)\big)} \times \frac{\mathrm{df}_1(\kappa) - \mathrm{df}_2(\kappa)}{\kappa} \times \Big(\frac{\sigma^2}{\kappa}\mathrm{df}_1(\kappa) + (1 - \frac{\sigma^2}{\kappa})\mathrm{df}_2(\kappa)\Big) \\
&= \frac{\big(\mathrm{df}_1(\kappa) - \mathrm{df}_2(\kappa)\big) \times \big(\sigma^2 \mathrm{df}_1(\kappa) + (\kappa - \sigma^2)\mathrm{df}_2(\kappa)\big)}{n - \mathrm{df}_2\big(\kappa(\sigma^2)\big)} \\
&=: \Delta(n, \sigma^2, \Lambda)
\end{aligned}
$$

$$
\Delta(n, \sigma^2, \Lambda) = \frac{(\mathrm{df}_1(\kappa) - \mathrm{df}_2(\kappa))\big(\sigma^2 \mathrm{df}_1(\kappa) + (\kappa - \sigma^2)\mathrm{df}_2(\kappa)\big)}{n - \mathrm{df}_2(\kappa)} \tag{25}
$$

Now, marginalized over space and direction, this is only a function of the population spectrum, sample number and noise scale. Note $n$ is the sample number, so it makes sense when $n$ goes to infinity, then $\hat{\boldsymbol{\Sigma}} \to \boldsymbol{\Sigma}$ and $\kappa \to \sigma^2$, the variance reduces to zero.

Basically the higher the $\kappa$, the smaller the $\mathrm{df}_2(\kappa)$, so $n - \mathrm{df}_2\big(\kappa(\sigma^2)\big)$ will be larger, which scales down $\frac{1}{n - \mathrm{df}_2\big(\kappa(\sigma^2)\big)} \frac{\kappa(\sigma^2)^2}{\sigma^4}$.

When we compare our theory with empirical measurements of denoiser or sample deviations between two splits, we use the following lemma to predict the expected MSE deviation from the variance.

**Lemma C.3** (Expected MSE between two i.i.d. samples doubles the variance)**.** *Let* $X, Y$ *be i.i.d. random variables with variance* $S = \mathrm{Var}(X)$. *Then their mean squared error (MSE) is double the variance.*

$$
\mathbb{E}\big[(X - Y)^2\big] = 2S.
$$

*Proof.* Expanding and using independence,

$$
\mathbb{E}\big[(X - Y)^2\big] = \mathbb{E}[X^2] + \mathbb{E}[Y^2] - 2\,\mathbb{E}[XY] = 2\,\mathbb{E}[X^2] - 2\,\mathbb{E}[X]\mathbb{E}[Y].
$$

Since $\mathrm{Var}(X) = \mathbb{E}[X^2] - \big(\mathbb{E}[X]\big)^2 = S$, this simplifies to $2S$. $\qquad\square$

## C.4. Integral representation of matrix fractional powers

**Lemma C.4** (Scalar beta integral identity (Balakrishnan, 1960))**.** *We have the integral identity*

$$\int_0^\infty \frac{t^{-\alpha}dt}{\lambda + t} = \frac{\pi}{\sin(\pi\alpha)}\lambda^{-\alpha}, \quad \alpha \in (0,1)$$

*Proof.* Recall the definition of Beta function,

$$B(p,q) = \int_0^1 u^{p-1}(1-u)^{q-1}du = \frac{\Gamma(p)\Gamma(q)}{\Gamma(p+q)}$$

We can turn it into beta function via change of variable $u = \frac{t}{\lambda+t}$, then $t \in [0,\infty)$ maps to $u \in [0,1)$.

$$t = \frac{u\lambda}{1-u}$$

$$dt = \frac{\lambda}{(1-u)^2}du$$

$$\int_0^\infty \frac{t^{-\alpha}dt}{\lambda + t} = \int_0^\infty (\frac{t}{t+\lambda})t^{-1-\alpha}dt$$

$$= \int_0^1 u(\frac{u\lambda}{1-u})^{-1-\alpha}\frac{\lambda}{(1-u)^2}du$$

$$= \lambda^{-\alpha}\int_0^1 u^{-\alpha}(1-u)^{\alpha-1}du$$

$$= \lambda^{-\alpha}B(1-\alpha,\alpha)$$

and using Euler's reflection formula, we have

$$B(1-\alpha,\alpha) = \frac{\pi}{\sin(\pi\alpha)}$$

Thus,

$$\int_0^\infty \frac{t^{-\alpha}dt}{\lambda + t} = \frac{\pi}{\sin(\pi\alpha)}\lambda^{-\alpha}$$

$\square$

**Corollary 1** (Integral formula for power one half)**.** *In the special case of $\alpha = 1/2$*

$$\pi\lambda^{-1/2} = \int_0^\infty \frac{t^{-1/2}dt}{\lambda + t} = 2\int_0^\infty \frac{ds}{\lambda + s^2}$$

*Proof.* Use simple change of variable $t \to s^2$,

$$\pi\lambda^{-1/2} = \int_0^\infty \frac{t^{-1/2}dt}{\lambda + t} = \int_0^\infty \frac{s^{-1}ds^2}{\lambda + s^2} = \int_0^\infty \frac{2ds}{\lambda + s^2}$$

$\square$

**Corollary 2** (Integral representation of fractional matrix power)**.** *The matrix version of such identity, for self-adjoint, positive semi definite matrix $A \succeq 0$,*

$$\int_0^\infty (A + tI)^{-1}t^{-\alpha}dt = \frac{\pi}{\sin(\pi\alpha)}A^{-\alpha}, \quad \alpha \in (0,1)$$

*Similarly, for $z \geq 0, z \in \mathbb{R}$,*

$$\int_0^\infty (A + (z+t)I)^{-1}t^{-\alpha}dt = \frac{\pi}{\sin(\pi\alpha)}(A + zI)^{-\alpha}, \quad \alpha \in (0,1)$$

**Corollary 3** (Integral representation of matrix one half)**.** *The matrix version of such identity, for self-adjoint, positive semi definite matrix $A \succeq 0$,*

$$A^{-1/2} = \frac{1}{\pi}\int_0^\infty (A + tI)^{-1}t^{-1/2}dt = \frac{2}{\pi}\int_0^\infty (A + s^2I)^{-1}ds$$

**Lemma C.5** (Resolvent Identity). *When $u \neq s$, we have the identity*

$$(A + sI)^{-1}(A + uI)^{-1} = \frac{1}{s-u}\left((A + uI)^{-1} - (A + sI)^{-1}\right)$$

$$A(A + sI)^{-1}(A + uI)^{-1} = \frac{1}{s-u}\left(A(A + uI)^{-1} - A(A + sI)^{-1}\right)$$

$$= \frac{s(A + sI)^{-1} - u(A + uI)^{-1}}{s-u}$$

*Proof.* Note that

$$\left((A + sI) - (A + uI)\right)(A + sI)^{-1}(A + uI)^{-1}$$
$$= (A + uI)^{-1} - (A + sI)^{-1}$$
$$= (s - u)(A + sI)^{-1}(A + uI)^{-1}$$

Thus,

$$(A + sI)^{-1}(A + uI)^{-1} = \frac{1}{(s-u)}\left((A + uI)^{-1} - (A + sI)^{-1}\right)$$

as corollary

$$A(A + sI)^{-1}(A + uI)^{-1} = \frac{1}{s-u}\left(A(A + uI)^{-1} - A(A + sI)^{-1}\right)$$

$$= \frac{1}{s-u}\left(I - u(A + uI)^{-1} - I + s(A + sI)^{-1}\right)$$

$$= \frac{s(A + sI)^{-1} - u(A + uI)^{-1}}{s-u}$$

Note that this formula has no real pole, and it behaves well when denominator vanishes, and the RHS becomes a derivative.

$$\lim_{s \to u} \frac{s(A + sI)^{-1} - u(A + uI)^{-1}}{s-u} = \frac{d}{ds}s(A + sI)^{-1}$$

$$= (A + sI)^{-1} - s(A + sI)^{-2}$$

$$= A(A + sI)^{-2}$$

$$\lim_{s \to u} \frac{1}{(s-u)}\left((A + uI)^{-1} - (A + sI)^{-1}\right) = -\frac{d}{du}(A + uI)^{-1}$$

$$= (A + uI)^{-2}$$

$\square$

## C.5. Proof of the sampling-map expectation equivalence: infinite-$\sigma_T$ approximation (Proposition 5.1)

Using empirical covariance and mean to realize the sampling, we have

$$\mathbf{x}(\mathbf{x}_{\sigma_T}, \sigma_0) = \hat{\boldsymbol{\mu}} + (\hat{\boldsymbol{\Sigma}} + \sigma_0^2 I)^{1/2}(\hat{\boldsymbol{\Sigma}} + \sigma_T^2 I)^{-1/2}(\mathbf{x}_{\sigma_T} - \hat{\boldsymbol{\mu}})$$

For the final sampling outcome $\sigma_0 \to 0$, this reads

$$\mathbf{x}(\mathbf{x}_{\sigma_T}, 0) = \hat{\boldsymbol{\mu}} + \hat{\boldsymbol{\Sigma}}^{1/2}(\hat{\boldsymbol{\Sigma}} + \sigma_T^2 I)^{-1/2}(\mathbf{x}_{\sigma_T} - \hat{\boldsymbol{\mu}})$$

As before, assume the sample mean equals the population one, then the finite sample effect comes from the matrix $\hat{\boldsymbol{\Sigma}}^{1/2}(\hat{\boldsymbol{\Sigma}} + \sigma_T^2 I)^{-1/2}$

$$\mathbf{x}(\mathbf{x}_{\sigma_T}, 0) = \boldsymbol{\mu} + \hat{\boldsymbol{\Sigma}}^{1/2}(\hat{\boldsymbol{\Sigma}} + \sigma_T^2 I)^{-1/2}(\mathbf{x}_{\sigma_T} - \boldsymbol{\mu})$$

Note that for sampling, under the EDM convention, the initial noise is sampled with variance $\sigma_T^2 I$, $\mathbf{x}_{\sigma_T} \sim \mathcal{N}(0, \sigma_T^2 I)$, notably for practical diffusion models, initial noise variances are large, $\sigma_T^2 \sim 6000$. Thus we can define a normalized initial noise $\bar{\mathbf{x}} = (\mathbf{x}_{\sigma_T} - \boldsymbol{\mu})/\sigma_T$.

As a large initial noise limit, given that $\boldsymbol{\Sigma}$ has finite spectral norm,

$$\lim_{\sigma \to \infty} \sigma \boldsymbol{\Sigma}^{1/2}(\boldsymbol{\Sigma} + \sigma^2 I)^{-1/2} = \boldsymbol{\Sigma}^{1/2}$$

and when $\sigma_T \to \infty$ the normalized initial noise is sampled from standard Gaussian, $\bar{\mathbf{x}} \sim \mathcal{N}(0, I)$.

Equivalently, we can consider expansion as orders of $1/\sigma$,

$$\sigma \mathbf{\Sigma}^{1/2}(\mathbf{\Sigma} + \sigma^2 I)^{-1/2} = \mathbf{\Sigma}^{1/2}(I + \frac{1}{\sigma^2}\mathbf{\Sigma})^{-1/2}$$

$$\approx \mathbf{\Sigma}^{1/2}(I - \frac{1}{2}\frac{1}{\sigma^2}\mathbf{\Sigma} + ...)$$

$$\approx \mathbf{\Sigma}^{1/2} - \frac{1}{2}\frac{1}{\sigma^2}\mathbf{\Sigma}^{3/2} + ...$$

If we keep the zeroth-order term, then we get the approximation

$$\sigma \mathbf{\Sigma}^{1/2}(\mathbf{\Sigma} + \sigma^2 I)^{-1/2} \approx \mathbf{\Sigma}^{1/2}$$

Consider approximation,

$$\mathbf{x}(\mathbf{x}_{\sigma_T}, 0) = \boldsymbol{\mu} + \hat{\mathbf{\Sigma}}^{1/2}(\hat{\mathbf{\Sigma}} + \sigma_T^2 I)^{-1/2}(\mathbf{x}_{\sigma_T} - \boldsymbol{\mu})$$

$$\approx \boldsymbol{\mu} + \hat{\mathbf{\Sigma}}^{1/2}(\frac{\mathbf{x}_{\sigma_T} - \boldsymbol{\mu}}{\sigma_T})$$

$$= \boldsymbol{\mu} + \hat{\mathbf{\Sigma}}^{1/2}\bar{\mathbf{x}}$$

then we can study the finite-sample effect on the sampling mapping via the matrix $\hat{\mathbf{\Sigma}}^{1/2}$.

**Proposition C.6.** *Deterministic equivalence of empirical covariance matrix one half*

$$\hat{\mathbf{\Sigma}}^{1/2} = \frac{2}{\pi}\int_0^\infty \hat{\mathbf{\Sigma}}(\hat{\mathbf{\Sigma}} + u^2 I)^{-1}du \tag{26}$$

$$\asymp \frac{2}{\pi}\int_0^\infty \mathbf{\Sigma}\Big(\mathbf{\Sigma} + \kappa(u^2)I\Big)^{-1}du \tag{27}$$

*Proof.* Combining Lemma 3 with deterministic equivalence of one point (Eq. DE). $\quad\square$

This result can be compared to population covariance half, when renormalization effect vanish $\kappa(u^2) \to u^2$.

$$\mathbf{\Sigma}^{1/2} = \frac{2}{\pi}\int_0^\infty \mathbf{\Sigma}(\mathbf{\Sigma} + u^2 I)^{-1}du$$

Since $\kappa(u^2) > u^2$ point by point in the integral, the sample version leads to larger shrinkage.

$$\mathbf{v}^\top \hat{\mathbf{\Sigma}}^{1/2}\mathbf{v} < \mathbf{v}^\top \mathbf{\Sigma}^{1/2}\mathbf{v}$$

Concretely, if we measure along spectral modes $\mathbf{u}_k$ of population covariance,

$$\mathbf{u}_k^\top \hat{\mathbf{\Sigma}}^{1/2}\mathbf{u}_k \asymp \frac{2}{\pi}\int_0^\infty \mathbf{u}_k^\top \mathbf{\Sigma}\Big(\mathbf{\Sigma} + \kappa(u^2)I\Big)^{-1}\mathbf{u}_k\, du$$

$$= \frac{2}{\pi}\int_0^\infty \frac{\lambda_k}{\lambda_k + \kappa(u^2)}du$$

$$< \frac{2}{\pi}\int_0^\infty \frac{\lambda_k}{\lambda_k + u^2}du$$

$$= \lambda_k^{1/2}$$

### C.6. Proof of the sampling-map expectation equivalence: finite $\sigma_T$

Next, we consider the finite $\sigma_T$ case, which involves two matrix square roots and their equivalence. To prove this, we proceed in two steps: 1) use an integral identity to represent matrices of the form $A^{1/2}(A + zI)^{-1/2}$, and 2) apply one-point deterministic equivalence.

**Proposition C.7.** *Integral representation, for self-adjoint, positive semidefinite matrix $A \succeq 0$,*

$$A^{1/2}(A + zI)^{-1/2} = \frac{4}{\pi^2} \int_0^\infty \int_0^\infty A(A + u^2 I)^{-1}(A + (z + v^2)I)^{-1} du dv$$

$$= \frac{4}{\pi^2} \int_0^\infty \int_0^\infty \frac{A(A + (z + u^2)I)^{-1} - A(A + v^2 I)^{-1}}{v^2 - u^2 - z} du dv$$

*Proof.* We can study matrix of this form,

$$A^{1/2}(A + zI)^{-1/2}$$

using the integral representation above twice, we have

$$A^{1/2}(A + zI)^{-1/2}$$
$$= AA^{-1/2}(A + zI)^{-1/2}$$
$$= \frac{1}{\pi^2} A \int_0^\infty (A + sI)^{-1} s^{-1/2} ds \int_0^\infty (A + (z + t)I)^{-1} t^{-1/2} dt$$
$$= \frac{1}{\pi^2} \int_0^\infty \int_0^\infty A(A + sI)^{-1}(A + (z + t)I)^{-1} t^{-1/2} s^{-1/2} ds dt$$
$$= \frac{4}{\pi^2} \int_0^\infty \int_0^\infty A(A + u^2 I)^{-1}(A + (z + v^2)I)^{-1} du dv$$

To deal with this *product of resolvents*, we can turn it into a *difference of resolvents* via Lemma C.5,

$$(A + sI)^{-1}(A + tI)^{-1} = \frac{1}{(s - t)}\left((A + tI)^{-1} - (A + sI)^{-1}\right)$$

Now using the identity, we have

$$A^{1/2}(A + zI)^{-1/2}$$
$$= \frac{1}{\pi^2} \int_0^\infty \int_0^\infty A(A + sI)^{-1}(A + (z + t)I)^{-1} t^{-1/2} s^{-1/2} ds dt$$
$$= \frac{1}{\pi^2} \int_0^\infty \int_0^\infty \frac{A(A + (z + t)I)^{-1} - A(A + sI)^{-1}}{s - z - t} t^{-1/2} s^{-1/2} ds dt$$

Putting it together,

$$A^{1/2}(A + zI)^{-1/2} = \frac{1}{\pi^2} \int_0^\infty \int_0^\infty \frac{A(A + (z + t)I)^{-1} - A(A + sI)^{-1}}{s - t - z} t^{-1/2} s^{-1/2} ds dt$$
$$= \frac{4}{\pi^2} \int_0^\infty \int_0^\infty \frac{A(A + (z + u^2)I)^{-1} - A(A + v^2 I)^{-1}}{v^2 - u^2 - z} du dv$$

$\square$

Next we are ready to use the one-point deterministic equivalence.

**Proposition C.8.** *For sample covariance matrix $\hat{\Sigma}$, the following expression has deterministic equivalent to the double integral of population covariance,*

$$\hat{\Sigma}^{1/2}(\hat{\Sigma} + \sigma^2 I)^{-1/2} \asymp \frac{4}{\pi^2} \int_0^\infty \int_0^\infty \frac{\kappa(\sigma^2 + u^2) - \kappa(v^2)}{(\sigma^2 + u^2) - v^2} \Sigma(\Sigma + \kappa(\sigma^2 + u^2)I)^{-1}(\Sigma + \kappa(v^2)I)^{-1} du dv$$

*Proof.* Using Proposition C.7, set $A \to \hat{\boldsymbol{\Sigma}}$. We can then apply deterministic equivalence for resolvents:

$$\hat{\boldsymbol{\Sigma}}^{1/2}(\hat{\boldsymbol{\Sigma}} + \sigma^2 I)^{-1/2} = \hat{\boldsymbol{\Sigma}}\hat{\boldsymbol{\Sigma}}^{-1/2}(\hat{\boldsymbol{\Sigma}} + \sigma^2 I)^{-1/2}$$

$$= \frac{1}{\pi^2}\int_0^\infty \int_0^\infty \hat{\boldsymbol{\Sigma}}(\hat{\boldsymbol{\Sigma}} + sI)^{-1}(\hat{\boldsymbol{\Sigma}} + (\sigma^2 + t)I)^{-1} t^{-1/2} s^{-1/2} ds dt$$

$$= \frac{1}{\pi^2}\int_0^\infty \int_0^\infty \frac{\hat{\boldsymbol{\Sigma}}(\hat{\boldsymbol{\Sigma}} + (\sigma^2 + t)I)^{-1} - \hat{\boldsymbol{\Sigma}}(\hat{\boldsymbol{\Sigma}} + sI)^{-1}}{s - t - \sigma^2} t^{-1/2} s^{-1/2} ds dt$$

$$\asymp \frac{1}{\pi^2}\int_0^\infty \int_0^\infty \frac{\boldsymbol{\Sigma}(\boldsymbol{\Sigma} + \kappa(\sigma^2 + t)I)^{-1} - \boldsymbol{\Sigma}(\boldsymbol{\Sigma} + \kappa(s)I)^{-1}}{s - t - \sigma^2} t^{-1/2} s^{-1/2} ds dt$$

Note there is no pole in this double integral, i.e. when $s = t + \sigma^2$, $\boldsymbol{\Sigma}(\boldsymbol{\Sigma} + \kappa(\sigma^2 + t)I)^{-1} = \boldsymbol{\Sigma}(\boldsymbol{\Sigma} + \kappa(s)I)^{-1}$, thus both numerator and denominator vanish, and the limit is well defined as a derivative!

$$RHS = \frac{1}{\pi^2}\int_0^\infty \int_0^\infty \frac{\boldsymbol{\Sigma}(\boldsymbol{\Sigma} + \kappa(\sigma^2 + t)I)^{-1} - \boldsymbol{\Sigma}(\boldsymbol{\Sigma} + \kappa(s)I)^{-1}}{s - t - \sigma^2} t^{-1/2} s^{-1/2} ds dt$$

$$= \frac{1}{\pi^2}\int_0^\infty \int_0^\infty \frac{(\kappa(s) - \kappa(\sigma^2 + t))\boldsymbol{\Sigma}(\boldsymbol{\Sigma} + \kappa(\sigma^2 + t)I)^{-1}(\boldsymbol{\Sigma} + \kappa(s)I)^{-1}}{s - t - \sigma^2} t^{-1/2} s^{-1/2} ds dt$$

$$= \frac{1}{\pi^2}\int_0^\infty \int_0^\infty \frac{\kappa(s) - \kappa(\sigma^2 + t)}{s - (\sigma^2 + t)} \boldsymbol{\Sigma}(\boldsymbol{\Sigma} + \kappa(\sigma^2 + t)I)^{-1}(\boldsymbol{\Sigma} + \kappa(s)I)^{-1} t^{-1/2} s^{-1/2} ds dt$$

This formulation shows that there are no real poles.

We can remove the singularity at 0 via the change of variables $t \to u^2, s \to v^2$:

$$RHS = \frac{4}{\pi^2}\int_0^\infty \int_0^\infty \frac{\kappa(\sigma^2 + u^2) - \kappa(v^2)}{(\sigma^2 + u^2) - v^2} \boldsymbol{\Sigma}(\boldsymbol{\Sigma} + \kappa(\sigma^2 + u^2)I)^{-1}(\boldsymbol{\Sigma} + \kappa(v^2)I)^{-1} du dv$$

Thus, we obtain the desired equivalence,

$$\hat{\boldsymbol{\Sigma}}^{1/2}(\hat{\boldsymbol{\Sigma}} + \sigma^2 I)^{-1/2} \asymp \frac{4}{\pi^2}\int_0^\infty \int_0^\infty \frac{\kappa(\sigma^2 + u^2) - \kappa(v^2)}{(\sigma^2 + u^2) - v^2} \boldsymbol{\Sigma}(\boldsymbol{\Sigma} + \kappa(\sigma^2 + u^2)I)^{-1}(\boldsymbol{\Sigma} + \kappa(v^2)I)^{-1} du dv$$

Note that the coefficient $\frac{\kappa(\sigma^2+u^2)-\kappa(v^2)}{(\sigma^2+u^2)-v^2}$ behaves well when $(\sigma^2 + u^2) - v^2 \to 0$, i.e. it becomes a derivative of $\kappa$ (Lemma C.5). Thus, there is no singularity in the integrand. $\qquad\square$

**Interpretation** We can compare it to the sampling mapping with the population covariance, i.e. infinite data limit. Using Prop C.7, setting $A \to \boldsymbol{\Sigma}$, the double integral representation of the denoiser mapping reads,

$$\boldsymbol{\Sigma}^{1/2}(\boldsymbol{\Sigma} + \sigma^2 I)^{-1/2} = \frac{1}{\pi^2}\int_0^\infty \int_0^\infty \boldsymbol{\Sigma}\left(\boldsymbol{\Sigma} + (\sigma^2 + t)I\right)^{-1}(\boldsymbol{\Sigma} + sI)^{-1} t^{-1/2} s^{-1/2} ds dt$$

$$= \frac{4}{\pi^2}\int_0^\infty \int_0^\infty \boldsymbol{\Sigma}\left(\boldsymbol{\Sigma} + (\sigma^2 + u^2)I\right)^{-1}(\boldsymbol{\Sigma} + v^2 I)^{-1} du dv$$

Indeed, since $\kappa(\sigma^2 + u^2) > (\sigma^2 + u^2)$ and $\kappa(v^2) > v^2$, this creates a larger shrinkage, especially at small eigen dimensions.

### C.7. Proof of the sampling-map fluctuation equivalence: infinite-$\sigma_T$ approximation (Proposition 5.2)

Now let us consider the variance of the generated outcome with the infinite $\sigma_T$ approximation, ignoring estimation error in $\boldsymbol{\mu}$,

$$\mathbf{x}_{\sigma_0} = \boldsymbol{\mu} + \hat{\boldsymbol{\Sigma}}^{1/2}(\hat{\boldsymbol{\Sigma}} + \sigma_T^2 I)^{-1/2}(\mathbf{x}_{\sigma_T} - \boldsymbol{\mu})$$

$$\approx \boldsymbol{\mu} + \hat{\boldsymbol{\Sigma}}^{1/2}\left(\frac{\mathbf{x}_{\sigma_T} - \boldsymbol{\mu}}{\sigma_T}\right)$$

$$= \boldsymbol{\mu} + \hat{\boldsymbol{\Sigma}}^{1/2}\bar{\mathbf{x}}$$

Thus, the variance of the generated output comes from the sample estimate of the covariance. Let $\bar{\mathbf{x}} := \frac{\mathbf{x}_{\sigma_T} - \boldsymbol{\mu}}{\sigma_T}$, i.e. the normalized deviation from the center.

**Proposition C.9** (Main result, variance of generated sample under empirical data covariance.)**.**

$$Var_{\hat{\boldsymbol{\Sigma}}}[\mathbf{v}^\top\hat{\boldsymbol{\Sigma}}^{1/2}\bar{\mathbf{x}}] \asymp \frac{4}{\pi^2}\int_0^\infty\int_0^\infty\Big\{\frac{\kappa\kappa'}{n-\mathrm{df}_2(\kappa,\kappa')}\big[\mathbf{v}^\top\boldsymbol{\Sigma}(\boldsymbol{\Sigma}+\kappa I)^{-1}(\boldsymbol{\Sigma}+\kappa' I)^{-1}\mathbf{v}\big]$$
$$\times\big[\bar{\mathbf{x}}^\top\boldsymbol{\Sigma}(\boldsymbol{\Sigma}+\kappa I)^{-1}(\boldsymbol{\Sigma}+\kappa' I)^{-1}\bar{\mathbf{x}}\big]\Big\}dudv$$

*where $\kappa := \kappa(u^2), \kappa' := \kappa(v^2)$ are the variables to be integrated over.*

*Proof.* Representing the variance by moments,

$$Var_{\hat{\boldsymbol{\Sigma}}}[\mathbf{v}^\top\hat{\boldsymbol{\Sigma}}^{1/2}\bar{\mathbf{x}}]$$
$$= \mathbb{E}_{\hat{\boldsymbol{\Sigma}}}[(\mathbf{v}^\top\hat{\boldsymbol{\Sigma}}^{1/2}\bar{\mathbf{x}})^2] - \mathbb{E}_{\hat{\boldsymbol{\Sigma}}}[\mathbf{v}^\top\hat{\boldsymbol{\Sigma}}^{1/2}\bar{\mathbf{x}}]^2$$
$$= \mathbb{E}_{\hat{\boldsymbol{\Sigma}}}[\mathbf{v}^\top\hat{\boldsymbol{\Sigma}}^{1/2}\bar{\mathbf{x}}\bar{\mathbf{x}}^\top\hat{\boldsymbol{\Sigma}}^{1/2}\mathbf{v}] - \mathbb{E}_{\hat{\boldsymbol{\Sigma}}}[\mathbf{v}^\top\hat{\boldsymbol{\Sigma}}^{1/2}\bar{\mathbf{x}}]\mathbb{E}_{\hat{\boldsymbol{\Sigma}}}[\bar{\mathbf{x}}^\top\hat{\boldsymbol{\Sigma}}^{1/2}\mathbf{v}] \qquad \textit{using Eq. 26}$$
$$= \mathbb{E}_{\hat{\boldsymbol{\Sigma}}}\Big\{\mathbf{v}^\top\Big[\frac{2}{\pi}\int_0^\infty\hat{\boldsymbol{\Sigma}}(\hat{\boldsymbol{\Sigma}}+u^2I)^{-1}du\Big]\bar{\mathbf{x}}\bar{\mathbf{x}}^\top\Big[\frac{2}{\pi}\int_0^\infty\hat{\boldsymbol{\Sigma}}(\hat{\boldsymbol{\Sigma}}+v^2I)^{-1}dv\Big]\mathbf{v}\Big\}$$
$$- \mathbb{E}_{\hat{\boldsymbol{\Sigma}}}\Big\{\mathbf{v}^\top\Big[\frac{2}{\pi}\int_0^\infty\hat{\boldsymbol{\Sigma}}(\hat{\boldsymbol{\Sigma}}+u^2I)^{-1}du\Big]\bar{\mathbf{x}}\Big\}\mathbb{E}_{\hat{\boldsymbol{\Sigma}}}\Big\{\bar{\mathbf{x}}^\top\Big[\frac{2}{\pi}\int_0^\infty\hat{\boldsymbol{\Sigma}}(\hat{\boldsymbol{\Sigma}}+v^2I)^{-1}dv\Big]\mathbf{v}\Big\}$$

Using the integral representation and exchanging the integral with expectation,

$$RHS = \frac{4}{\pi^2}\int_0^\infty\int_0^\infty\Big\{\mathbb{E}_{\hat{\boldsymbol{\Sigma}}}\Big\{\mathbf{v}^\top\hat{\boldsymbol{\Sigma}}(\hat{\boldsymbol{\Sigma}}+u^2I)^{-1}\bar{\mathbf{x}}\bar{\mathbf{x}}^\top\hat{\boldsymbol{\Sigma}}(\hat{\boldsymbol{\Sigma}}+v^2I)^{-1}\mathbf{v}\Big\}$$
$$- \mathbb{E}_{\hat{\boldsymbol{\Sigma}}}\Big[\mathbf{v}^\top\hat{\boldsymbol{\Sigma}}(\hat{\boldsymbol{\Sigma}}+u^2I)^{-1}\bar{\mathbf{x}}\Big]\mathbb{E}_{\hat{\boldsymbol{\Sigma}}}\Big[\bar{\mathbf{x}}^\top\hat{\boldsymbol{\Sigma}}(\hat{\boldsymbol{\Sigma}}+v^2I)^{-1}\mathbf{v}\Big]\Big\}dudv \qquad \textit{integral representation of matrix half}$$
$$\asymp \frac{4}{\pi^2}\int_0^\infty\int_0^\infty\Big\{\mathbb{E}_{\hat{\boldsymbol{\Sigma}}}\Big\{\mathbf{v}^\top\hat{\boldsymbol{\Sigma}}(\hat{\boldsymbol{\Sigma}}+u^2I)^{-1}\bar{\mathbf{x}}\bar{\mathbf{x}}^\top\hat{\boldsymbol{\Sigma}}(\hat{\boldsymbol{\Sigma}}+v^2I)^{-1}\mathbf{v}\Big\}$$
$$- \Big[\mathbf{v}^\top\boldsymbol{\Sigma}(\boldsymbol{\Sigma}+\kappa(u^2)I)^{-1}\bar{\mathbf{x}}\Big]\Big[\bar{\mathbf{x}}^\top\boldsymbol{\Sigma}(\boldsymbol{\Sigma}+\kappa(v^2)I)^{-1}\mathbf{v}\Big]\Big\}dudv \qquad \textit{using one point equivalence}$$
$$\asymp \frac{4}{\pi^2}\int_0^\infty\int_0^\infty\Big\{\mathrm{Tr}\big[\mathbf{v}\mathbf{v}^\top T_{\boldsymbol{\Sigma}}\bar{\mathbf{x}}\bar{\mathbf{x}}^\top T'_{\boldsymbol{\Sigma}}\big] + \frac{\kappa\kappa'}{n-\mathrm{df}_2(\kappa,\kappa')}\,\mathrm{Tr}\big[\mathbf{v}\mathbf{v}^\top G_{\boldsymbol{\Sigma}}\boldsymbol{\Sigma}G'_{\boldsymbol{\Sigma}}\big]\,\mathrm{Tr}\big[\bar{\mathbf{x}}\bar{\mathbf{x}}^\top G'_{\boldsymbol{\Sigma}}\boldsymbol{\Sigma}G_{\boldsymbol{\Sigma}}\big]$$
$$- \Big[\mathbf{v}^\top\boldsymbol{\Sigma}(\boldsymbol{\Sigma}+\kappa(u^2)I)^{-1}\bar{\mathbf{x}}\Big]\Big[\bar{\mathbf{x}}^\top\boldsymbol{\Sigma}(\boldsymbol{\Sigma}+\kappa(v^2)I)^{-1}\mathbf{v}\Big]\Big\}dudv \qquad \textit{using two point equivalence}$$
$$= \frac{4}{\pi^2}\int_0^\infty\int_0^\infty\Big\{\frac{\kappa\kappa'}{n-\mathrm{df}_2(\kappa,\kappa')}\,\mathrm{Tr}\big[\mathbf{v}\mathbf{v}^\top G_{\boldsymbol{\Sigma}}\boldsymbol{\Sigma}G'_{\boldsymbol{\Sigma}}\big]\,\mathrm{Tr}\big[\bar{\mathbf{x}}\bar{\mathbf{x}}^\top G'_{\boldsymbol{\Sigma}}\boldsymbol{\Sigma}G_{\boldsymbol{\Sigma}}\big]\Big\}dudv \qquad \textit{first trace cancels out.}$$
$$= \frac{4}{\pi^2}\int_0^\infty\int_0^\infty\Big\{\frac{\kappa\kappa'}{n-\mathrm{df}_2(\kappa,\kappa')}\big[\mathbf{v}^\top G_{\boldsymbol{\Sigma}}\boldsymbol{\Sigma}G'_{\boldsymbol{\Sigma}}\mathbf{v}\big]\big[\bar{\mathbf{x}}^\top G'_{\boldsymbol{\Sigma}}\boldsymbol{\Sigma}G_{\boldsymbol{\Sigma}}\bar{\mathbf{x}}\big]\Big\}dudv$$
$$= \frac{4}{\pi^2}\int_0^\infty\int_0^\infty\Big\{\frac{\kappa\kappa'}{n-\mathrm{df}_2(\kappa,\kappa')}\big[\mathbf{v}^\top\boldsymbol{\Sigma}(\boldsymbol{\Sigma}+\kappa(v^2)I)^{-1}(\boldsymbol{\Sigma}+\kappa(u^2)I)^{-1}\mathbf{v}\big]$$
$$\times\big[\bar{\mathbf{x}}^\top\boldsymbol{\Sigma}(\boldsymbol{\Sigma}+\kappa(v^2)I)^{-1}(\boldsymbol{\Sigma}+\kappa(u^2)I)^{-1}\bar{\mathbf{x}}\big]\Big\}dudv$$
$$= \frac{4}{\pi^2}\int_0^\infty\int_0^\infty\Big\{\frac{\kappa\kappa'}{n-\mathrm{df}_2(\kappa,\kappa')}\big[\mathbf{v}^\top\boldsymbol{\Sigma}(\boldsymbol{\Sigma}+\kappa I)^{-1}(\boldsymbol{\Sigma}+\kappa' I)^{-1}\mathbf{v}\big]$$
$$\times\big[\bar{\mathbf{x}}^\top\boldsymbol{\Sigma}(\boldsymbol{\Sigma}+\kappa I)^{-1}(\boldsymbol{\Sigma}+\kappa' I)^{-1}\bar{\mathbf{x}}\big]\Big\}dudv$$

Thus, we arrive at our result

$$Var[\mathbf{v}^\top\hat{\boldsymbol{\Sigma}}^{1/2}\bar{\mathbf{x}}] \asymp \frac{4}{\pi^2}\int_0^\infty\int_0^\infty\Big\{\frac{\kappa\kappa'}{n-\mathrm{df}_2(\kappa,\kappa')}\big[\mathbf{v}^\top\boldsymbol{\Sigma}(\boldsymbol{\Sigma}+\kappa I)^{-1}(\boldsymbol{\Sigma}+\kappa' I)^{-1}\mathbf{v}\big]$$
$$\times\big[\bar{\mathbf{x}}^\top\boldsymbol{\Sigma}(\boldsymbol{\Sigma}+\kappa I)^{-1}(\boldsymbol{\Sigma}+\kappa' I)^{-1}\bar{\mathbf{x}}\big]\Big\}dudv$$

$\square$

### C.7.1. INTERPRETATION

**Anisotropy: effect of the probe vector**    If we marginalize over $\bar{\mathbf{x}}$, assuming $\bar{\mathbf{x}} \sim \mathcal{N}(0, I)$ from white noise, and consider only the effect of probe direction $\mathbf{v}$,

$$
\mathbb{E}_{\bar{\mathbf{x}}} Var[\mathbf{v}^\top \hat{\boldsymbol{\Sigma}}^{1/2} \bar{\mathbf{x}}] \asymp \frac{4}{\pi^2} \int_0^\infty \int_0^\infty \left\{ \frac{\kappa \kappa'}{n - \mathrm{df}_2(\kappa, \kappa')} \left[ \mathbf{v}^\top \boldsymbol{\Sigma}(\boldsymbol{\Sigma} + \kappa I)^{-1}(\boldsymbol{\Sigma} + \kappa' I)^{-1} \mathbf{v} \right] \right.
$$
$$
\left. \times \mathbb{E}_{\bar{\mathbf{x}}} \left[ \bar{\mathbf{x}}^\top \boldsymbol{\Sigma}(\boldsymbol{\Sigma} + \kappa I)^{-1}(\boldsymbol{\Sigma} + \kappa' I)^{-1} \bar{\mathbf{x}} \right] \right\} du dv
$$
$$
\asymp \frac{4}{\pi^2} \int_0^\infty \int_0^\infty \left\{ \frac{\kappa \kappa' \, \mathrm{Tr}[\boldsymbol{\Sigma}(\boldsymbol{\Sigma} + \kappa I)^{-1}(\boldsymbol{\Sigma} + \kappa' I)^{-1}]}{n - \mathrm{df}_2(\kappa, \kappa')} \left[ \mathbf{v}^\top \boldsymbol{\Sigma}(\boldsymbol{\Sigma} + \kappa I)^{-1}(\boldsymbol{\Sigma} + \kappa' I)^{-1} \mathbf{v} \right] \right\} du dv
$$

The mixed resolvent term can be simplified,

$$
\mathrm{Tr}[\boldsymbol{\Sigma}(\boldsymbol{\Sigma} + \kappa I)^{-1}(\boldsymbol{\Sigma} + \kappa' I)^{-1}] = \frac{1}{\kappa} \mathrm{Tr}[(\boldsymbol{\Sigma} + \kappa I - \boldsymbol{\Sigma})\boldsymbol{\Sigma}(\boldsymbol{\Sigma} + \kappa I)^{-1}(\boldsymbol{\Sigma} + \kappa' I)^{-1}]
$$
$$
= \frac{1}{\kappa} \mathrm{Tr}[\boldsymbol{\Sigma}(\boldsymbol{\Sigma} + \kappa' I)^{-1}] - \frac{1}{\kappa} \mathrm{Tr}[\boldsymbol{\Sigma}^2(\boldsymbol{\Sigma} + \kappa I)^{-1}(\boldsymbol{\Sigma} + \kappa' I)^{-1}]
$$
$$
= \frac{1}{\kappa} \mathrm{df}_1(\kappa') - \frac{1}{\kappa} \mathrm{df}_2(\kappa, \kappa')
$$
$$
= \frac{1}{\kappa'} \mathrm{df}_1(\kappa) - \frac{1}{\kappa'} \mathrm{df}_2(\kappa, \kappa')
$$

Using this identity

$$
\mathbb{E}_{\bar{\mathbf{x}}} Var[\mathbf{v}^\top \hat{\boldsymbol{\Sigma}}^{1/2} \bar{\mathbf{x}}] \asymp \frac{4}{\pi^2} \int_0^\infty \int_0^\infty \left\{ \frac{\kappa' \big( \mathrm{df}_1(\kappa') - \mathrm{df}_2(\kappa, \kappa') \big)}{n - \mathrm{df}_2(\kappa, \kappa')} \left[ \mathbf{v}^\top \boldsymbol{\Sigma}(\boldsymbol{\Sigma} + \kappa I)^{-1}(\boldsymbol{\Sigma} + \kappa' I)^{-1} \mathbf{v} \right] \right\} du dv \tag{28}
$$

Setting the direction $\mathbf{v}$ to the eigenvector $\mathbf{u}_k$, with corresponding eigenvalue $\lambda_k$, the variance along the eigen direction reads,

$$
\mathbb{E}_{\bar{\mathbf{x}}} Var[\mathbf{u}_k^\top \hat{\boldsymbol{\Sigma}}^{1/2} \bar{\mathbf{x}}] \asymp \frac{4}{\pi^2} \int_0^\infty \int_0^\infty \left\{ \frac{\kappa' \big( \mathrm{df}_1(\kappa') - \mathrm{df}_2(\kappa, \kappa') \big)}{n - \mathrm{df}_2(\kappa, \kappa')} \frac{\lambda_k}{(\lambda_k + \kappa)(\lambda_k + \kappa')} \right\} du dv \tag{29}
$$

**Inhomogeneity: effect of initial noise**    Since the variance is symmetric in $\bar{\mathbf{x}}$ and $\mathbf{v}$, we can also marginalize over $\mathbf{v}$ while keeping the $\bar{\mathbf{x}}$ dependency. Note that we assume $\mathbf{v}$ is unit norm, so summation over $\mathbf{u}_k$ eigenvectors (instead of expectation) is equivalent to trace.

$$
\sum_k Var[\mathbf{u}_k^\top \hat{\Sigma}^{1/2} \bar{\mathbf{x}}] = \mathrm{Tr}\, Var[\hat{\Sigma}^{1/2} \bar{\mathbf{x}}] \asymp \frac{4}{\pi^2} \int_0^\infty \int_0^\infty \left\{ \frac{\kappa' \big( \mathrm{df}_1(\kappa') - \mathrm{df}_2(\kappa, \kappa') \big)}{n - \mathrm{df}_2(\kappa, \kappa')} \left[ \bar{\mathbf{x}}^\top \Sigma(\Sigma + \kappa I)^{-1}(\Sigma + \kappa' I)^{-1} \bar{\mathbf{x}} \right] \right\} du dv
$$
$$
\tag{30}
$$

**Scaling: effect of sample size**    Finally, marginalizing over both factors, we have the overall scaling.

$$
\mathbb{E}_{\bar{\mathbf{x}}} \sum_k Var[\mathbf{u}_k^\top \hat{\Sigma}^{1/2} \bar{\mathbf{x}}] = \mathbb{E}_{\bar{\mathbf{x}}} \mathrm{Tr}\, Var[\hat{\Sigma}^{1/2} \bar{\mathbf{x}}] \asymp \frac{4}{\pi^2} \int_0^\infty \int_0^\infty \left\{ \frac{\big( \mathrm{df}_1(\kappa') - \mathrm{df}_2(\kappa, \kappa') \big)\big( \mathrm{df}_1(\kappa) - \mathrm{df}_2(\kappa, \kappa') \big)}{n - \mathrm{df}_2(\kappa, \kappa')} \right\} du dv \tag{31}
$$

## D. Experimental Details

### D.1. Numerical methods

**Numerical evaluation of renormalized Ridge $\kappa(z)$.** We computed $\kappa(z)$ as the solution to the self-consistent Silverstein equation

$$\kappa(z) - z \;=\; \gamma \sum_{k=1}^{p} w_k \, \frac{\kappa(z)\,\lambda_k}{\kappa(z) + \lambda_k}, \tag{32}$$

where $\{\lambda_k\}$ are the eigenvalues of $\boldsymbol{\Sigma}$ and $\{w_k\}$ are their normalized weights. For scalar $z$, we solved this nonlinear equation using Newton's method with analytical derivative. Using implicit derivative with respect to $\kappa$, we have

$$1 - \frac{dz}{d\kappa} = \gamma \sum_{k=1}^{p} w_k \, \frac{d}{d\kappa} \left[ \frac{\kappa(z)\,\lambda_k}{\kappa(z) + \lambda_k} \right]$$

$$\left( \kappa'(z) \right)^{-1} \;=\; 1 \;-\; \gamma \sum_{k=1}^{p} w_k \, \frac{\lambda_k^2}{\left( \kappa(z) + \lambda_k \right)^2},$$

This falls back to a robust root-finder for purely real inputs. For a sequence of $z$ values along a path, we used an "analytic continuation" procedure in which the solution at the previous $z$ served as the initial guess for the next, ensuring branch continuity and numerical stability, particularly for small $z$. Further, we generally start the path from $z$ with high norm and solve with continuation back to small $z$. A caching mechanism stored previously computed $(z, \kappa)$ pairs, with nearest-neighbor retrieval for initial guesses, further accelerating repeated evaluations. This approach yields accurate and smooth $\kappa(z)$ profiles suitable for downstream quadrature-based integration.

**Numerical evaluation of the integral over deterministic equivalence** The analytical results in Eqs. 9,10, which involve integrals to infinity, are nontrivial to evaluate. To avoid truncation error, we used the following scheme, which maps the integration variable onto a finite domain.

We approximated the double integral

$$\frac{4}{\pi^2} \int_0^\infty \int_0^\infty \frac{\kappa\,\kappa'\,\mathrm{Tr}\left[ \boldsymbol{\Sigma}(\boldsymbol{\Sigma} + \kappa I)^{-1}(\boldsymbol{\Sigma} + \kappa' I)^{-1} \right]}{n - \mathrm{df}_2(\kappa, \kappa')} \left[ \mathbf{v}^\top \boldsymbol{\Sigma}(\boldsymbol{\Sigma} + \kappa I)^{-1}(\boldsymbol{\Sigma} + \kappa' I)^{-1} \mathbf{v} \right] du\, dv \tag{33}$$

using a Gauss–Legendre quadrature scheme combined with the tangent mapping $u = \tan\theta$ to transform the semi-infinite domain $[0, \infty)$ to a finite interval $[0, \pi/2]$.

We first generated $n_{\mathrm{nodes}}$ Gauss–Legendre nodes $\theta_i$ and weights $w_i$ on $[0, \pi/2]$, then applied the transformation $u = \tan\theta$ with Jacobian $J(\theta) = 1/\cos^2\theta$ to obtain quadrature points on $[0, \infty)$. This was performed independently for $u$ and $v$, and their 2D tensor product provided the integration grid.

The $\kappa$ values were computed at each $u^2$ and $v^2$ using a numerically stable, vectorized evaluation of the spectral mapping function $\kappa(z)$ derived from the eigenspectrum of $\boldsymbol{\Sigma}$. The integrand was then assembled by evaluating the trace term, the scalar bilinear form $\mathbf{v}^\top(\cdot)\mathbf{v}$, and the denominator $n - \mathrm{df}_2(\kappa, \kappa')$ on the full 2D grid. Quadrature weights and Jacobians were applied multiplicatively, and the sum over all grid points yielded the numerical approximation to the integral.

A similar quadrature is used for the single-integral equivalence Eq. 9, where we integrate over a 1D grid.

This approach yields high accuracy while avoiding explicit truncation of the infinite domain, as the nonlinear mapping concentrates quadrature nodes where the integrand varies most rapidly.

### D.2. Linear denoiser experiments

To cross-validate against our theory and numerical scheme, we performed extensive validation using empirical linear denoisers.

We computed the empirical covariance of a dataset and then used the following functions implementing the linear one-step denoiser and the full sampling map (Wiener filter).

```python
def dnoised_X(x, Xmean, sample_cov, sigma2,):
    # single step denoiser
    return x + sigma2 * (Xmean - x ) @ torch.inverse(sample_cov + torch.eye(sample_cov.shape[0],
        device=x.device) * sigma2)

def wiener_gen_X(x, Xmean, wiener_matrix, sigmaT,):
    if x.dim() == 1:
        # Single vector case
        return Xmean + wiener_matrix @ (x * sigmaT - Xmean)
    else:
        # Batched vector case - x should be shape (batch_size, ndim)
        return Xmean[None,:] + (x * sigmaT - Xmean[None,:]) @ wiener_matrix.T

def build_wiener_matrix(eigvals, eigvecs, sigmaT=80.0, sigma0=0.0, EPS=1E-16, clip=True):
    if clip:
        eigvals = torch.clamp(eigvals, min=EPS)
    scaling = ((eigvals + sigma0**2) / (eigvals + sigmaT**2)).sqrt()
    return eigvecs @ torch.diag(scaling) @ eigvecs.T
```

We set $\sigma_0 = 0$ in theory. In practice, it is usually set to a small positive number, e.g., 0.002. We tested this in a few cases and report the results in the appendix. Generally, a positive $\sigma_0$ acts as a floor for generated variance, thereby reducing the overshrinking effect.

We found that when the dataset size is insufficient, e.g., when $\hat{\Sigma}$ is rank deficient, the eigendecomposition can be unstable and sometimes produces negative eigenvalues, which affects the matrix square-root operation in the Wiener matrix. Even if we clip them, numerical artifacts often remain in small eigenspaces. As a remedy, using higher precision `float64` numbers yields results that closely match the theory.

### D.3. Deep neural network experiments

We used the following preconditioning scheme, inspired by (Karras et al., 2022), for all architectures in our comparison.

```python
class EDMPrecondWrapper(nn.Module):
    def __init__(self, model, sigma_data=0.5, sigma_min=0.002, sigma_max=80, rho=7.0):
        super().__init__()
        self.model = model
        self.sigma_data = sigma_data
        self.sigma_min = sigma_min
        self.sigma_max = sigma_max
        self.rho = rho

    def forward(self, X, sigma, cond=None, ):
        sigma[sigma == 0] = self.sigma_min
        ## edm preconditioning for input and output
        ## https://github.com/NVlabs/edm/blob/main/training/networks.py#L632
        # unsqueeze sigma to have same dimension as X (which may have 2-4 dim)
        sigma_vec = sigma.view([-1, ] + [1, ] * (X.ndim - 1))
        c_skip = self.sigma_data ** 2 / (sigma_vec ** 2 + self.sigma_data ** 2)
        c_out = sigma_vec * self.sigma_data / (sigma_vec ** 2 + self.sigma_data ** 2).sqrt()
        c_in = 1 / (self.sigma_data ** 2 + sigma_vec ** 2).sqrt()
        c_noise = sigma.log() / 4
        model_out = self.model(c_in * X, c_noise, cond=cond)
        return c_skip * X + c_out * model_out
```

**EDM Loss Function**  We employ the loss function $\mathcal{L}_{\text{EDM}}$ introduced in the Elucidated Diffusion Model (EDM) paper (Karras et al., 2022), which is one specific weighting scheme for training diffusion models with $x_0$ prediction with

variance-exploding noise schedule.

For each data point $\mathbf{x} \in \mathbb{R}^d$, the loss is computed as follows. The noise level for each data point is sampled from a log-normal distribution with hyperparameters $P_{\text{mean}}$ and $P_{\text{std}}$ (e.g., $P_{\text{mean}} = -1.2$ and $P_{\text{std}} = 1.2$). Specifically, the noise level $\sigma$ is sampled via

$$\sigma = \exp\left(P_{\text{mean}} + P_{\text{std}}\,\epsilon\right), \quad \epsilon \sim \mathcal{N}(0, 1).$$

The weighting function per noise scale is defined as:

$$w(\sigma) = \frac{\sigma^2 + \sigma_{\text{data}}^2}{\left(\sigma\,\sigma_{\text{data}}\right)^2},$$

with hyperparameter $\sigma_{\text{data}}$ (e.g., $\sigma_{\text{data}} = 0.5$). The noisy input $\mathbf{y}$ is created by the following,

$$\mathbf{y} = \mathbf{x} + \sigma\mathbf{n}, \quad \mathbf{n} \sim \mathcal{N}\left(\mathbf{0}, \mathbf{I}_d\right),$$

Let $\mathbf{D}_\theta(\mathbf{y}, \sigma, \text{labels})$ denote the output of the denoising network when given the noisy input $\mathbf{y}$, the noise level $\sigma$, and optional conditioning labels. The EDM loss per data point can be computed as:

$$\mathcal{L}(\mathbf{x}) = w(\sigma)\left\|\mathbf{D}_\theta(\mathbf{x} + \sigma\mathbf{n}, \sigma, \text{labels}) - \mathbf{x}\right\|^2.$$

Taking expectation over the data points and noise scales, the overall loss reads

$$\mathcal{L}_{EDM} = \mathbb{E}_{\mathbf{x}\sim p_{data}}\mathbb{E}_{\mathbf{n}\sim\mathcal{N}(0,\mathbf{I}_d)}\mathbb{E}_\sigma\left[w(\sigma)\left\|\mathbf{D}_\theta(\mathbf{x} + \sigma\mathbf{n}, \sigma, \text{labels}) - \mathbf{x}\right\|^2\right] \tag{34}$$

**Hyperparameter Settings: DiT**  All experiments use DiT backbones with consistent architectural and optimization settings unless otherwise specified. Key hyperparameters:

- **Model architecture:** patch size 2 or 4 (used once for FFHQ64, discarded for worse performance), hidden size 384, depth 6 layers, 6 attention heads, MLP ratio 4.

- **Datasets:** FFHQ-32, AFHQ-32, CIFAR-32, and FFHQ-64; subsampled at varying sizes (300, 1k, 3k, 10k, 30k) with two non-overlapping splits per size.

- **Training objective:** Denoising Score Matching (DSM) under EDM parametrization.

- **Training schedule:** 50000 steps with batch size 256, Adam optimizer with learning rate $1 \times 10^{-4}$.

- **Evaluation:** fixed-noise seed, sampling with 35 steps with Heun sampler; evaluation sample size 1000, batch size 512.

**Hyperparameter Settings: UNet**  All CNN-UNet experiments follow consistent architectural and optimization settings unless noted. Key hyperparameters:

- **Model architecture:** UNet with base channels $128$ ; channel multipliers $\{1, 2, 2, 2\}$; self-attention at resolution 8.

- **Datasets:** FFHQ-32/64, AFHQ-32, CIFAR10-32, CIFAR100-32, LSUN-church-32/64, LSUN-bedroom-32/64; subsampled at varying sizes (300, 1k, 3k, 10k, 30k) with two non-overlapping splits per size.

- **Training objective:** Denoising Score Matching (DSM) under EDM parametrization.

- **Training schedule:** 50000 steps, batch size 256, Adam with learning rate $1 \times 10^{-4}$.

- **Evaluation:** fixed-noise seed, sampling with 35 steps with Heun sampler; evaluation sample size 1000, batch size 512.

**Computation Cost**  All experiments were conducted on NVIDIA A100 or H100 GPUs. Training DiT and CNN models on $32 \times 32$ resolution datasets typically required 5–8 hours to complete. In contrast, DiT models trained on FFHQ64 were substantially more expensive, taking approximately 24 hours per run.

## E. Usage of LLMs

We used LLMs in three ways. First, as a research assistant, to look up tools related to deterministic equivalence and to point us toward integral identities for fractional matrix powers, which we then verified and derived independently. Second, as a coding agent to help us generate plotting and analysis code for our results. Third, as a writing aid, for polishing technical text and providing feedback on clarity and presentation of the whole paper.

