# A Random Matrix Theory Perspective on the Consistency of Diffusion Models

## Abstract

Diffusion models trained on different, non-overlapping subsets of a dataset often produce strikingly similar outputs when given the same noise seed. We trace this consistency to a simple linear effect: the shared Gaussian statistics across splits already predict much of the generated images. To formalize this, we develop a random matrix theory (RMT) framework that quantifies how finite datasets shape the expectation and variance of the learned denoiser and sampling map in the linear setting. For expectations, sampling variability acts as a renormalization of the noise level through a self-consistent relation $\sigma^2 \mapsto \kappa(\sigma^2)$, explaining why limited data overshrink low-variance directions and pull samples toward the dataset mean. For fluctuations, our variance formulas reveal three key factors behind cross-split disagreement: *anisotropy* across eigenmodes, *inhomogeneity* across inputs, and overall scaling with dataset size. Extending deterministic-equivalence tools to fractional matrix powers further allows us to analyze entire sampling trajectories. The theory sharply predicts the behavior of linear diffusion models, and we validate its predictions on UNet and DiT architectures in their non-memorization regime, identifying where and how samples deviates across training data split. This provides a principled baseline for reproducibility in diffusion training, linking spectral properties of data to the stability of generative outputs.

## 1. Introduction

Diffusion models and their relatives such as flow matching have become the dominant generative modeling paradigm across diverse domains, including images, video, and proteins. By learning a time-dependent vector field, these models transform Gaussian noise into structured samples through an ordinary differential equation (ODE) or its stochastic variants (Song et al., 2021; Albergo et al., 2023).

A distinctive feature of diffusion models is their striking *consistency across training runs* (Figure 1). When trained on the same distribution, even with disjoint datasets, different architectures, or repeated initializations, diffusion models often map the same noise seed to highly similar outputs under the deterministic probability flow (Kadkhodaie et al., 2024; Zhang et al., 2024). This phenomenon contrasts with other generative modeling frameworks including GANs and VAEs, where the isotropic Gaussian latent space admits arbitrary rotations, leading to run-to-run variability in the mapping from latent codes to data (Martinez & Pearson, 2022).

**Why does consistency matter?** Consistency across non-overlapping data splits suggests that diffusion models recover aspects of the underlying *data manifold* that are insensitive to the specific training set. This raises fundamental questions about how such models generalize beyond their training samples, to what extent they memorize idiosyncratic data, and whether their outputs reflect universal statistical regularities of the distribution. These issues connect to emerging theoretical and empirical debates on generalization, memorization, and creativity in diffusion models (Kamb & Ganguli, 2024; Niedoba et al., 2024; Kadkhodaie et al., 2024; Chen, 2025; Vastola, 2025; Bonnaire et al., 2025); see also further discussion in App. A.

**Our approach.** We analyze this phenomenon through the lens of random matrix theory (RMT), beginning with the observation that the consistency effect can already be predicted by a linear Gaussian model (Fig. 1). Building on the linear denoiser framework, we develop a precise RMT analysis of how finite-sample variability in the empirical covariance affects both the expectation and fluctuation of denoisers and sampling maps (Fig. 2**A**). We then validate these theoretical predictions against deep diffusion models (CNNs and DiTs), showing that the same RMT principles still govern their inhomogeneity of consistency across data splits. Our **main contributions** are as follows:

- **Linear origin of consistency:** We show that shared Gaussian statistics i.e. linear denoiser already predict

[1]Anonymous Institution, Anonymous City, Anonymous Region, Anonymous Country. Correspondence to: Anonymous Author <anon.email@domain.com>.

Preliminary work. Under review by the International Conference on Machine Learning (ICML). Do not distribute.

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

{\circleddash(\mathbf{v}; \kappa, \kappa', \boldsymbol{\Sigma})}_{anisotropy}\underbrace{\circleddash(\bar{\mathbf{x}}; \kappa, \kappa', \boldsymbol{\Sigma})}_{inhomogeneity}\,du\,dv$$

*where $\circleddash(\mathbf{a}; \kappa, \kappa', \boldsymbol{\Sigma}) := \mathbf{a}^\top\boldsymbol{\Sigma}\,(\boldsymbol{\Sigma} + \kappa I)^{-1}(\boldsymbol{\Sigma} + \kappa' I)^{-1}\,\mathbf{a}$, and $\kappa := \kappa(u^2), \kappa' := \kappa(v^2)$ are variables to be integrated over. Proof in App. C.7.*

**Interpretation** The variance of sampling map Eq. 9 simplifies to a double integral of the denoiser-variance (Eq. 6). The integrand factorizes into a direction-dependent term (*anisotropy*), a initial noise-dependent term (*inhomogeneity*), and a scaling term. Note the anisotropy and inhomogeneity factors rely on the same $\circleddash(.; \kappa, \kappa', \boldsymbol{\Sigma})$ function, showing that dependency on $\mathbf{v}$ and $\bar{\mathbf{x}}$ has the same spectral structure.

We resort to numerical simulation to provide more intuition. We note that integrals in Eqs. 8,9 are nontrivial to evaluate; we describe our numerical scheme in App. D.1. Using this procedure, the theoretical predictions align closely with direct computations of linear diffusion (Fig. 4). **Inhomogeneity.** Spatially, when initial noise $\bar{\mathbf{x}}$ deviates more along the top eigenspace of $\boldsymbol{\Sigma}$, there will be larger uncertainty (Fig. 4C), this enables us to predict the sample difference point by point. **Anisotropy.** Directionally, the dependency on $\mathbf{v}$ has the same structure, in absolute term, the deviation is larger at higher eigenspace (Fig. 4B). Note that when scaling up the dataset size, the variance in the top eigenspace decay immediately from small sample size; while the deviation in lower eigenspace will stay put and start decaying only later at larger dataset size (Fig. 4D). This shows that the fine detail of the samples needs a larger dataset size to be consistency across training.

## 6. Validating Predictions on Deep Networks

Finally, given that linear diffusion behavior is well captured by our random matrix theory (RMT), we test the applicability of its prediction to practical deep diffusion networks.

**Setup.** We trained UNet- and DiT-based denoisers under the EDM framework on FFHQ64, FFHQ32, AFHQ32 (Choi et al., 2020), LSUN church and bedroom at 32 and 64 pixels (Yu et al., 2015), CIFAR10, and CIFAR100 (UNet on all;

duces this finite sample bias, *i.e.*, $\boldsymbol{\Sigma} = \mathbb{E}[\hat{\boldsymbol{\Sigma}}] = \mathbb{E}[\hat{\boldsymbol{\Sigma}}^{1/2}\hat{\boldsymbol{\Sigma}}^{1/2}] \neq (\mathbb{E}[\hat{\boldsymbol{\Sigma}}^{1/2}])^2$.

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

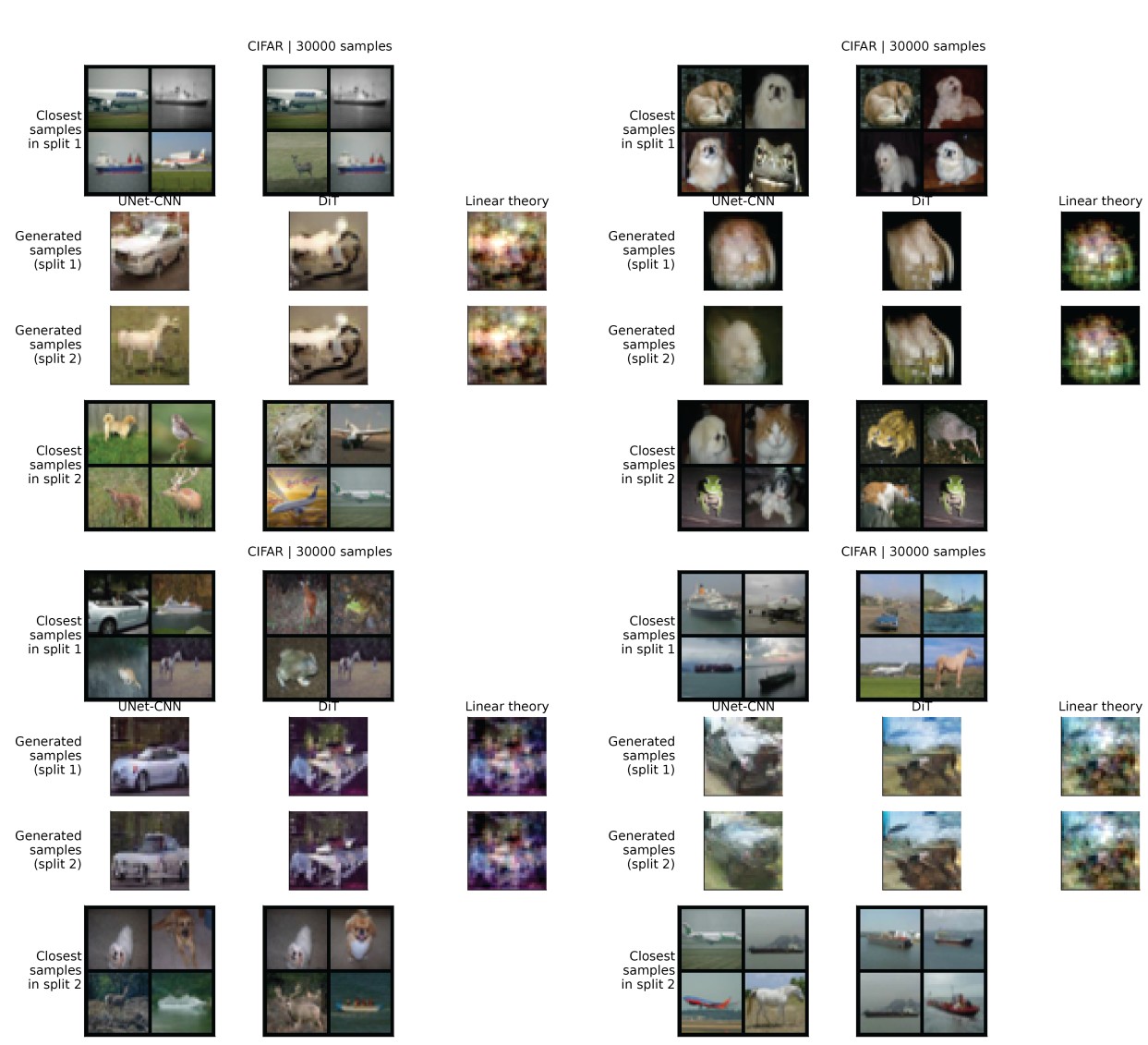

*Figure 13.* **Extended visual comparison of generation consistency and the linear theory for CIFAR10 dataset**. Similar format to Fig. 1A. Generated samples from DNN and linear theory from initial noise seed 0,1,3,4

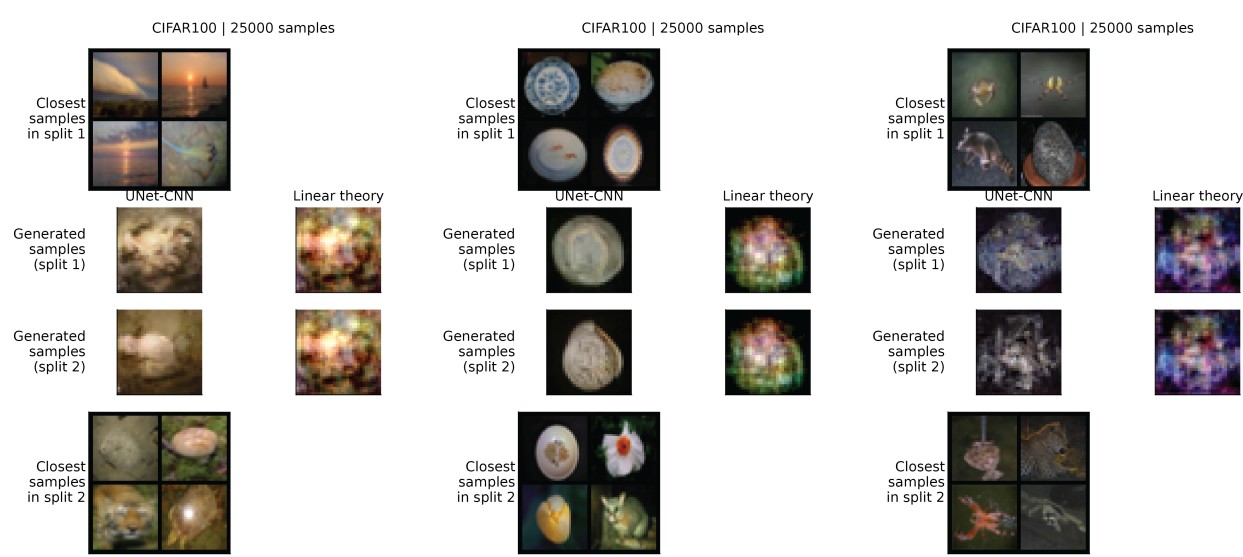

*Figure 14.* **Extended visual comparison of generation consistency and the linear theory for CIFAR100 dataset**. Similar format to Fig. 1A. Generated samples from DNN and linear theory from initial noise seed 0,1,3

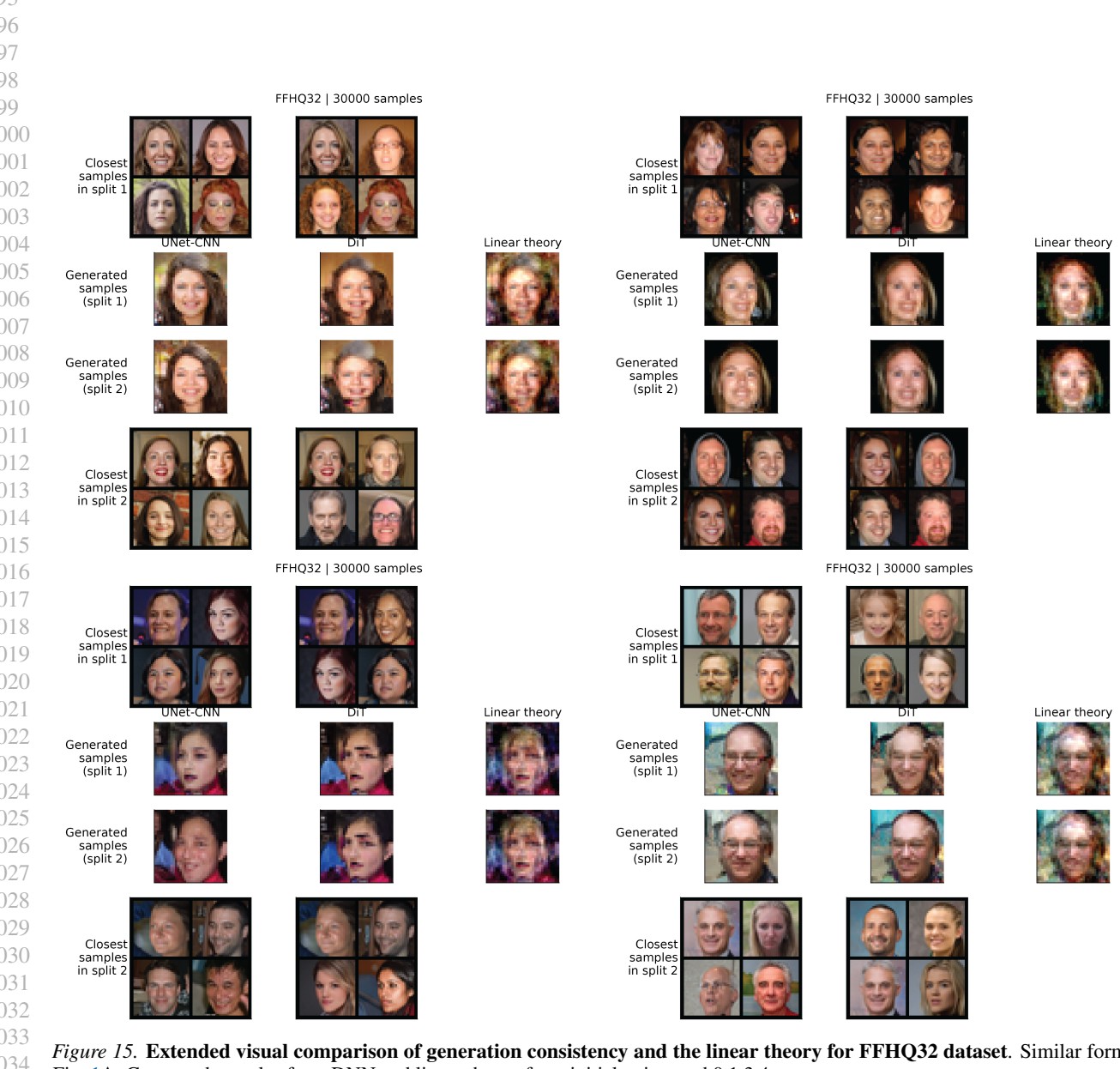

*Figure 15.* **Extended visual comparison of generation consistency and the linear theory for FFHQ32 dataset**. Similar format to Fig. 1**A**.

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

B.4.3. GENERATED SAMPLES FROM DNN APPROACHES GAUSSIAN PREDICTOR AT HIGHER DATASET SIZES

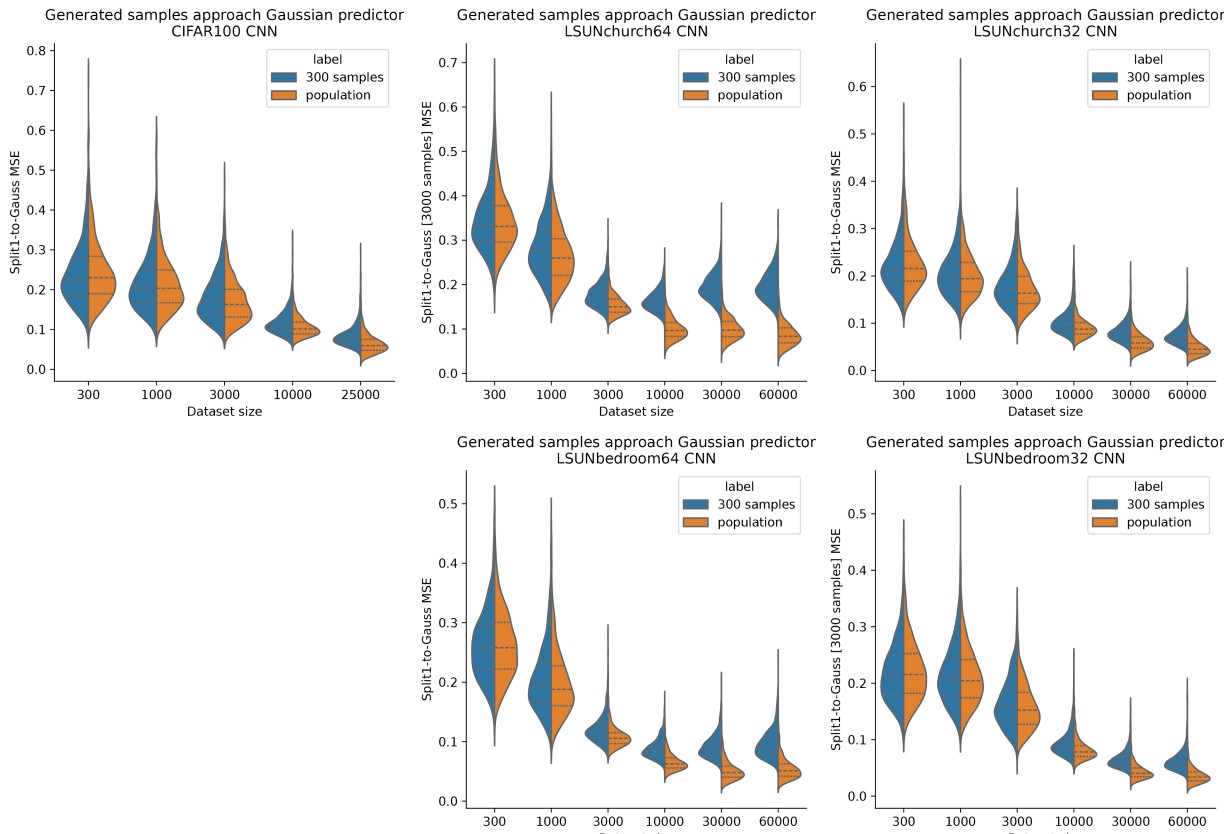

*Figure 34.* **DNN generated samples approach the linear theory predictor (with finite sample or population covariance).** With increasing dataset size $n$, the generated sample from DNN (trained on split 1) with a fixed noise seed gradually approach the linear theory predictor using the same initial seed. The left violin plot shows the linear theory predictor using empirical mean and covariance $\hat{\mu}, \hat{\Sigma}$ computed from only 300 samples; right violin plot shows the predictor using the population mean and covariance (whole dataset). The result is consistent across datasets.

### B.4.4. VARIANCE AND DEVIATION OF GENERATION ALONG THE SPECTRUM

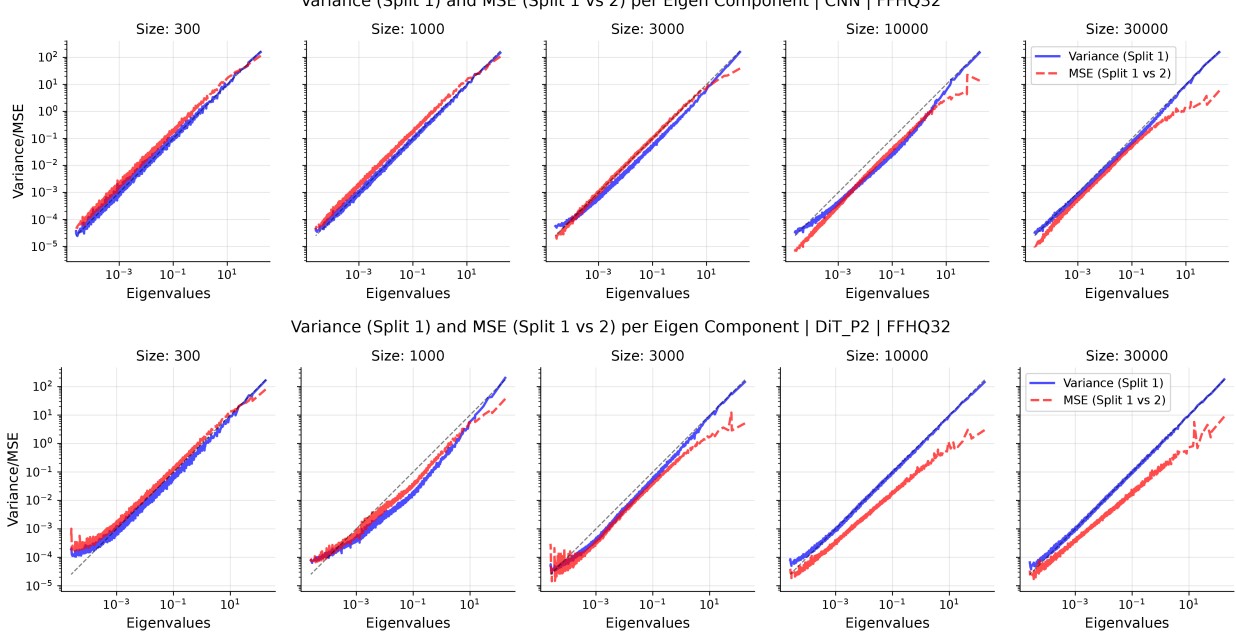

*Figure 35.* **DNN validation experiments, Anisotropy and overshrinking (AFHQ32)**

*Figure 36.* **DNN validation experiments, Anisotropy and overshrinking (FFHQ32)**

### B.4.5. RMT PREDICTS SPATIAL INHOMOGENEITY ACROSS RANDOM NOISE SEEDS

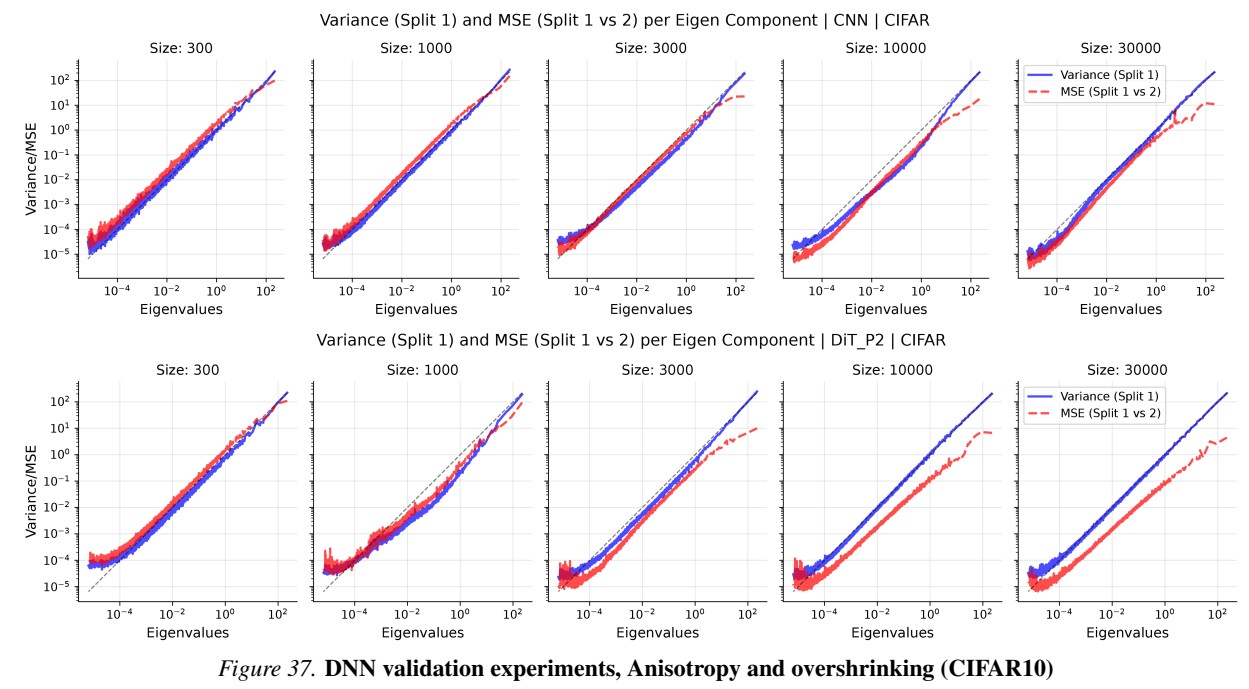

*Figure 37.* **DNN validation experiments, Anisotropy and overshrinking (CIFAR10)**

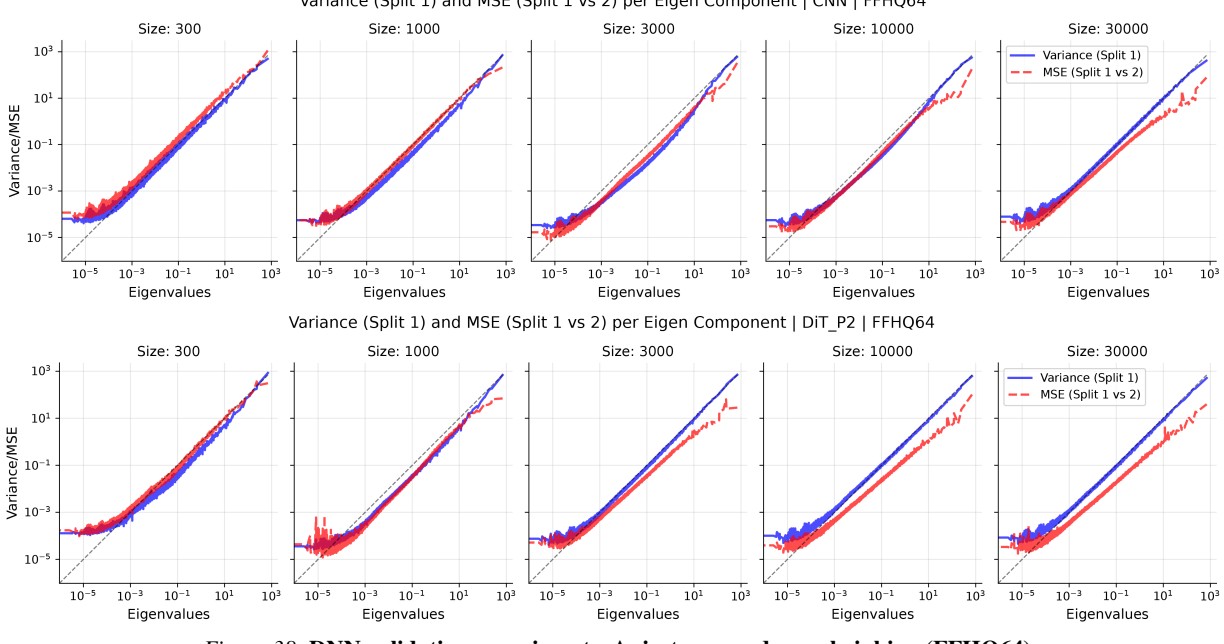

*Figure 38.* **DNN validation experiments, Anisotropy and overshrinking (FFHQ64)**

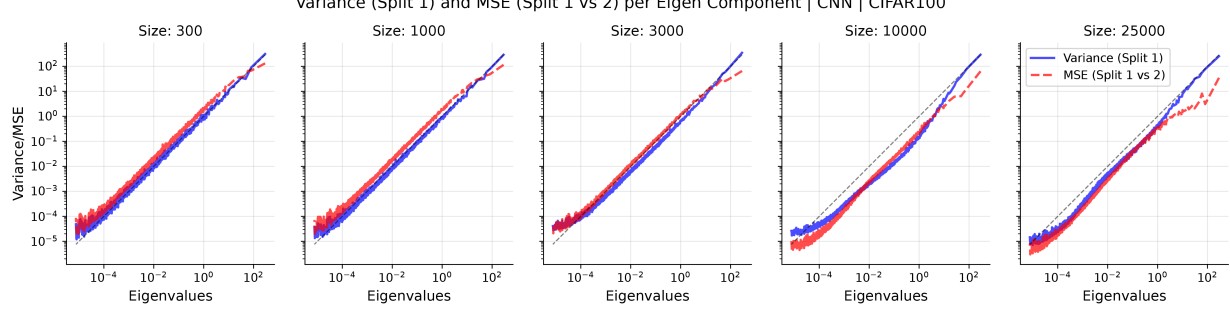

*Figure 39.* **DNN validation experiments, Anisotropy and overshrinking (CIFAR100)**

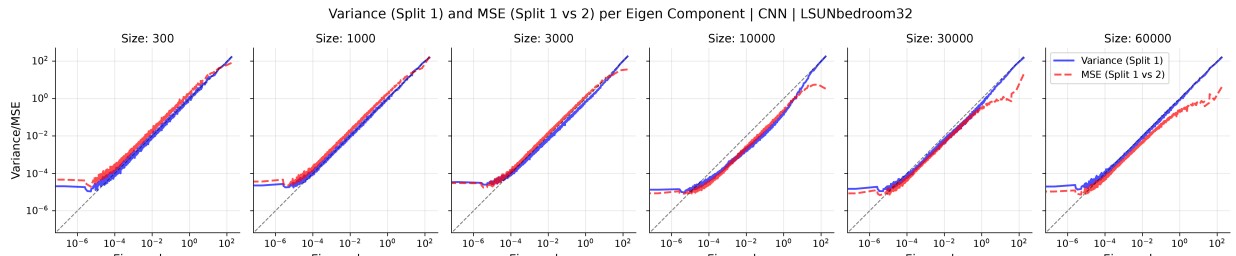

*Figure 40.* **DNN validation experiments, Anisotropy and overshrinking (LSUN bedroom 32)**

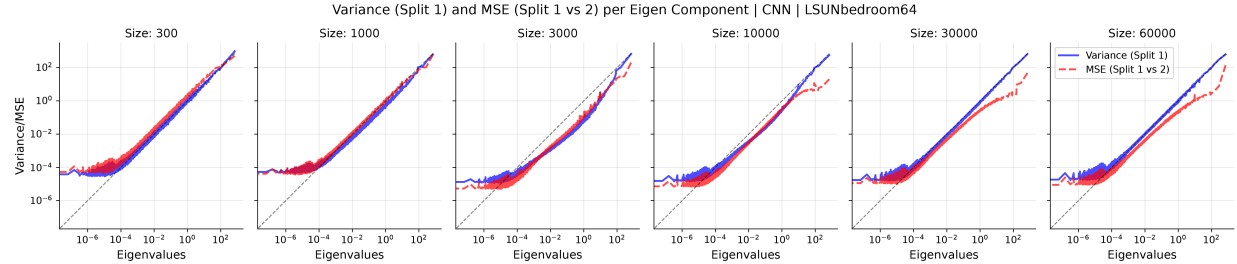

*Figure 41.* **DNN validation experiments, Anisotropy and overshrinking (LSUN bedroom 64)**

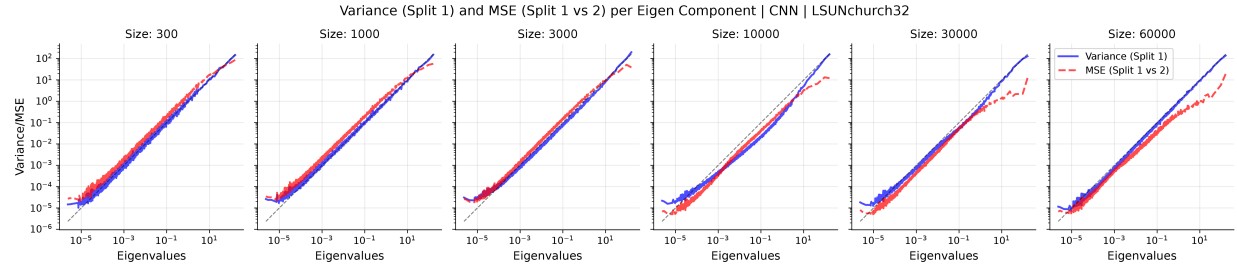

*Figure 42.* **DNN validation experiments, Anisotropy and overshrinking (LSUN church 32)**

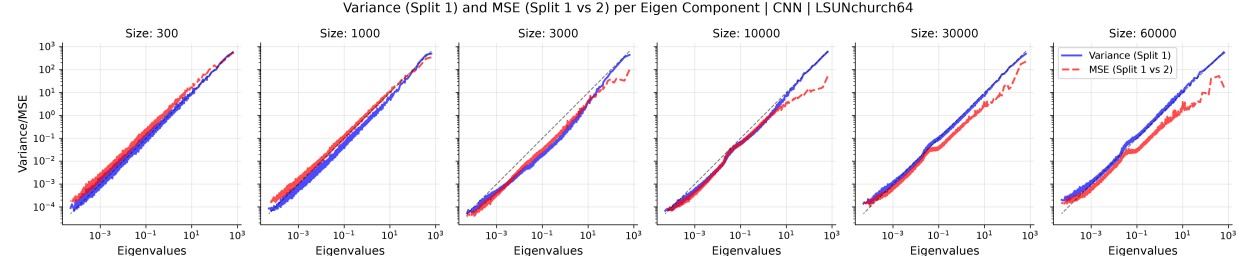

*Figure 43.* **DNN validation experiments, Anisotropy and overshrinking (LSUN church 64)**

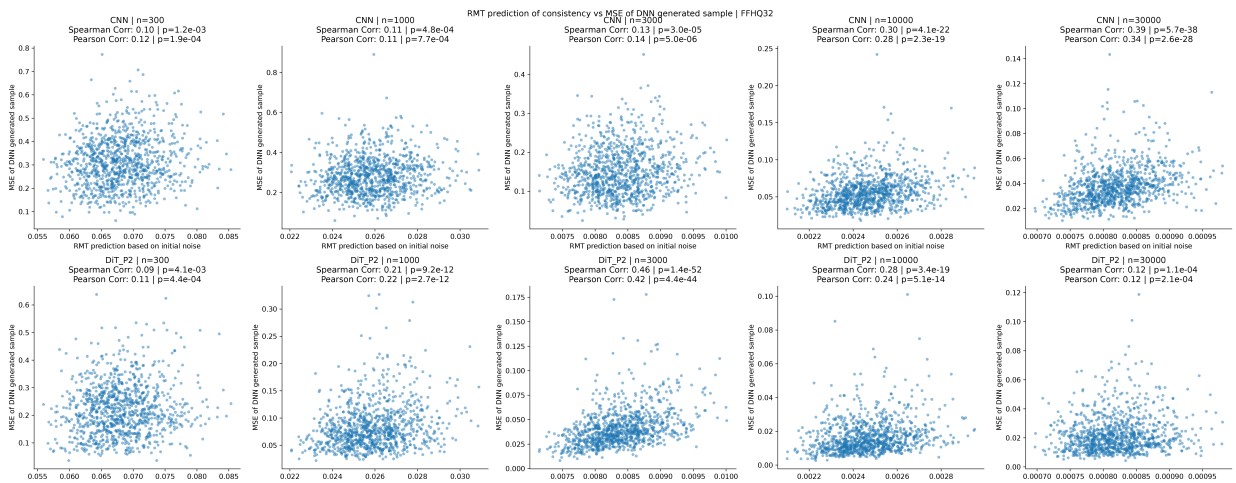

*Figure 44.* **DNN validation experiments, RMT predicting inhomogeneity (FFHQ32)**

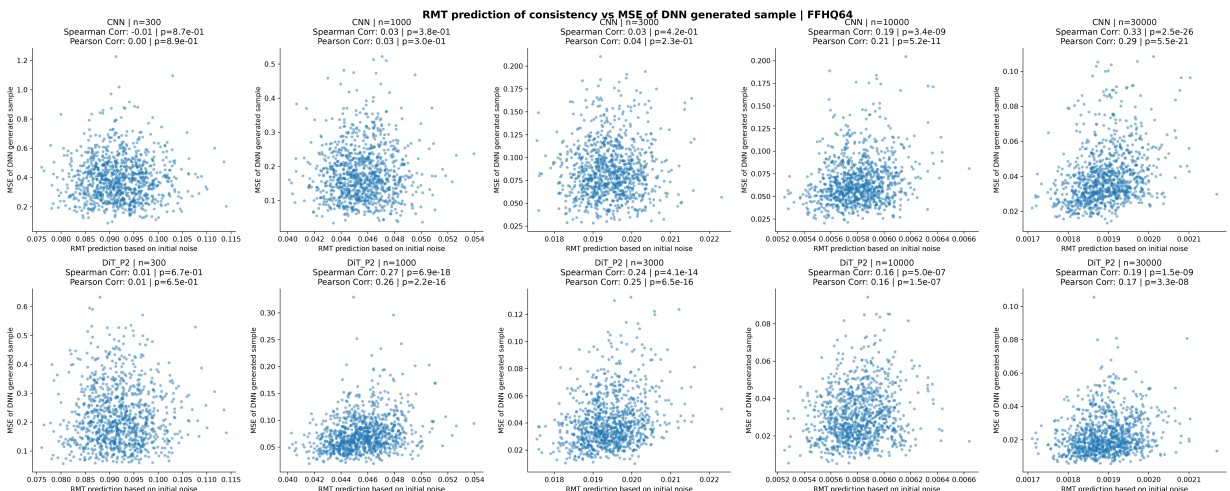

*Figure 45.* **DNN validation experiments, RMT predicting inhomogeneity (FFHQ64)**

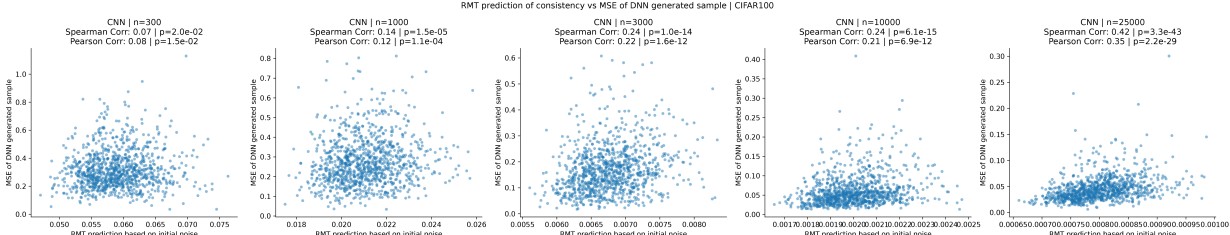

*Figure 46.* **DNN validation experiments, RMT predicting inhomogeneity (CIFAR100)**

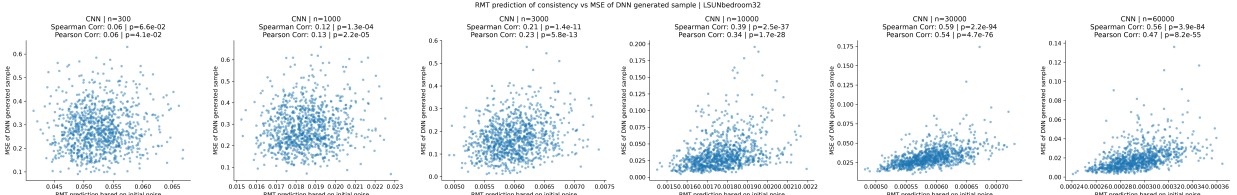

*Figure 47.* **DNN validation experiments, RMT predicting inhomogeneity (LSUN bedroom 32)**

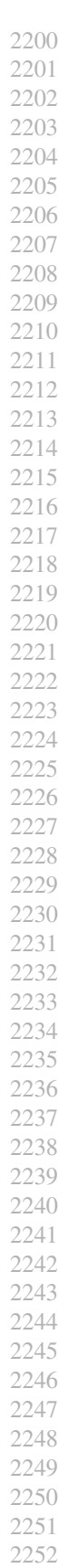

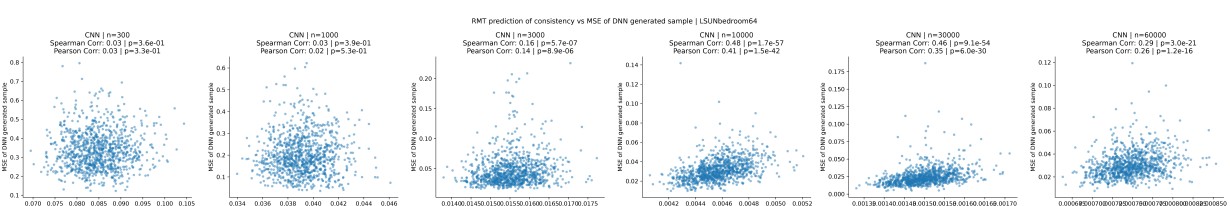

*Figure 48.* **DNN validation experiments, RMT predicting inhomogeneity (LSUN bedroom 64)**

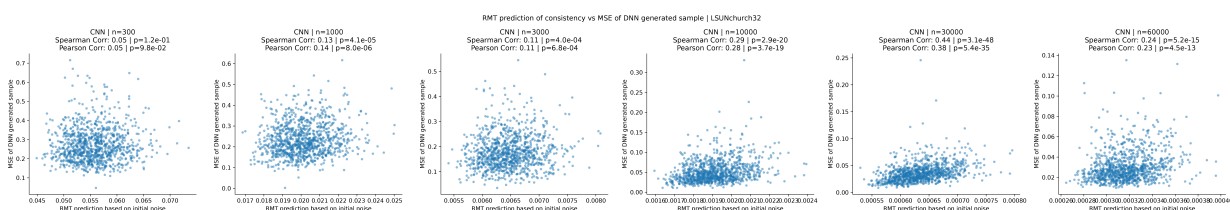

*Figure 49.* **DNN validation experiments, RMT predicting inhomogeneity (LSUN church 32)**

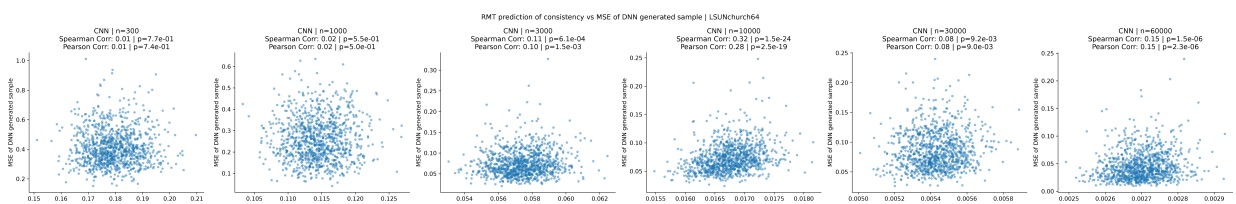

*Figure 50.* **DNN validation experiments, RMT predicting inhomogeneity (LSUN church 64)**

### B.4.6. ANISOTROPY OF INITIAL NOISE SPACE

**Top 50 eigenspace**
**std x 1.5**

**Top 50 eigenspace**
**std x 0.1**

*Figure 51.* **Anisotropic structure of the initial noise space (CNN-UNet FFHQ64). Left:** Amplifying the initial-noise amplitude in the top-50 eigenspace. **Right:** Decreasing the initial-noise amplitude in the top-50 eigenspace. Top: initial noise; Bottom: generated samples (same noise seed). Increasing noise in the dominant eigendirections introduces more visual artifacts by amplifying the top eigen-structure of the generative map (Eq. 3). Conversely, reducing noise in these dimensions yields a cleanly segmented face against a gray background.

Top200 eigenspace
std x 1.5

Top 200 eigenspace
std x 0.1

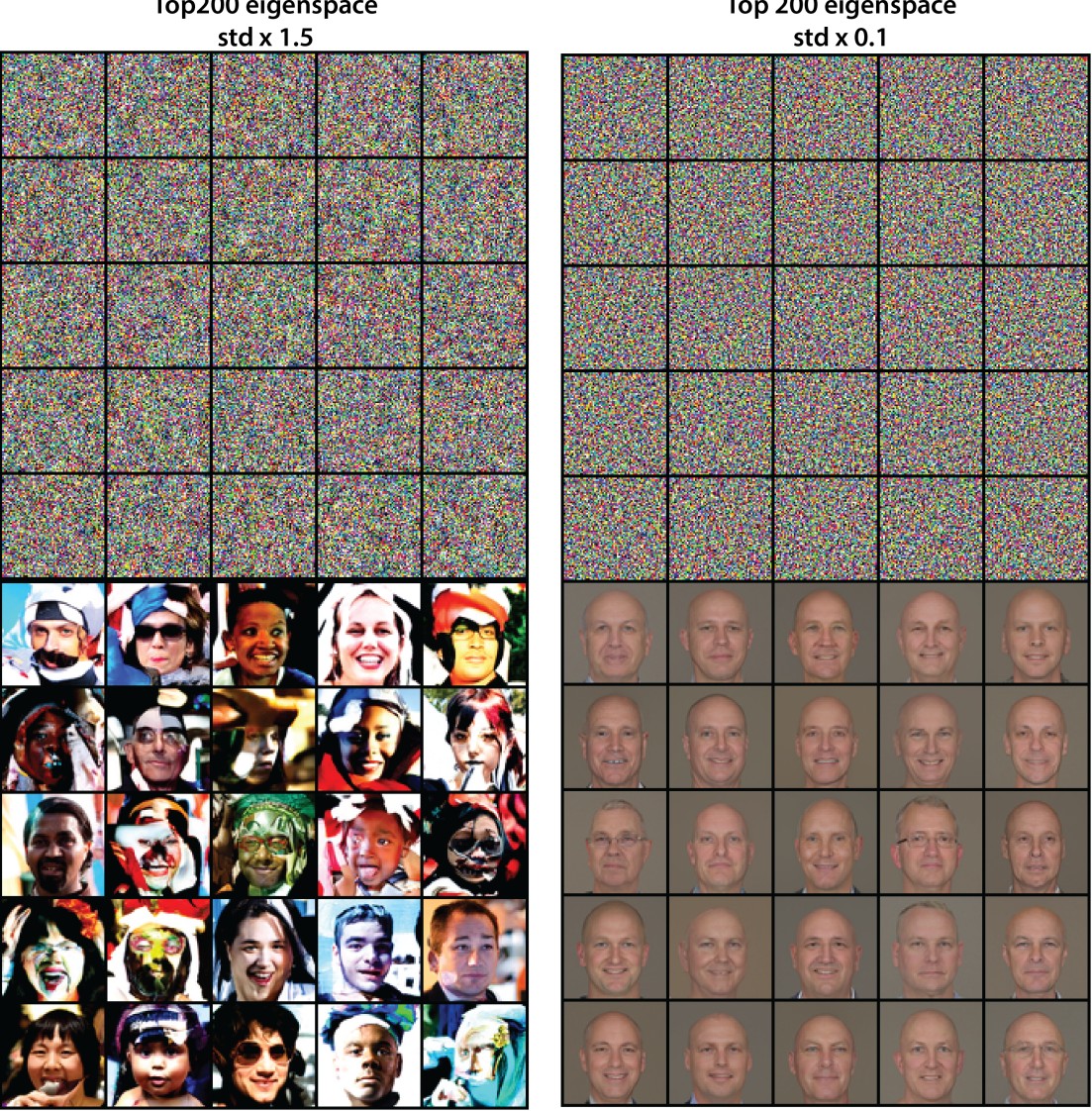

*Figure 52.* **Anisotropic structure of the initial noise space (CNN-UNet FFHQ64). Left:** Amplifying the initial-noise amplitude in the top-200 eigenspace. **Right:** Decreasing the initial-noise amplitude in the top-200 eigenspace. Top: initial noise; Bottom: generated samples (same noise seed). Similar to the top-50 case: stronger noise in leading eigendirections amplifies dominant structure, producing visible artifacts. Suppressing noise yields an even more homogeneous and simplified face, reflecting the removal of additional variation modes.

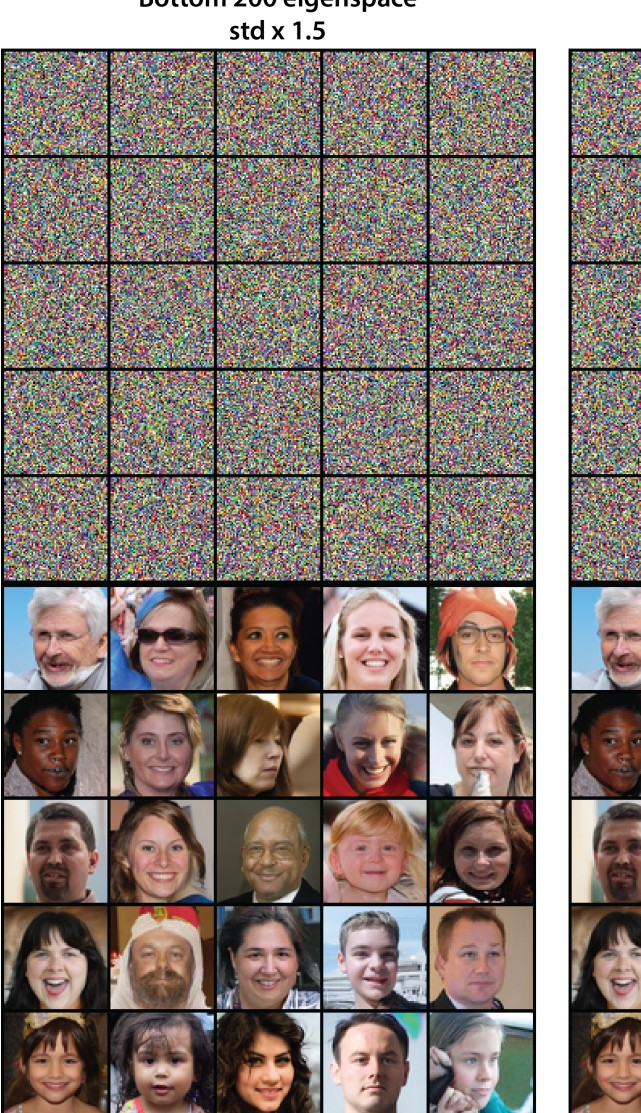

*Figure 53*. **Anisotropic structure of the initial noise space (CNN-UNet FFHQ64). Left:** Amplifying the initial-noise amplitude in the bottom-200 eigenspace. **Right:** Decreasing the initial-noise amplitude in the bottom-200 eigenspace. Top: initial noise; Bottom: generated samples (same noise seed). In contrast to perturbations along top eigendirections, manipulating the least-significant eigenspaces produces minimal perceptual impact on the generated image.

## C. Proof and Derivations

### C.1. Deterministic equivalence relations

Here we collect the one-point and two-point deterministic equivalent relationships from (Atanasov et al., 2024b; 2025; Bach, 2024), rewritten in a unified notation.

**Set up**   Using similar notation as Bach (2024), we consider data matrix $X \in \mathbb{R}^{n \times d}$, where each row is an i.i.d. sample $\mathbf{x}_i$. The population covariance of these samples is denoted as $\mathbf{\Sigma}$. The key object of analysis is their empirical covariance

$$\hat{\mathbf{\Sigma}} = \frac{1}{n} X^\top X$$

We use $\gamma = d/n$ to denote the aspect ratio of $X$, i.e. the dimensionality of data over the number of data points.

**Self-consistency equation for renormalized variable**   The spectral properties of a matrix are determined by the Stieltjes transform. We consider the Stieltjes transform of the kernel matrix $\frac{1}{n} X X^\top$, defined as $\hat{\varphi}(z) := \mathrm{Tr}[(X X^\top - nzI)^{-1}]$. At the large matrix limit, the limiting variable $\varphi(z)$ satisfy the following self consistent equation,

$$\frac{1}{\varphi(z)} + z = \gamma \int_0^\infty \frac{s d\boldsymbol{\mu}(s)}{1 + s\varphi(z)} \tag{10}$$

where $\boldsymbol{\mu}(s)$ is the limiting spectral measure of the population covariance $\mathbf{\Sigma}$. This follows from the leave-one-out arguments in the Appendix of Bach (2024), as well as Bai et al. (2010); Ledoit & Péché (2011).

This can be translated to the self-consistent equation of the renormalized ridge variable $\kappa(z) := \frac{1}{\varphi(-z)}$, which is used throughout the paper,

$$\frac{1}{\varphi(-z)} - z = \gamma \int_0^\infty \frac{s d\boldsymbol{\mu}(s)}{1 + s\varphi(-z)}$$

$$\kappa(z) - z = \gamma \int_0^\infty \frac{s d\boldsymbol{\mu}(s)}{1 + s \frac{1}{\kappa(z)}}$$

$$\kappa(z) - z = \gamma \kappa(z) \int_0^\infty \frac{s d\boldsymbol{\mu}(s)}{\kappa(z) + s}$$

$$z = \kappa(z) \Big[ 1 - \gamma \int_0^\infty \frac{s d\boldsymbol{\mu}(s)}{\kappa(z) + s} \Big]$$

Practically, when solving such equations, given a finite size population covariance matrix, the integral over the spectral measure can be represented as normalized trace, leading to the Silverstein equation (Eq.4).

$$\kappa(\lambda) - \lambda = \gamma \kappa(\lambda) \mathrm{tr}[\mathbf{\Sigma}(\mathbf{\Sigma} + \kappa(\lambda)I)^{-1}] \tag{Silverstein}$$

**Degrees of Freedom**   We define the degree of freedom functions with unnormalized trace, similar to convention in (Bach, 2024), unlike (Atanasov et al., 2025) which used normalized trace.

$$\mathrm{df}_1(\lambda) := \mathrm{Tr}[\mathbf{\Sigma}(\mathbf{\Sigma} + \lambda I)^{-1}] \tag{11}$$

$$\mathrm{df}_2(\lambda) := \mathrm{Tr}[\mathbf{\Sigma}^2(\mathbf{\Sigma} + \lambda I)^{-2}]. \tag{12}$$

Note that

$$\mathrm{df}_2(\kappa) - \mathrm{df}_1(\kappa) = \mathrm{Tr}[\mathbf{\Sigma}^2(\mathbf{\Sigma} + \kappa I)^{-2}] - \mathrm{Tr}[\mathbf{\Sigma}(\mathbf{\Sigma} + \kappa I)^{-1}]$$

$$= \mathrm{Tr}[(\mathbf{\Sigma}(\mathbf{\Sigma} + \kappa I)^{-1} - I)\mathbf{\Sigma}(\mathbf{\Sigma} + \kappa I)^{-1}]$$

$$= \kappa \mathrm{Tr}[\mathbf{\Sigma}(\mathbf{\Sigma} + \kappa I)^{-2}]$$

$$> 0.$$

Note that both $\mathrm{df}_2(\kappa), \mathrm{df}_1(\kappa)$ are smaller than the number on non-zero eigenvalues of $\mathbf{\Sigma}$, i.e. $\mathrm{rank}(\mathbf{\Sigma})$. Thus, we have the chain of inequalities

$$\min(n, d) \geq \mathrm{rank}(\mathbf{\Sigma}) > \mathrm{df}_2(\kappa) > \mathrm{df}_1(\kappa)$$

**Basic deterministic equivalences** Following Proposition 1 of Bach (2024), we use the shorthand $\kappa(z) := 1/\varphi(-z)$ to express the deterministic equivalences in the more convenient forms below. In what follows, $A$ and $B$ are test matrices of bounded spectral norm. For the resolvent, we have:

$$\mathrm{Tr}\big[A\,(\hat{\mathbf{\Sigma}} + \lambda I)^{-1}\big] \asymp \frac{\kappa(\lambda)}{\lambda}\,\mathrm{Tr}\big[A\big(\mathbf{\Sigma} + \kappa(\lambda)I\big)^{-1}\big] \tag{13}$$

and

$$\begin{aligned}
\mathrm{Tr}\big[A(\hat{\mathbf{\Sigma}} + \lambda I)^{-1}B(\hat{\mathbf{\Sigma}} + \lambda I)^{-1}\big] \\
\asymp \frac{\kappa(\lambda)^2}{\lambda^2}\,\mathrm{Tr}\big[A\big(\mathbf{\Sigma} + \kappa(\lambda)I\big)^{-1}B\big(\mathbf{\Sigma} + \kappa(\lambda)I\big)^{-1}\big] \\
+ \frac{\kappa(\lambda)^2}{\lambda^2}\frac{1}{n - \mathrm{df}_2\big(\kappa(\lambda)\big)}\,\mathrm{Tr}\big[A\big(\mathbf{\Sigma} + \kappa(\lambda)I\big)^{-2}\mathbf{\Sigma}\big]\,\mathrm{Tr}\big[B\big(\mathbf{\Sigma} + \kappa(\lambda)I\big)^{-2}\mathbf{\Sigma}\big].
\end{aligned} \tag{14}$$

Equivalently,

$$\mathrm{Tr}\big[A\hat{\mathbf{\Sigma}}(\hat{\mathbf{\Sigma}} + \lambda I)^{-1}\big] \asymp \mathrm{Tr}\big[A\,\mathbf{\Sigma}\big(\mathbf{\Sigma} + \kappa(\lambda)I\big)^{-1}\big] \tag{15}$$

and

$$\begin{aligned}
\mathrm{Tr}\big[A\,\hat{\mathbf{\Sigma}}\,(\hat{\mathbf{\Sigma}} + \lambda I)^{-1}\,B\,\hat{\mathbf{\Sigma}}\,(\hat{\mathbf{\Sigma}} + \lambda I)^{-1}\big] \\
\asymp \mathrm{Tr}\big[A\,\mathbf{\Sigma}\big(\mathbf{\Sigma} + \kappa(\lambda)I\big)^{-1}B\,\mathbf{\Sigma}\big(\mathbf{\Sigma} + \kappa(\lambda)I\big)^{-1}\big] \\
+ \frac{\kappa^2(\lambda)}{n - \mathrm{df}_2\big(\kappa(\lambda)\big)}\,\mathrm{Tr}\big[A\big(\mathbf{\Sigma} + \kappa(\lambda)I\big)^{-2}\mathbf{\Sigma}\big]\,\mathrm{Tr}\big[B\big(\mathbf{\Sigma} + \kappa(\lambda)I\big)^{-2}\

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

$$\mathrm{Tr}\left[A(\hat{\boldsymbol{\Sigma}} + zI)^{-1}\right] \sim \frac{\kappa(z)}{z}\mathrm{Tr}\left[A(\boldsymbol{\Sigma} + \kappa(z)I)^{-1}\right]$$

$$\mathrm{Tr}\big[A(\hat{\boldsymbol{\Sigma}} + zI)^{-1}B(\hat{\boldsymbol{\Sigma}} + zI)^{-1}\big] \sim \frac{\kappa(z)^2}{z^2}\,\mathrm{Tr}\big[A\left(\boldsymbol{\Sigma} + \kappa(z)I\right)^{-1}B\left(\boldsymbol{\Sigma} + \kappa(z)I\right)^{-1}\big]$$

$$+ \frac{\kappa(z)^2}{z^2}\,\mathrm{Tr}\big[A\left(\boldsymbol{\Sigma} + \kappa(z)I\right)^{-2}\boldsymbol{\Sigma}\big]\,\mathrm{Tr}\big[B\left(\boldsymbol{\Sigma} + \kappa(z)I\right)^{-2}\boldsymbol{\Sigma}\big]\frac{1}{n - \mathrm{df}_2\big(\kappa(z)\big)}$$

The 2nd moment term is equivalent to,

$$\mathrm{Tr}\big[A(\hat{\boldsymbol{\Sigma}} + zI)^{-1}B(\hat{\boldsymbol{\Sigma}} + zI)^{-1}\big]$$

$$\sim \frac{\kappa(z)^2}{z^2}\,\mathrm{Tr}\big[\mathbf{v}\mathbf{v}^\top\left(\boldsymbol{\Sigma} + \kappa(z)I\right)^{-1}(\boldsymbol{\mu} - \mathbf{x})(\boldsymbol{\mu} - \mathbf{x})^\top\left(\boldsymbol{\Sigma} + \kappa(z)I\right)^{-1}\big]$$

$$+ \frac{\kappa(z)^2}{z^2}\frac{1}{n - \mathrm{df}_2\big(\kappa(z)\big)}\,\mathrm{Tr}\big[\mathbf{v}\mathbf{v}^\top\left(\boldsymbol{\Sigma} + \kappa(z)I\right)^{-2}\boldsymbol{\Sigma}\big]\,\mathrm{Tr}\big[(\boldsymbol{\mu} - \mathbf{x})(\boldsymbol{\mu} - \mathbf{x})^\top\left(\boldsymbol{\Sigma} + \kappa(z)I\right)^{-2}\boldsymbol{\Sigma}\big]$$

$$= \frac{\kappa(z)^2}{z^2}\Big(\mathbf{v}^\top\left(\boldsymbol{\Sigma} + \kappa(z)I\right)^{-1}(\boldsymbol{\mu} - \mathbf{x})\Big)^2$$

$$+ \frac{\kappa(z)^2}{z^2}\frac{1}{n - \mathrm{df}_2\big(\kappa(z)\big)}\Big(\mathbf{v}^\top\left(\boldsymbol{\Sigma} + \kappa(z)I\right)^{-2}\boldsymbol{\Sigma}\mathbf{v}\Big)\Big((\boldsymbol{\mu} - \mathbf{x})^\top\left(\boldsymbol{\Sigma} + \kappa(z)I\right)^{-2}\boldsymbol{\Sigma}(\boldsymbol{\mu} - \mathbf{x})\Big)$$

The first moment term is equivalent to,

$$\mathrm{Tr}\left[(\hat{\boldsymbol{\Sigma}} + zI)^{-1}(\boldsymbol{\mu} - \mathbf{x})\mathbf{v}^\top\right] \sim \frac{\kappa(z)}{z}\,\mathrm{Tr}\big[\left(\boldsymbol{\Sigma} + \kappa(z)I\right)^{-1}(\boldsymbol{\mu} - \mathbf{x})\mathbf{v}^\top\big]$$

$$= \frac{\kappa(z)}{z}\mathbf{v}^\top\left(\boldsymbol{\Sigma} + \kappa(z)I\right)^{-1}(\boldsymbol{\mu} - \mathbf{x})$$

Thus, combining the two terms, we obtain the variance of score at noised datapoint $\mathbf{x}$, along direction $\mathbf{v}$,

$$\mathbf{v}^\top \mathcal{S}_s(\mathbf{x})\mathbf{v} = Var_{\hat{\boldsymbol{\Sigma}}}[\mathbf{v}^\top \mathbf{s}_{\hat{\boldsymbol{\Sigma}}}^*(\mathbf{x}; \sigma)]$$

$$= \mathbb{E}_{\hat{\boldsymbol{\Sigma}}}\,\mathrm{Tr}\left[\mathbf{v}\mathbf{v}^\top(\hat{\boldsymbol{\Sigma}} + \sigma^2 I)^{-1}(\boldsymbol{\mu} - \mathbf{x})(\boldsymbol{\mu} - \mathbf{x})^\top(\hat{\boldsymbol{\Sigma}} + \sigma^2 I)^{-1}\right]$$

$$- \left(\mathbb{E}_{\hat{\boldsymbol{\Sigma}}}\,\mathrm{Tr}\left[(\hat{\boldsymbol{\Sigma}} + \sigma^2 I)^{-1}(\boldsymbol{\mu} - \mathbf{x})\mathbf{v}^\top\right]\right)^2$$

$$\sim \frac{\kappa(z)^2}{z^2}\Big(\mathbf{v}^\top\left(\boldsymbol{\Sigma} + \kappa(z)I\right)^{-1}(\boldsymbol{\mu} - \mathbf{x})\Big)^2$$

$$+ \frac{\kappa(z)^2}{z^2}\Big(\mathbf{v}^\top\left(\boldsymbol{\Sigma} + \kappa(z)I\right)^{-2}\boldsymbol{\Sigma}\mathbf{v}\Big)\Big((\boldsymbol{\mu} - \mathbf{x})^\top\left(\boldsymbol{\Sigma} + \kappa(z)I\right)^{-2}\boldsymbol{\Sigma}(\boldsymbol{\mu} - \mathbf{x})\Big)\frac{1}{n - \mathrm{df}_2\big(\kappa(z)\big)}$$

$$- \Big(\frac{\kappa(z)}{z}\mathbf{v}^\top\left(\boldsymbol{\Sigma} + \kappa(z)I\right)^{-1}(\boldsymbol{\mu} - \mathbf{x})\Big)^2$$

$$= \frac{1}{n - \mathrm{df}_2\big(\kappa(z)\big)}\frac{\kappa(z)^2}{z^2}\Big(\mathbf{v}^\top\left(\boldsymbol{\Sigma} + \kappa(z)I\right)^{-2}\boldsymbol{\Sigma}\mathbf{v}\Big)\Big((\boldsymbol{\mu} - \mathbf{x})^\top\left(\boldsymbol{\Sigma} + \kappa(z)I\right)^{-2}\boldsymbol{\Sigma}(\boldsymbol{\mu} - \mathbf{x})\Big)$$

$$(z \mapsto \sigma^2) = \frac{1}{n - \mathrm{df}_2\big(\kappa(\sigma^2)\big)}\frac{\kappa(\sigma^2)^2}{\sigma^4}\Big(\mathbf{v}^\top\left(\boldsymbol{\Sigma} + \kappa(\sigma^2)I\right)^{-2}\boldsymbol{\Sigma}\mathbf{v}\Big)\Big((\boldsymbol{\mu} - \mathbf{x})^\top\left(\boldsymbol{\Sigma} + \kappa(\sigma^2)I\right)^{-2}\boldsymbol{\Sigma}(\boldsymbol{\mu} - \mathbf{x})\Big)$$

Per simple scaling, the variance of denoisers reads,

$$\mathbf{v}^\top \mathcal{S}_D(\mathbf{x})\mathbf{v} = \sigma^4 \mathbf{v}^\top \mathcal{S}_s(\mathbf{x})\mathbf{v}$$

$$\sim \frac{\kappa(\sigma^2)^2}{n - \mathrm{df}_2\big(\kappa(\sigma^2)\big)}\underbrace{\Big(\mathbf{v}^\top\left(\boldsymbol{\Sigma} + \kappa(\sigma^2)I\right)^{-2}\boldsymbol{\Sigma}\mathbf{v}\Big)}_{\diamond(\mathbf{v}, \kappa, \boldsymbol{\Sigma})}\underbrace{\Big((\boldsymbol{\mu} - \mathbf{x})^\top\left(\boldsymbol{\Sigma} + \kappa(\sigma^2)I\right)^{-2}\boldsymbol{\Sigma}(\boldsymbol{\mu} - \mathbf{x})\Big)}_{\diamond(\boldsymbol{\mu} - \mathbf{x}, \kappa, \boldsymbol{\Sigma})}$$

$\square$

C.3.1. INTERPRETATION AND DERIVATIONS

**Dependency on probe direction v**   This dependency on $\mathbf{v}$ tells us about the *anisotropy of uncertainty*, or variance of the score / denoiser prediction on different directions.

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

and when $\sigma_T \to \infty$ the normalized initial noise are sampled from standard Gaussian, $\bar{\mathbf{x}} \sim \mathcal{N}(0, I)$.

Equivalently, we can consider expansion as orders of $1/\sigma$,

$$\sigma \boldsymbol{\Sigma}^{1/2}(\boldsymbol{\Sigma} + \sigma^2 I)^{-1/2} = \boldsymbol{\Sigma}^{1/2}(I + \frac{1}{\sigma^2}\boldsymbol{\Sigma})^{-1/2}$$

$$\approx \boldsymbol{\Sigma}^{1/2}(I - \frac{1}{2}\frac{1}{\sigma^2}\boldsymbol{\Sigma} + ...)$$

$$\approx \boldsymbol{\Sigma}^{1/2} - \frac{1}{2}\frac{1}{\sigma^2}\boldsymbol{\Sigma}^{3/2} + ...$$

If we keep the zeroth-order term, then we get the approximation

$$\sigma \boldsymbol{\Sigma}^{1/2}(\boldsymbol{\Sigma} + \sigma^2 I)^{-1/2} \approx \boldsymbol{\Sigma}^{1/2}$$

Consider approximation,

$$\mathbf{x}(\mathbf{x}_{\sigma_T}, 0) = \boldsymbol{\mu} + \hat{\boldsymbol{\Sigma}}^{1/2}(\hat{\boldsymbol{\Sigma}} + \sigma_T^2 I)^{-1/2}(\mathbf{x}_{\sigma_T} - \boldsymbol{\mu})$$

$$\approx \boldsymbol{\mu} + \hat{\boldsymbol{\Sigma}}^{1/2}(\frac{\mathbf{x}_{\sigma_T} - \boldsymbol{\mu}}{\sigma_T})$$

$$= \boldsymbol{\mu} + \hat{\boldsymbol{\Sigma}}^{1/2}\bar{\mathbf{x}}$$

then we can study the effect of finite sample on sampling mapping via the matrix $\hat{\boldsymbol{\Sigma}}^{1/2}$.

**Proposition C.6.** *Deterministic equivalence of empirical covariance matrix one half*

$$\hat{\boldsymbol{\Sigma}}^{1/2} = \frac{2}{\pi} \int_0^\infty \hat{\boldsymbol{\Sigma}}(\hat{\boldsymbol{\Sigma}} + u^2 I)^{-1} du \tag{21}$$

$$\asymp \frac{2}{\pi} \int_0^\infty \boldsymbol{\Sigma}\left(\boldsymbol{\Sigma} + \kappa(u^2)I\right)^{-1} du \tag{22}$$

*Proof.* Combining Lemma 3 with deterministic equivalence of one point (Eq. DE). $\qquad\square$

This result can be compared to population covariance half, when renormalization effect vanish $\kappa(u^2) \to u^2$.

$$\boldsymbol{\Sigma}^{1/2} = \frac{2}{\pi} \int_0^\infty \boldsymbol{\Sigma}(\boldsymbol{\Sigma} + u^2 I)^{-1} du$$

Since $\kappa(u^2) > u^2$ point by point in the integral, the sample version leads to larger shrinkage.

$$\mathbf{v}^\top \hat{\boldsymbol{\Sigma}}^{1/2} \mathbf{v} < \mathbf{v}^\top \boldsymbol{\Sigma}^{1/2} \mathbf{v}$$

Concretely, if we measure along spectral modes $\mathbf{u}_k$ of population covariance,

$$
\begin{aligned}
\mathbf{u}_k^\top \hat{\boldsymbol{\Sigma}}^{1/2} \mathbf{u}_k &\asymp \frac{2}{\pi} \int_0^\infty \mathbf{u}_k^\top \boldsymbol{\Sigma} \Big(\boldsymbol{\Sigma} + \kappa(u^2) I\Big)^{-1} \mathbf{u}_k du \\
&= \frac{2}{\pi} \int_0^\infty \frac{\lambda_k}{\lambda_k + \kappa(u^2)} du \\
&< \frac{2}{\pi} \int_0^\infty \frac{\lambda_k}{\lambda_k + u^2} du \\
&= \lambda_k^{1/2}
\end{aligned}
$$

**C.6. Proof for expectation of the sampling mapping (full version, finite $\sigma_T$)**

Next, we consider the finite $\sigma_T$ case, which involves two matrix half and their equivalence. To prove this, we proceed in two steps 1) use integral identity to represent matrix of this form $A^{1/2}(A + zI)^{-1/2}$, 2) apply one point deterministic equivalence.

**Proposition C.7.** *Integral representation, for self-adjoint, positive semi definite matrix $A \succeq 0$,*

$$
\begin{aligned}
A^{1/2}(A + zI)^{-1/2} &= \frac{4}{\pi^2} \int_0^\infty \int_0^\infty A(A + u^2 I)^{-1}(A + (z + v^2)I)^{-1} du dv \\
&= \frac{4}{\pi^2} \int_0^\infty \int_0^\infty \frac{A(A + (z + u^2)I)^{-1} - A(A + v^2 I)^{-1}}{v^2 - u^2 - z} du dv
\end{aligned}
$$

*Proof.* We can study matrix of this form,
$$A^{1/2}(A + zI)^{-1/2}$$

using the integral representation above twice, we have

$$
\begin{aligned}
&A^{1/2}(A + zI)^{-1/2} \\
=&A A^{-1/2}(A + zI)^{-1/2} \\
=&\frac{1}{\pi^2} A \int_0^\infty (A + sI)^{-1} s^{-1/2} ds \int_0^\infty (A + (z + t)I)^{-1} t^{-1/2} dt \\
=&\frac{1}{\pi^2} \int_0^\infty \int_0^\infty A(A + sI)^{-1}(A + (z + t)I)^{-1} t^{-1/2} s^{-1/2} ds dt \\
=&\frac{4}{\pi^2} \int_0^\infty \int_0^\infty A(A + u^2 I)^{-1}(A + (z + v^2)I)^{-1} du dv
\end{aligned}
$$

To deal with this *product of resolvent*, we can turn it into *difference of resolvent*

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

We found when the dataset size is not enough, e.g. rank deficient $\hat{\Sigma}$, the eigendecomposition is not stable, sometimes generating negative eigenvalues, which affects the matrix square root operation in Wiener matrix. Even if we clip them, there is often numerical artifacts at small eigenspaces. One solution is, we use higher precision float64 number to yield similar results with the theory.

### D.3. Deep neural network experiments

We used following preconditioning scheme inspired by (Karras et al., 2022), for all our architectures for comparison.