# OpenReview forum: "A Random Matrix Perspective on the Consistency of Diffusion Models"
_ICML.cc/2026/Conference — ICML 2026 spotlight_

### Official Review · Reviewer_KkST · 2026-03-10

**Soundness:** 3
**Presentation:** 4
**Significance:** 3
**Originality:** 3
**Overall Recommendation:** 5
**Confidence:** 4

**Summary:**

The authors investigate the phenomenon, first observed by Zhang et al 2024, that diffusion methods tend to generate very similar images, when initiated with the same “generative noise”, regardless of methods and architectures. They provide a convincing theoretical explanation for this phenomenon. They investigate the problem as estimating the population statistics given limited data and analyze the inaccuracies occurred, specifically for the covariance matrix, in this setting. A complete analysis is performed for s linear generator using tools from random matrix theory. They basically conclude that lower image frequencies, which are determined early on in the generative process, are estimated more accurately using limited data. The authors partition the variance of denoisers across datasets into 3 components: anisotropy, inhomogeneity and global scaling and explain the tradeoffs. The paper is well written, claims and propositions are followed by comprehensive proofs in the appendix and experiments appear to support the main claims of the paper.

**Compliance With Llm Reviewing Policy:**

Affirmed.

**Final Justification:**

My concerns are resolved, I reamain with my positive rating. I think this paper can be strengthened if the open issues, as discussed above by the authors, would be incorporated in the paper (or supp), esp. regarding higher order statistics and the reason the theory applies well for diffusion generative models but not for GANs and VAEs.

**Key Questions For Authors:**

1. Can you expand more on higher statistical moments, beyond covariance?
2. How different architectures play a role in your setting and assumptions?
3. What are the reasons according to your analysis that these properties hold for diffusion generative models but not for other models?

**Limitations:**

Essential analysis is limited to the linear denoiser / generator setting, but it seems to generalize well.

**Strengths And Weaknesses:**

The paper is sound and well written. It is of significance to the community. No major weaknesses.

---

> ### Author Rebuttal · Authors · 2026-03-30
>
> We thank Reviewer KkST for their positive evaluation, recognizing the paper as sound, well-written, and of significance to the community. We are glad the reviewer found our theoretical explanation convincing and the claims well-supported by comprehensive proofs and experiments. We appreciate the stimulating questions about higher-order moments, the role of architecture, and the specificity of the consistency phenomenon to diffusion models, which help us articulate the broader implications of our work. We address each question below.
>
> ### Point 1: Higher statistical moments beyond covariance
>
> > **[Q1]** *Can you expand more on higher statistical moments, beyond covariance?*
> >
>
> Higher moments contribute to finer-grained image features and become relevant at lower noise levels where the posterior is no longer well-approximated by a Gaussian. Analyzing these within our framework would require going beyond the linear denoiser to nonlinear estimators, where the relevant RMT tools are considerably more involved (e.g. the random feature case is likely feasible and we are working on it). We expect that higher moments are estimated *less* consistently across data splits (since they require more samples to estimate accurately), which would explain why fine details show more cross-split variability while global structure remains consistent and why the generations from trained neural networks are less consistent than the linear denoisers. We will add a discussion of this point.
>
> ### Point 2: Role of different architectures
>
> > **[Q2]** *How do different architectures play a role in your setting and assumptions?*
> >
>
> Very good question! Indeed this is one limitation of our framework. Currently the results presented in the paper does not quite explain the factor of network architecture, since it takes an architecture-independent assumption about it (e.g. linear constraint). Empirically, we can see the approximation work for both DiT and UNet but to different degree, we cannot explain their difference.
>
> To discuss the effect of architecture in future work, one may need to revise the theory. One potential next step on the theory side is to treat different architectures as different kernel machines, then one can write down the optimal denoiser as the kernel regression solution. In that case, one can discuss the effect of different feature spaces on the solution and their consistency and generation quality. The more general theory beyond the kernel regime would definitely need more development.
>
> ### Point 3: Why diffusion models but not other generative models?
>
> > **[Q3]** *What are the reasons according to your analysis that these properties hold for diffusion generative models but not for other models?*
> >
>
> This is a very intriguing question. I think the key answer lies in the fact that diffusion models are using neural networks to approximate the score function $\nabla \log p$, for which the simplest, linear approximation rely on the mean and covariance, which is the Gaussian statistics of the data. As our RMT analysis shows, these statistics and certain observable derived from these Gaussian statistics are highly consistent.
>
> For autoregressive model, the stochasticity comes from the sampling next token from the softmax distribution $p(x_t | x_{1:t-1})$, thus it does not behave like the deterministic sampler of diffusion as we analyzed in the paper.
>
> In contrast, VAE, GAN both learn a mapping from (usually) a lower dimensional space to a higher dimensional space, and the lower dimensional latent space usually samples from an isotropic Gaussian $x=f(z),z\sim\mathcal N(0,I)$. For this reason, the mapping $f$ can be composed with an orthogonal transform $O$ and not changing the mapping $f\circ O$. So the mapping is in principle not identifiable, thus training on different splits could results in different rotations $O$, and the same noise pattern $z$ will not necessarily map to similar output.
>
> One thing we note is that, for VAE or GAN, one could in principle align the latent space or fixing the orthogonal transform by choosing the right singular vector basis of the Jacobian $\partial f/\partial z$. Some previous work show that using this basis, then the axis of different GAN trained on human face dataset will have relatively consistent semantics (not necesarily the same samples) [Wang, Ponce, ICLR, 2021, Figure 10].
>
> If there is general interest, we could add this discussion to the final paper.

---

> > ### Author Rebuttal · Reviewer_KkST · 2026-04-04
> >
> > My concerns are resolved, I reamain with my positive rating. I think this paper can be strengthened if the open issues, as discussed above by the authors, would be incorporated in the paper (or supp), esp. regarding higher order statistics and the reason the theory applies well for diffusion generative models but not for GANs and VAEs.

---

> > > ### Author Response · Authors · 2026-04-05
> > >
> > > Thank you a lot for your suggestion, we will make sure to add these intriguing discussions to our appendix in the camera ready version! We appreciate the reviewers for efforts and maintaining their positive evaluation of our paper!

---

### Official Review · Reviewer_QzAN · 2026-03-11

**Soundness:** 4
**Presentation:** 4
**Significance:** 4
**Originality:** 4
**Overall Recommendation:** 5
**Confidence:** 4

**Summary:**

The paper explains why diffusion models trained on different data splits generate similar outputs from the same noise seed. Using a random matrix theory framework, it shows shared Gaussian statistics largely determine samples. The theory quantifies finite-data effects, sampling variability, and cross-split differences, and matches experiments on linear models, UNet, and DiT.

**Compliance With Llm Reviewing Policy:**

Affirmed.

**Final Justification:**

The rebuttal satisfactorily addressed my concerns, and the paper is worthy of acceptance.

**Key Questions For Authors:**

How sensitive are the theoretical predictions to non-Gaussian (but i.i.d) data distributions? Can we extend the results to the sub-Gaussian case?

**Limitations:**

Yes.

**Strengths And Weaknesses:**

Pros:

1 The paper studies a instereting phenomenon: diffusion models trained on different dataset splits producing nearly identical outputs when given the same noise seed. This problem is striking but under-discussed.

2 The use of RMT to analyze finite-dataset effects in diffusion models is really elegant. The derivation of self-consistent noise renormalization and variance formulas provides a clear theoretical explanation for the observed behavior.

3 The paper provides Insightful decomposition of variability including spectral anisotropy, input inhomogeneity, and dataset size scaling.

4 Importantly, despite being a theory-heavy paper, it  includes extensive empirical evaluations across several benchmarks. I appreciate the authors’ effort in conducting a broad empirical study.

Cons:

1 The explanation relies heavily on Gaussian statistics shared across dataset splits.

2 The experiments validate theoretical predictions but appear mainly focused on confirming the theory rather than demonstrating or inspiring  new methods.

3 Not a weakness but a suggestion. The paper focuses primarily on the analysis of the mean estimator. Extending the analysis to a consistent estimator of the covariance of the generated data could potentially lead to more interesting and informative results. Just like many previous RMT works on the consistent estimator for the sample coviance.

---

> ### Author Rebuttal · Authors · 2026-03-30
>
> We thank Reviewer QzAN for their enthusiastic assessment. We are pleased that the reviewer found the phenomenon striking and under-discussed, the RMT analysis elegant, the variance decomposition insightful, and the breadth of empirical evaluation appreciated. We also value the constructive suggestions regarding the reliance on Gaussian assumptions, the potential for methodological applications, and the excellent idea of extending the analysis to consistent covariance estimation. We address each point below.
>
> > **[W1]** *The explanation relies heavily on Gaussian statistics shared across dataset splits.*
> >
>
> Great point! This is exactly the point we want to make, i.e. we are arguing that it is the shared Gaussian statistics leading to consistency in generation, and violation of this will decrease the consistency effect (Appendix B.2). We think this assumption or argument generally holds for practical datasets. We will add a remark clarifying the range of distributional assumptions under which our results apply.
>
>
> > **[W2]** *The experiments validate theoretical predictions but appear mainly focused on confirming the theory rather than demonstrating or inspiring new methods.*
> >
>
> We agree that the current paper is primarily analytical and the experiments focused on consolidate the relevance of the theory instead of inspiring new methods.
>
> Regarding inference, one potentially practical application of the theory (not just RMT per se, but the linear predictor) is that the initial noise space has a well defined eigenframe, and the alignment of the initial noise sample to that eigenframe can impact many things including the quality of generated images and the consistency of generated samples across dataset splits. We showed evidence of this in App. B.4.6, where projecting and manipulating the initial noise in each eigen subspaces can effectively ablate the background or different facial features. So I think this idea could potentially inspire new inference time control and edit techniques.
>
> Regarding training, some fruitful direction may be to connect the linear solution to the learning dynamics of diffusion, see how the learning of the Gaussian statistics unfolds and how that learning dynamics depend on learning rate, batch size, EMA etc. Further, different of data augmentation can potentially help or hurt the efficiency of learning these statistics.
>
> > **[W3]** *Extending the analysis to a consistent estimator of the covariance of the generated data could lead to more interesting results, analogous to RMT works on consistent estimation for the sample covariance.*
> >
>
> This is an excellent suggestion!! Indeed, if we understand it correctly, consistent covariance estimation (linear shrinkage per Ledoit-Wolf 2004 or the nonlinear shrinkage estimators of Ledoit & Wolf 2020) could be used to *improve* the linear denoiser by correcting for finite-sample spectral bias. This would connect our work to the optimal shrinkage literature [A, E from Reviewer vuve's references] and could yield a principled denoiser that is both better calibrated and more consistent across splits. We will add this as a highlighted future direction!
>
> > **[Q1]** *How sensitive are the theoretical predictions to non-Gaussian (but i.i.d) data distributions? Can we extend the results to the sub-Gaussian case?*
> >
>
> As noted above, technically, our core results rely on concentration of sample covariance eigenvalues and certain bilinear forms of the covariance, which extends to sub-Gaussian distributions via classic RMT universality results (e.g., Bai & Silverstein). For heavier-tailed distributions, consistency may degrade. We will include a formal remark on this.

---

> > ### Author Rebuttal · Reviewer_QzAN · 2026-04-01
> >
> > After reading the authors’ response, most of my concerns are addressed. I vote for accept.

---

> > > ### Author Response · Authors · 2026-04-01
> > >
> > > We appreciate the reviewer for their positive evaluation of our work!

---

### Official Review · Reviewer_eYh8 · 2026-03-12

**Soundness:** 3
**Presentation:** 3
**Significance:** 3
**Originality:** 3
**Overall Recommendation:** 5
**Confidence:** 3

**Summary:**

The paper investigates the empirical observation that diffusion models trained
on disjoint subsets of a given dataset (and possibly also employing different architectures) tend to generate very similar images when started from the same
Gaussian noise seed. It employs Random Matrix Theory to study a linear simplification of diffusion models and analyzes how properties of the data samples
(respectively their underlying distribution) influences the behavior of this linear
model. A range of empirical results are provides to illustrate these theoretical
considerations.

**Compliance With Llm Reviewing Policy:**

Affirmed.

**Final Justification:**

The authors successfully addressed my concerns in their rebuttal, causing me to change the score from 3 to 5.

**Key Questions For Authors:**

1. To what extent does the similarity of the output of diffusion model to the
output of the linear model rely on the diffusion model being insufficiently
trained, i.e. when properly trained does it simply produce similar outputs with less artifacts and
more refined details or does it diverge further from the linear model?
2. Can you provide a theoretical argument why your linear theory already
captures important aspects of diffusion models?
3. Can you condense the paper to a more reasonable (under 30 pages) length?

**Limitations:**

yes

**Strengths And Weaknesses:**

Strengths:
1. The approach of considering optimal linear denoising steps (instead of
learned) ones in order to make the problem mathematically tractable
seems sensible.
2. Random matrix theory is well established and it seems reasonable to apply
it to this problem. There are no obvious flaws in the application of the
theory.

Weaknesses:
1. While the simplification to a linear model is a reasonable approach in
principle, it is important to either theoretically or empirically consider
how this simplified model is related to actual (non-linear and trained)
diffusion models. This is missing from the paper.
2. A key modeling claim of the paper is that the linear theory already captures the behavior of diffusion models (at least with regards to consistency). In light of this, it is concerning that the generated image in, e.g.,
Figures 1 and 6 are of very noticeably lower quality than is possible in
the EDM framework (for the FFHQ64 case a pretrained network, which
generated significantly better images, can be found via the EDM github),
likely due to insufficient training iterations. Despite the length of the paper there is only very limited evidence presented that the linear model is
a reasonable approximation of diffusion models (note that I did not check
all 50+ Appendix pages in detail).
3. While is is common to put some technical proofs and additional illustrations in the Appendix, 64 total pages seems excessive, when the page limit
for the main paper is 8. It is important to pick and choose which aspects
are important and assemble you results into a concise presentation. If
you want to make additional illustrations and numerical results available
putting them on, e.g., github and adding a link seems more appropriate.
Similarly if extensive proofs and calculations are needed, a longer form
publication like a journal seems more appropriate.

---

> ### Author Rebuttal · Authors · 2026-03-30
>
> We thank Reviewer eYh8 for their careful reading and for raising important questions about the relationship between our linear analysis and practical diffusion models. We appreciate that the reviewer found our linear simplification reasonable and the RMT application sound. The core concern — that the paper needs stronger evidence connecting the linear theory to trained, nonlinear diffusion models — is well-taken, and we have made substantial efforts to address it. We also take the feedback on paper length seriously.
>
> > **Q1:** *Does the similarity to the linear model rely on the diffusion model being insufficiently trained? When properly trained, does it simply produce similar outputs with fewer artifacts or does it diverge further from the linear model?*
>
> This is a key question. Our expectation is that the linear model captures the "backbone" of consistency, while nonlinear models add refinement on top of this shared backbone.
>
> Following the reviewer's suggestion, we tracked how closely DNN-generated samples align with the linear (Wiener filter) predictor across training, at five dataset scales on FFHQ 64. We re-ran training with an extended schedule (250k steps, up from 50k), and report results up to ~115k steps in Table A (full results pending).
>
> The pattern is consistent across scales: all models initially approach the linear predictor, reaching minimum MSE between steps ~400–4,000 (Table A, "Step @ Min"). After this minimum, behavior differs sharply by generalization vs memorization regime. In the **memorization regime** (N=300, 1,000), MSE rebounds substantially (2.79–2.81× above minimum), indicating the model departs from the linear Gaussian solution as it memorizes the finite training set. In the **generalization regime** (N=10,000, 30,000), the rise is mild (1.08–1.64×) and final MSE stays low (0.063–0.091), indicating the model remains close to the linear predictor throughout training. Across all scales, larger N delays the departure step and reduces the final MSE, with N=30,000 showing nearly no departure (rise ratio 1.08×, final MSE 0.063). N=3,000 sits at the transition, with an intermediate rise ratio of 1.91×. These results directly support our theory: in the generalization regime, the diffusion model's learned mapping stays close to the linear Gaussian solution; departures at small N are signatures of memorization, not failures of the theory. We will update this table with the complete 250k-step results upon completion.
>
> **Table A:** MSE between generated samples (n=1000) and linear predictor throughout training. Final MSE measured ~110k steps.
>
> | N | Min MSE | Step @ Min | Final MSE | Rise Ratio |
> |---|---------|------------|-----------|------------|
> | 300 | 0.093 | ~450 | 0.261 | 2.81× |
> | 1,000 | 0.077 | ~1,100 | 0.214 | 2.79× |
> | 3,000 | 0.064 | ~400 | 0.122 | 1.91× |
> | 10,000 | 0.055 | ~2,300 | 0.091 | 1.64× |
> | 30,000 | 0.058 | ~4,000 | 0.063 | 1.08× |
>
> > **Q2:** *Can you provide a theoretical argument why your linear theory already captures important aspects of diffusion models?*
>
> We offer two arguments: **(I) Inductive bias:** The Wiener filter is optimal for the denoiser loss under linear constraints. If DNNs exhibit inductive bias toward simpler, linear vector fields, they will behave like linear denoisers after learning. **(II) Far-field approximation:** At high noise scale $\sigma$, the score field of a bounded point cloud approaches the linear (Gaussian) score field — the zeroth and first order expansions coincide (cf. Sec 4.1, Wang & Vastola, TMLR 2024).
>
> Crucially, **high noise scales are where high-variance features of samples get determined**. Since the linear theory well-predicts the score at these scales, it captures many high-variance (low-frequency) aspects of samples, while differing mainly in high-frequency details. Empirically, the range of $\sigma$ where linear approximation works extends well beyond the far-field assumption, suggesting that network inductive bias toward linear scores is the key factor.
>
> > **Q3:** *Condense the paper to a more reasonable (under 30 pages) length?*
>
> The theoretical derivation is ~17 pages; supplementary figures occupy ~32 pages, included for completeness and to demonstrate we are not cherry-picking. We include numerous visual examples for this purpose. Upon acceptance, we will condense analysis figures into a polished subset and host extended visual examples on GitHub/a webpage for more interactive viewing, substantially reducing length to around 30 pages.

---

> > ### Author Rebuttal · Reviewer_eYh8 · 2026-04-01
> >
> > The authors response has addressed my concerns well enough and convinced me of a sufficient connection between their linear simplification and practical diffusion models. They have also addressed my concerns regarding the presentation reasonably well.
> > Consequently I am changing my recommendation to 'accept'.

---

> > > ### Author Response · Authors · 2026-04-01
> > >
> > > We are glad that the reviewer thinks our additional longer training experiments convincing.
> > > We appreciate the reviewers for their positive evaluation of our work!!

---

### Official Review · Reviewer_vuve · 2026-03-13

**Soundness:** 4
**Presentation:** 4
**Significance:** 3
**Originality:** 3
**Overall Recommendation:** 5
**Confidence:** 4

**Summary:**

This paper considers the consistency question for linear denoising diffusion models. Specifically, if we use a linear denoiser at each step, if we train the network on two independent data samples from the same distributions. Then starting from the same noise seed the models produce similar images.

The paper analyzes this setting using Random Matrix Theory to provide theoretical evidence for the linear case. There is also some numerical evidence to extend to the non-linear case.

**Compliance With Llm Reviewing Policy:**

Affirmed.

**Key Questions For Authors:**

See weaknesses

**Limitations:**

Discussed

**Strengths And Weaknesses:**

**Strengths**

I think the paper has quite a few strengths.

1. The paper is very well written and easy to follow.

2. The problem studied by the paper is of significant importance.

3. Finally, the results obtained in the linear case are clean, interpretable and interesting. Progressing from one step consistency to consistency of the whole trajectory is an important result.

**Weaknesses**

I think the empirical evidence that tries to extend the analysis to the non-linear case is a bit weak. This is acknowledged in the paper

Second, I wished the paper had a deeper discussion of related works. I believe that there are many works that consider provide theoretical analysis of linear denoisers using random matrix theory (see [A,B,C,D,E]). This leads me to two questions

a. First can such an analysis be extended to deeper linear networks?

b. Can we incorporate the role of training into this analysis?

[A] Raj Rao Nadakuditi, OptShrink: An Algorithm for Improved Low-Rank Signal Matrix Denoising by Optimal, Data-Driven Singular Value Shrinkage (2014).

[B] Chinmaya Kausik, Kashvi Srivastava, and Rishi Sonthalia, Double Descent and Overfitting under Noisy Inputs and Distribution Shift for Linear Denoisers (TMLR 2024).

[C] Hugo Cui and Lenka Zdeborová, High-dimensional asymptotics of denoising autoencoders (NeurIPS 2023).

[D] Jonghyun Ham, Maximilian Fleissner, and Debarghya Ghoshdastidar, Impact of Bottleneck Layers and Skip Connections on the Generalization of Linear Denoising Autoencoders (arXiv:2505.24668, 2025).

[E] Gavish and Donoho, Optimal Shrinkage of Singular Values

---

> ### Author Rebuttal · Authors · 2026-03-30
>
> We thank Reviewer vuve for their thoughtful and positive evaluation. We are glad that the reviewer found the paper well-written, the problem of significant importance, and the results in the linear case clean, interpretable, and interesting — particularly the progression from single-step to full-trajectory consistency. We appreciate the reviewer's constructive suggestions regarding the empirical evidence for the nonlinear case and the pointers to related RMT literature on linear denoisers, which have helped us strengthen the paper. We address each point below.
>
> ### Point 1: Empirical evidence for nonlinear extension is weak
>
> > **[Reviewer]** *The empirical evidence that tries to extend the analysis to the non-linear case is a bit weak, which is acknowledged in the paper.*
> >
>
> We appreciate the reviewer noting that we acknowledged this limitation. Indeed, currently our RMT-based linear theory can predict spatial inhomogeneity of consistency (effect of random seed), spectral effect of consistency (more consistent in high var components), and certain overshrinkage effect of intermediate sample sizes. Indeed, the prediction is statistically significant but not quantitatively accurate (e.g. far from $R^2=1$), esp. that the linear models are orders of magnitude more consistent than the nonlinear neural networks. So we think there is still much work to be done on the theory side, to explain architecture specific nonlinear effects of consistency.
>
> ### Point 2: Deeper discussion of related work on RMT + linear denoisers
>
> > **[Reviewer]** *There are many works that provide theoretical analysis of linear denoisers using random matrix theory [A–E: OptShrink (Nadakuditi 2014), Kausik et al. TMLR 2024, Cui & Zdeborová NeurIPS 2023, Ham et al. 2025, Gavish & Donoho]. These are missing from the discussion.*
> >
>
> We thank the reviewer for these excellent references at the intersection of RMT and denoising. Indeed, the technical tools share common roots in RMT, and we will make these connections explicit in a dedicated related work discussion.
>
> Our contribution differs in focus from these works: [A, E] study optimal singular value shrinkage for denoising, [B] analyzes double descent and generalization under distribution shift, [C] characterizes high-dimensional asymptotics of denoising autoencoders, and [D] examines the effects of bottleneck layers and skip connections on generalization. While the models and techniques are closely related, our work addresses a distinct question — namely, the *cross-dataset consistency* of the full generative trajectory in diffusion models, i.e., why models trained on different data subsets produce similar outputs from the same noise seed. We believe this complements the above line of work and will clearly position our contribution relative to it in the revision.
>
> ### Point 2a: Can the analysis extend to deeper linear networks? Point 2b: Can we incorporate the role of training?
>
> > **[Reviewer]** *Can such an analysis be extended to deeper linear networks?*
> **[Reviewer]** *Can we incorporate the role of training into this analysis?*
> >
>
> This are very interesting and interconnected questions. Since compositions of linear maps collapse to a single linear map, the end-to-end function of a deeper linear network remains linear. So if there is no rank bottleneck, deeper linear networks can represent the optimal linear denoiser (Wiener) solution presented in the paper, so most results from our work should directly apply to deep linear network is we consider the asymptotic case or optimal solution.
>
> The more subtle question is how depth affects the *implicit regularization* via training dynamics — e.g., gradient descent on deep linear networks is known to favor low-rank solutions (Saxe et al., 2014; Arora et al., 2019). One mechanism is that some modes will be learned earlier than others (e.g. spectral bias), so early stopping will result in systematic deviation from the Wiener filter solution. We note that some recent works (Wang, Pehlevan, NeurIPS 2025) discussed the training dynamics of deep linear network under diffusion loss (Appendix F). One limitation of their analysis is that they considered full batch gradient flow, and took the expectation over the whole noised training distribution. Extend their learning dynamics analysis to finite data case and finite noised sample per batch could potentially reveal more accurate learning and batch effect.
>
> Incorporating these learning dynamics based inductive bias into our RMT framework is a natural extension, and we will add a remark pointing towards this direction.

---

> > ### Author Rebuttal · Reviewer_vuve · 2026-04-01
> >
> > Thank you for the response. I maintain my positive accept score for the paper.

---

> > > ### Author Response · Authors · 2026-04-01
> > >
> > > We appreciate the reviewers for their careful reading and positive evaluation of our work!!

---

### Decision · Program_Chairs · 2026-04-30

**Decision:**

Accept (spotlight)

**Comment:**

This paper studies the consistency of diffusion models using a linear Gaussian model and random matrix theory (deterministic equivalence).
The authors show finite data realizations act as renormalization of the noise level and the renormalization effect behaves differently in different noise regimes and along different covariance eigendirections.
The reviewers agreed upon positive ratings. Overall I recommend accept.